# Significance of uncertain phasing between the onsets of stadial-interstadial transitions in different Greenland ice-core proxies

Keno Riechers[1,2] and Niklas Boers[1,2,3]

[1]Department of Mathematics and Computer Science, Freie Universität Berlin, Berlin, Germany
[2]Potsdam Institute for Climate Impact Research, Potsdam, Germany
[3]Department of Mathematics and Global Systems Institute, University of Exeter, Exeter, UK

**Correspondence:** Keno Riechers (riechers@pik-potsdam.de)

**Abstract.** Different paleoclimate proxy records evidence repeated abrupt climate transitions during previous glacial intervals. These transitions have been suggested to be associated with abrupt warming, sudden reorganization of the atmospheric circulation, retreat of perannial sea ice, and increase in local precipitation. The physical mechanism underlying these so-called Dansgaard-Oeschger (DO) events remains debated. A recent analysis of DO events evidenced in Greenland ice core proxy records found that transitions in $Na^+$ concentrations and $\delta^{18}O$ values are delayed by about one decade with respect to corresponding transitions in $Ca^{2+}$ concentrations and the annual layer thickness. These delays are interpreted as a temporal lag of sea ice retreat and Greenland warming with respect to a synoptic- and hemispheric-scale atmospheric reorganization at the onset of DO-events, and may thereby help constraining possible triggering mechanisms for the DO events. The explanatory power of these results is limited, however, by the uncertainty of the transition onset detection in noisy proxy records. Here, we extend previous work by testing the significance of the reported lags with respect to the null hypothesis of no systematically favoured transition sequence of the different proxy variables. If the detection uncertainties are averaged out the temporal delays in the $\delta^{18}O$ and $Na^+$ transitions with respect to their counterparts in $Ca^{2+}$ and the annual layer thickness are indeed pairwise statistically significant. In contrast, under rigorous propagation of uncertainty, according to several tests the null hypothesis cannot be rejected. We thus confirm the previously reported tendency of delayed transitions in the $\delta^{18}O$ and $Na^+$ concentration records. Yet, given the uncertainties in the determination of the transition onsets, it cannot be decided whether these tendencies are truly the imprint of a prescribed transition order or rather due to chance. The analyzed set of DO transitions can therefore not serve as evidence for systematic lead-lag relationships between the transitions in the different proxies, which in turn limits the power of the observed tendencies to constrain possible physical causes of the DO events.

 # 1 Introduction

In view of anthropogenic global warming, concerns have been raised that several subsystems of the earth's climate system may undergo abrupt and fundamental state transitions if temperatures exceed corresponding critical thresholds (Lenton and Schellnhuber, 2007; Lenton et al., 2008, 2019). Under sustained warming, the Atlantic Meridional Overturning Circulation (AMOC), the Amazon rainforest, or the Greenland ice sheet are, among others, possible candidates to abruptly transition to new equilibrium states that may differ strongly from their current states (Lenton et al., 2008). Understanding the physical mechanisms behind abrupt shifts in climatic subsystems is crucial for assessing the associated risks and for defining safe operating spaces in terms of cumulative greenhouse gas emissions. To date, empirical evidence for abrupt climate transitions only comes from paleoclimate proxy records encoding climate variability in the long-term past. First discovered in the $\delta^{18}$O records from Greenland ice cores, the so-called Dansgaard-Oeschger (DO) events are considered the archetype of past abrupt climate changes (see Fig. 1) (Johnsen et al., 1992; Dansgaard et al., 1993; Bond et al., 1993; Andersen et al., 2004). These events constitute a series of abrupt regional warming transitions that punctuated the last and previous glacial intervals at millennial recurrence periods. Amplitudes of these decadal-scale temperature increases reach from $5\,°C$ to $16.5\,°C$ over Greenland (Kindler et al., 2014; Huber et al., 2006; Landais et al., 2005). The abrupt warming is followed by gradual cooling over centuries to millennia, before the climate abruptly transitions back to cold conditions. The relatively cold (warm) intervals within the glacial episodes have been termed Greenland stadials (GS) (Greenland interstadials (GI)). GS typically persist over millennial time scale, before another abrupt warming starts a new cycle (Rasmussen et al., 2014; Ditlevsen et al., 2007). Despite being less pronounced, a global impact of DO events on climate and ecosystems is evident in manifold proxy records (e.g. Moseley et al., 2020; Buizert et al., 2015; Lynch-Stieglitz, 2017; Kim et al., 2012; Fleitmann et al., 2009; Voelker, 2002; Cheng et al., 2013).

Apart from $\delta^{18}$O, other Greenland ice core proxy variables, such as $Ca^{2+}$ and $Na^+$ concentrations as well as the annual layer thickness $\lambda$, also bear the signature of DO cycles, as can be seen in Fig. 1 (e.g., Erhardt et al., 2019; Fuhrer et al., 1999; Ruth et al., 2007). While $\delta^{18}$O is interpreted as a qualitative proxy for ice core site temperatures (e.g. Gkinis et al., 2014; Jouzel et al., 1997; Johnsen et al., 2001), changes in $Ca^{2+}$ concentrations – or equivalently dust – are believed to reflect changes in the atmospheric circulations (Ruth et al., 2007; Erhardt et al., 2019). $Na^+$ concentration records indicate past sea-salt aerosol concentrations and are thought to negatively correlate with the North Atlantic sea ice cover (Erhardt et al., 2019; Schüpbach et al., 2018). The annual layer thickness depends on past accumulation rates at the drilling site and hence indicates local precipitation driven by synoptic circulation patterns (Erhardt et al., 2019). According to this proxy variable interpretation, DO events are found to comprise not only sudden warming, but also sudden increase in local precipitation amounts, retreat of the North Atlantic sea ice cover, and changes of hemispheric circulation patterns.

In the search for the mechanism(s) causing or triggering the DO events, several attempts have been made to deduce the relative temporal order of these abrupt changes by analyzing the phasing of corresponding abrupt shifts detected in multi-proxy time series from Greenland ice cores (Erhardt et al., 2019; Thomas et al., 2009; Steffensen et al., 2008; Ruth et al., 2007). While Thomas et al. (2009) and Steffensen et al. (2008) report delayed Greenland warming with respect to atmospheric changes for the onsets of GI-8 and GI-1 and the Holocene, Ruth et al. (2007) find no systematic lead or lag between NGRIP

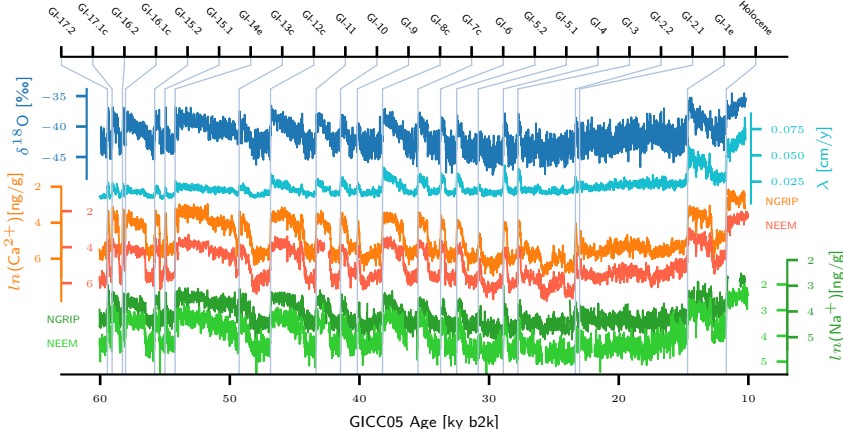

**Figure 1.** Time series of $\delta^{18}$O (blue), annual layer thickness $\lambda$ (cyan), Ca$^{2+}$ (orange), and Na$^+$ (green) from the NGRIP ice core, together with time series of Ca$^{2+}$ (red) and Na$^+$ (light green) from the NEEM ice core on the GICC05 timescale in ky b2k, at 10-year resolution. Light blue vertical lines mark the timings of DO events. The $\delta^{18}$O data and the GICC05 timescale are due to Andersen et al. (2004); Gkinis et al. (2014); Vinther et al. (2006); Rasmussen et al. (2006); Andersen et al. (2006); Svensson et al. (2008). All other time series are retrieved from Erhardt et al. (2019) and for the DO event timings and Greenland Interstadial (GI) notation we followed Rasmussen et al. (2014).

dust concentration and $\delta^{18}$O changes across the onsets of GI-1 to GI-24. However, the comprehensive study conducted by
Erhardt et al. (2019) concludes that on average initial changes in both, terrestrial dust aerosol concentrations (Ca$^{2+}$) and local precipitation ($\lambda$) have preceded the changes in local temperatures ($\delta^{18}$O) and sea salt aerosol concentrations (Na$^+$) by roughly one decade at the onset of DO events during the last glacial cycle.

These observation-based studies are complemented by numerous conceptual theories and modeling studies that explore a variety of mechanisms to explain the DO events. Many studies emphasize the role of the AMOC in the emergence of DO events
(Broecker et al., 1985; Clark et al., 2002; Ganopolski and Rahmstorf, 2001; Henry et al., 2016). In this context, Vettoretti and Peltier (2018) identified a self-sustained sea-salt oscillation mechanism to initiate transitions between stadials and interstadials in simulations of a comprehensive general circulation model run, while Boers et al. (2018) proposed a coupling between sea-ice growth, subsurface-warming, and AMOC changes to explain the DO cycles. Moreover, Li and Born (2019) draw attention to the subpolar gyre, a sensitive region that features strong interactions between atmosphere, ocean and sea ice. In line with the
empirical studies that suggest a delayed Greenland warming with respect to atmospheric changes, Kleppin et al. (2015) and Zhang et al. (2014) find DO-like transitions in model studies triggered by an abrupt reorganization of atmospheric circulation patterns.

Here, we refine the investigation of a potential pairwise lead-lag relationship between the four climate proxies Ca$^{2+}$, Na$^+$, $\delta^{18}$O, and the annual layer thickness $\lambda$ at DO transition onsets, as previously presented by Erhardt et al. (2019), by rigorously
taking into account the uncertainties of the DO onset detection in the different proxy records. We use the same data and the same probabilistic transition onset detection method as provided by Erhardt et al. (2019). The data comprises piece-wise high

resolution (7 years or higher) multi-proxy time series around 23 major DO events for the later half of the last glacial cycle, from the NEEM and the NGRIP ice cores (Erhardt et al., 2019). The fact that high-frequency internal climate variability blurs abrupt transitions limits the ability to precisely detect their onset in the proxy data and thereby constitutes the main obstacle

for the statistical analysis of the succession of events. The method designed by Erhardt et al. (2019) very conveniently takes this into account and instead of returning scalar estimators it quantifies the transition onsets in terms of Bayesian posterior probability densities that indicate the plausibility for a transition onset at a certain time in view of the data. This gives rise to a set of uncertain DO transition onset lags for each pair of proxies under study, whose statistical interpretation is the goal of this study.

While Erhardt et al. (2019) report transition onsets, midpoints, and endpoints, we restrict our investigation to the transition onset points, since we consider the leads and lags between the initial changes in the different proxy records to be the relevant quantity for a potential identification of the physical trigger of the DO events. We extend the previous work by interpreting the sets of uncertain lags as samples generated in random experiments from corresponding unknown populations - each proxy pair is associated with its own population of lags. This allows to investigate whether the reported average lags (Erhardt et al.,

2019) are a systematic feature or whether they might have emerged by chance in a random process that does in fact not favour any of the transition orders. In order to review the statistical evidence for potential systematic lags, we formalize the notion of a 'systematic lag': We call a lag systematic if it is enshrined in the random experiment in form of a population mean different from zero. Samples generated from such a population with non-zero mean would systematically (and not by chance) exhibit sample means different from zero. Accordingly, we formulate the null hypothesis that pairwise no transition sequence is

physically favoured, which corresponds to an underlying population of lags with mean zero. A rejection of this null hypothesis would statistically corroborate the interpretation that transitions in $\delta^{18}$O and Na$^+$ systematically lag their counterparts in $\lambda$ and Ca$^{2+}$. On the other hand, acceptance of the hypothesis would prevent us from ruling out that the observed lag tendencies are not a systematic feature but rather a coincidence. We have identified three different statistical tests suitable for this task, which all rely on slightly different assumptions. Therefore, in combination they yield a robust assessment of the observations. Most

importantly, we propagate the uncertainties that arise from the transition onset detection to the level of $p$-values of the different tests.

We will show that, if the uncertainties are averaged out at the level of the individual transition onset lags – thus ignoring the uncertainties in the onset detection – all tests indicate statistical significance (at 5% confidence level) of the observed tendencies toward delayed $\delta^{18}$O and Na$^+$ transition onsets with respect to the corresponding onsets in $\lambda$ and Ca$^{2+}$. Rigorous

uncertainty propagation yields, however, substantial probabilities for the derived transition onset lags to be non-significant with respect to the null hypothesis. We thus argue that the uncertainties in the transition onset detection are too large to infer a mean different from zero in the underlying lag population. In turn, this prevents the attribution of the observed lead-lag relations to a fundamental mechanism underlying the DO events. We discuss the difference between our approach and the one followed by Erhardt et al. (2019) in detail below.

In addition to the quantitative uncertainty discussed here, there is always qualitative uncertainty about the interpretation of climate proxies. Clearly, there is no one-to-one mapping between proxy variables and the climate variables they are assumed

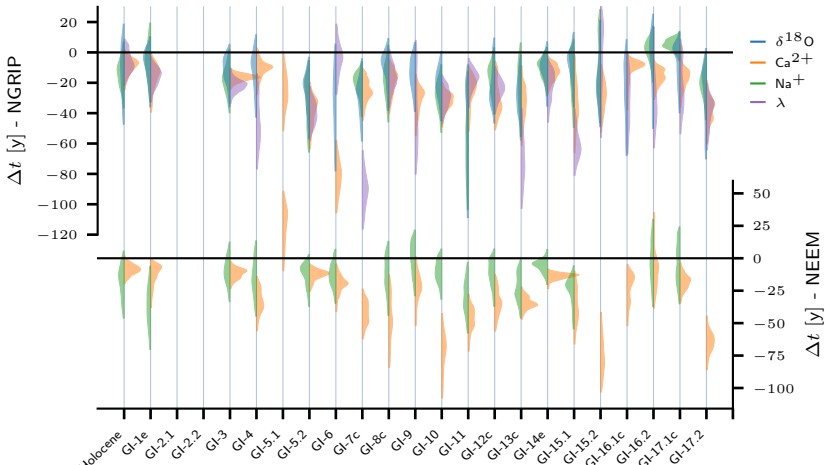

**Figure 2.** DO events (Greenland Interstadial onsets) for which Erhardt et al. (2019) provide high-resolution proxy data ($Ca^{2+}$, $Na^+$, and $\lambda$) for windows centered around the transitions. $\delta^{18}O$ data for the corresponding windows was retrieved from continuous $\delta^{18}O$ time series measured in 5cm-steps in the NGRIP ice core (see Fig.1). The posterior probability densities for the transition onsets with respect to the timing of the DO events according to Rasmussen et al. (2014) are shown in arbitrary units for all proxies. They were recalculated using the data and the method provided by Erhardt et al. (2019). The uncertain transition onsets are only shown for those transitions investigated in this study - the selection is adopted from Erhardt et al. (2019) to guarantee comparability.

to represent. To give an example, changes in the atmospheric circulation will simultaneously impact the transport efficiency of sea-salt aerosols to Greenland. Schüpbach et al. (2018) discuss in detail the entanglement of transport efficiency changes and source emission changes for aerosol proxies measured in Greenland ice cores. We restrict our analysis to those proxy pairs that have been found to show decadal-scale time lags by Erhardt et al. (2019) and leave aside those pairs which show almost simultaneous transition onsets according to Erhardt et al. (2019).

This article is structured as follows: First, the data used for the study is described. Second, we introduce our methodology in general terms, in order to facilitate potential adaptation to structurally similar problems. Within this section, we pay special attention to clarifying the differences between the approaches chosen in this study and by Erhardt et al. (2019). This is followed by the presentation of our results including a comparison to previous results. In the subsequent discussion, we give a statistical interpretation and explain how the two lines of inference lead to different conclusions. The last section summarizes the key conclusions that can be drawn from our analysis.

## 2 Data

In conjunction with their study, Erhardt et al. (2019) published 23 highly resolved time series for $Ca^{2+}$ and $Na^+$ concentrations from the NGRIP and NEEM ice cores for time intervals of 250 to 500 years centered around DO events from the later half of

the last glacial. The data set covers all major interstadial onsets from GI-17.2 to the Holocene, as determined by Rasmussen et al. (2014). The time resolution decreases from 2 to 4 years with increasing depth in the ice cores due to the thinning of the core. In addition, Erhardt et al. (2019) derived the annual layer thickness from the NGRIP aerosol data and published these records likewise for the time intervals described above. Furthermore, continuous 10-year resolution versions of the proxy records were published, which cover the period 60-10 kyr BP, shown in Fig. 1 (Erhardt et al., 2019). Finally, the NGRIP $\delta^{18}$O record at 5 cm resolution (corresponding to 4-7 years for the respective time windows) (Andersen et al., 2004) completes the dataset used in the study by Erhardt et al. (2019) and correspondingly in our study.

While $Ca^{2+}$ and $Na^+$ mass concentrations are interpreted as indicators of the past state of the atmospheric large-scale circulation and the past North Atlantic Sea ice extent, respectively, the annual layer thickness and $\delta^{18}$O records give qualitative measures of the local precipitation and temperature, respectively (Erhardt et al., 2019, and references therein). The high resolution and the shared origin of the time series makes them ideally suited to study the succession of events at the beginning of DO transitions. On top of that, the aerosol data have been co-registered in a continuous flow analysis allowing for highest possible comparability (Erhardt et al., 2019).

For their analysis, Erhardt et al. (2019) only considered time series around DO events that do not suffer from substantial data gaps. For the sake of comparability, we adopt their selection. From Fig. 2 it can be inferred which proxy records around which DO events have been included in this study. For details on the data and the proxy interpretations we refer to Erhardt et al. (2019) and the manifold references therein.

## 3  Methods

We first briefly review the probabilistic method that we adopted from Erhardt et al. (2019) in order to estimate the transition onset time $t_0$ of each proxy variable for each DO event comprised in the data (see Fig. 3). The Bayesian method accounts for the uncertainty inherent to the determination of $t_0$ by returning probability densities $\rho_{T_0}(t_0)$ instead of scalar estimators. From these distributions, corresponding probability distributions for the pairwise time lags between two proxies can be derived for all DO events. Second, a statistical perspective on the series of DO events is established. For a given proxy pair, the set of transition onset lags from the different DO events is treated as a sample of observations from an unknown underlying population. In this very common setup, naturally one would use hypothesis tests to constrain the population. In particular, the question whether any lag tendencies observed in the data are a systematic feature or whether they have instead occurred by chance can be assessed by testing the null hypothesis of a population mean lag equal to zero. However, the particularity that the individual observations that comprise the sample are themselves subject to uncertainty requires a generalization of the hypothesis tests. We propagate the uncertainty of the transition onset timings to the $p$-values of the tests and hence obtain uncertain $p$-values in terms of probability densities (see Fig. 4). While in common hypothesis tests the scalar $p$-value is compared to a predefined significance level, here we propose two criteria to project the $p$-value distribution onto the binary decision between acceptance and rejection of the null hypothesis. After this general characterization of the statistical problem, we introduce the tests which we employ for the analysis. Finally, we compare our approach to the one followed by Erhardt et al. (2019).

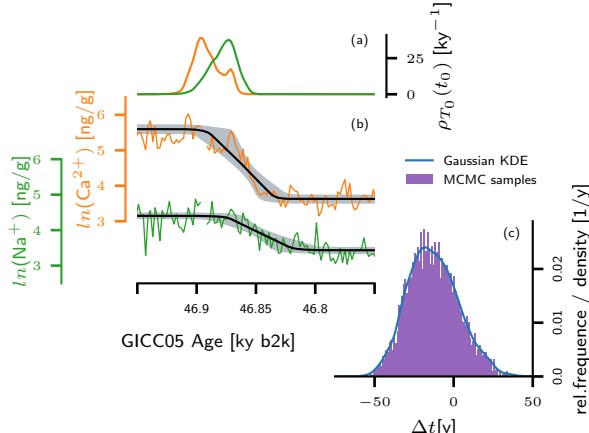

**Figure 3.** (a) Posterior probability distribution $\rho_{T_0}(t_0)$ for the onset of NGRIP $Ca^{2+}$ and $Na^+$ transitions associated with the onset of GI-12c, derived from $Ca^{2+}$ (orange) and $Na^+$ (green) values around the GI-12c onset at 2-year resolution, using the probabilistic ramp-fitting shown in (b). The black lines in (b) indicate the expected ramp, i.e., the average over all ramps determined by the posterior distributions of the ramp parameters. The grey shaded area indicates the 5th-95th percentiles of these ramps. (c) Histogram sampled from the posterior distribution for the transition onset lag $\Delta t$ between the two proxies (violet), together with the corresponding Gaussian kernel density estimate (KDE, blue).

### 3.1 Transition onset detection

Consider a fluctuating time series $\mathcal{D} = \{x(t_i)\}_{i=1,...,n}$ with $n$ data points, which includes one abrupt transition from one level of values to another, as shown in Fig. 3(b). For this setting, Erhardt et al. (2019) have designed a method to estimate the transitions onset time $t_0$ in a probabilistic, Bayesian sense. Instead of a point estimate, their method returns a so-called posterior probability density that indicates the plausibility of the respective onset time in view of the data (see Fig. 3(a)). For technical reasons, this probability density cannot be derived in form of a continuous function but only in form of a representative set

of values generated from it by means of a Markov-Chain-Monte-Carlo (MCMC) algorithm (Goodman and Weare, 2010). The application of the method to NGRIP $Ca^{2+}$ and $Na^+$ concentration data around the onset of GI-12c is illustrated in Fig. 3.

The key idea is to model the transition as a linear ramp $\mathcal{L}(t_i)$ perturbed by Gaussian red noise $\epsilon(t_i)$:

$$y(t_i) = \underbrace{\begin{cases} y_0 & t_i \leq t_0 \\ y_0 + \Delta y \, \frac{t_i - t_0}{\tau} & t_0 < t_i < t_0 + \tau \\ y_0 + \Delta y & t \geq t_i + \tau \end{cases}}_{\text{liner ramp } \mathcal{L}(t_i)} + \underbrace{\text{AR}(1)_{\sigma,\alpha}}_{\text{red noise } \epsilon(t_i)} . \tag{1}$$

This model is fully determined by the four ramp parameters $\{t_0, y_0, \tau, \Delta y\}$, the amplitude $\sigma$, and the autoregressive coefficient

$\alpha$ of the AR(1) process. For a given configuration $\theta$ of these six parameters, the probability for this stochastic model to exactly

reproduce the data $\mathcal{D}$ reads

$$\pi(\mathcal{D}|\theta) := \pi(y(t_i) = x(t_i) \forall i \in \{1,\ldots,n\}|\theta) \qquad\qquad = \frac{1}{(2\pi\sigma^2)^n} \prod_{i=1}^{n} \exp\left(-\frac{1}{2}\frac{(\delta_i - \alpha\delta_{i-1})^2}{\sigma^2}\right), \tag{2}$$

where $\delta_i = x(t_i) - \mathcal{L}(t_i)$ denote the residuals between the linear ramp and the observations and $\delta_0 = 0$. Bayes' Theorem immediately yields the posterior probability density for the model parameters $\pi(\theta|\mathcal{D})$ upon introduction of convenient priors $\pi(\theta)$:

$$\pi(\theta|\mathcal{D}) = \frac{\pi(\mathcal{D}|\theta)\,\pi(\theta)}{\pi(\mathcal{D})}, \tag{3}$$

where the normalization constant $\pi(\mathcal{D}) = \int \pi(\mathcal{D}|\theta)\pi(\theta)d\theta$ is the a priori probability for the observations. Since the parameter space is six-dimensional, Eq. 3 cannot be evaluated explicitly on a grid with reasonably fine spacing. Instead, an MCMC-algorithm is used to sample a representative set $\{\theta_1,\ldots,\theta_m\}$ of parameter configurations from the posterior distribution that approximates the continuous distribution in the sense that for smooth functions $f$

$$\int f(\theta)\rho_\Theta(\theta)\,d\theta \simeq \int f(\theta)\bar{\rho}_\Theta(\theta)\,d\theta = \frac{1}{m}\sum_{j=1}^{m} f(\theta_j), \tag{4}$$

where the notion of a so-called empirical distribution $\bar{\rho}_\Theta(\theta) = \frac{1}{m}\sum_{j=1}^{m}\delta(\theta - \theta_j)$ has been used. The use of the MCMC algorithm further allows to omit the normalization constant $\pi(\mathcal{D})$. The number $m$ of individuals comprised in the MCMC sample must be chosen large enough to ensure a good approximation in Eq. 4. The marginal distribution for the parameter $t_0$ relevant for our study can be obtained by integration over the remaining parameters $\theta^*$:

$$\rho_{T_0|\mathcal{D}}(t_0) = \int \pi(\theta|\mathcal{D})\,d\theta^*, \tag{5}$$

which reads

$$\bar{\rho}_{T_0}(t_0) = \frac{1}{m}\sum_{j=1}^{m}\delta(t_0 - t_{0,j}) \tag{6}$$

in terms of the empirical density induced by the MCMC sample.

Given the probability densities for the transition onsets of two proxy variables $p$ and $q$ at a chosen DO event $i$, the probability density for the lag $\Delta t_i^{p,q} = t_0^{p,i} - t_0^{q,i}$ between them reads

$$\rho_{\Delta T_i^{p,q}}(\Delta t_i^{p,q}) = \int\int \delta(t_0^{p,i} - t_0^{q,i} - \Delta t_i^{p,q})\rho_{T_0}^{p,i}(t_0^{p,i})\,\rho_{T_0}^{q,i}(t_0^{q,i})\,dt_0^{p,i}\,dt_0^{q,i}. \tag{7}$$

$\Delta T_i^{p,q}$ was chosen to denote the time lag which inherits the uncertainty from the transition onset detection and must thus mathematically be treated as a random variable. $\Delta t_i^{p,q}$ denotes a potential value that $\Delta T_i^{p,q}$ may assume. The set of probability densities $\{\rho_{\Delta T_i^{p,q}}(\Delta t_i^{p,q})\}_i$ derived from the different DO events conveniently describes the random vector of uncertain DO onset lag observations $\mathbf{\Delta T}^{p,q} = (\Delta T_1^{p,q},\ldots,\Delta T_n^{p,q})$ for the $(p,q)$ proxy pair in the sense that

$$\rho_{\mathbf{\Delta T}^{p,q}}(\mathbf{\Delta t}^{p,q}) = \prod_{i=1}^{n}\rho_{\Delta T_i^{p,q}}(\Delta t_i^{p,q}). \tag{8}$$

Note that the entries $\Delta T_i^{p,q}$ of the random vector $\boldsymbol{\Delta T}^{p,q}$ are independent from each other and follow their individual distributions $\rho_{\Delta T_i^{p,q}}(\Delta t_i^{p,q})$, such that the joint distribution is given by the product of the individual distributions. A cross-core comparison is not possible, because the relative dating uncertainties between the cores exceed the magnitude of the potential time lags.

For sake of simplicity, we omit the difference between the posterior density distribution and the empirical posterior density distribution. It is shown in Appendix A that all methods can be equivalently formulated in terms of the empirical posterior density distribution. The numerical computations themselves have of course been carried out with the empirical densities obtained from the MCMC sampler. Appendix B discusses the construction of numerically manageable empirical densites $\bar{\rho}_{\boldsymbol{\Delta T}^{p,q}}(\boldsymbol{\Delta t}^{p,q})$. Since substantial reduction of the available MCMC sampled data is required, a control group of alternative realizations of $\bar{\rho}_{\boldsymbol{\Delta T}^{p,q}}(\boldsymbol{\Delta t}^{p,q})$ is introduced. The high agreement of the results obtained from the control group with the results discussed in the main text confirms the validity of the initial $\bar{\rho}_{\boldsymbol{\Delta T}^{p,q}}(\boldsymbol{\Delta t}^{p,q})$ construction.

In the following all probability densities that represent uncertainties with origin in the transition onset observation will be referred to as uncertainty distributions or uncertainty densities. This facilitates to differentiate these from probability distributions that generically characterize random experiments. The random variables described by uncertainty distributions will be termed uncertain variables and will be marked with a hat. Generally, we denote all random (uncertain) variables by capital letters $X$ $(\hat{X})$, while realizations will be denoted with lower case letters $x$ $(\hat{x})$. Furthermore, distributions will always be subscripted with the random variables that they characterize, e.g. $\rho_X(x)$ $(\rho_{\hat{X}}(\hat{x}))$. For sake of readability, sometimes we omit the index $p,q$ when it is clear that a quantity refers to a pair of proxies $(p,q)$.

## 3.2 Statistical setting

Despite their diversity in terms of temperature amplitude, duration, and frequency across the last glacial, the reoccurring patterns and their common manifestation in different proxies suggest that the DO events follow a common physical mechanism. If this assumption holds true, this mechanism prescribes a fixed pattern of causes and effects for all DO events - at least on the scale of interactions between climatic subsystems represented by the proxies under study. However, natural variability will randomly delay or advance the individual parts of the event chain of the DO mechanism in each single realization, without violating the mechanistic causality. The observed pairwise transition onset lags can thus be regarded as realizations of independent and identically distributed (i.i.d.) random variables generated in a random experiment $(\Omega, \mathcal{F}, \mathcal{P}_{\Delta T}^{p,q})$ on the sample space $\Omega = \mathbb{R}$. Here, $\mathcal{F}$ is a $\sigma$-algebra defined on $\Omega$ and may be taken as the Borel algebra. $\mathcal{P}_{\Delta T}^{p,q}$ – the so-called population – denotes a probability measure with respect to $\mathcal{F}$ and fully characterizes the random lag $\Delta T^{p,q}$ between the proxies $p$ and $q$. Importantly, if any of the proxy variables investigated here was to represent a climate variable associated with the DO event trigger, we would expect an advanced initial change in the record of this proxy with respect to other proxies at DO events. In turn, a pronounced delay of a proxy record's transition onset contradicts the assumption that the proxy represents a climate variable associated with the trigger. Therefore, the identification of leads and lags between the transition onsets in the individual proxy time series may help in the search for the trigger of the DO events. Here, we formalize these arguments by testing whether the observed samples of transition lags are significant in a statistical sense with respect to the null hypothesis of a population

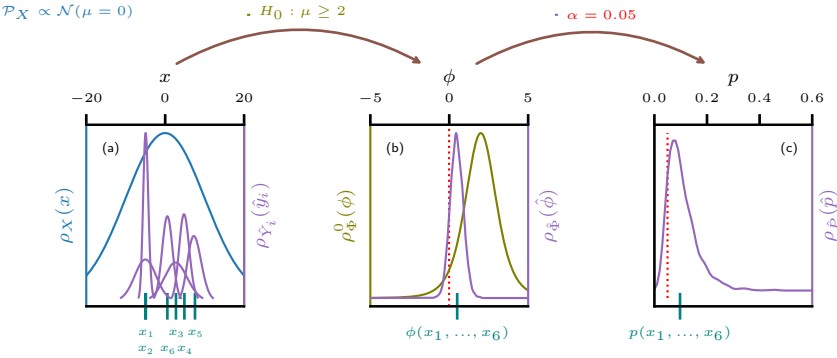

**Figure 4.** (a) Schematic representation of an uncertain observation of a sample (purple) generated from a population (blue) in a random experiment. The blue line indicates the probability density of the generating population $\mathcal{P}_X$. Turquoise lines indicate the true value of a sample $\mathbf{x} = (x_1, ..., x_6)$ realized from $\mathcal{P}_X$. If the observational process involves uncertainty, a second level of randomness is introduced and the values can at best be approximated by probability density functions depicted in purple. These uncertainty distributions indicate the informed estimate of the observer about how likely a certain value $\hat{y}_i$ for the estimator $\hat{Y}_i$ is to coincide with the true value $x_i$. Depending on the measurement process, the uncertainty distributions of the sample members may all exhibit individual shapes or they may share a common one. (b) Distribution of the uncertain test statistic $\hat{\Phi} = \phi(\hat{\mathbf{Y}})$ derived from the uncertain sample (purple) together with the corresponding value derived from the true sample (turquoise). In olive, the distribution of $\Phi$ under the null hypothesis is shown. The dotted red line separates the rejection region (left) from the acceptance region in a one-sided test setup. (c) Distribution of the uncertain $p$-value corresponding to the uncertain sample. In turquoise, the $p$-value of the certain sample is marked. The red line indicates the significance level $\alpha$.

with mean equal to zero. If this null hypothesis can be rejected based on the observations, this would constitute a strong hint for a systematic, physical lag, and would hence potentially yield valuable information on the search for the mechanism(s) and trigger(s) of the DO transitions.

According to the data selection by Erhardt et al. (2019) as explained in Sec. 2, for all studied pairs of proxies we compute either 16 or 20 transition lags from the different DO events, which we interpret as samples $\Delta\mathbf{t}^{p,q} = (\Delta t_1^{p,q}, ..., \Delta t_n^{p,q})$ from their respective populations $\mathcal{P}_{\Delta T}^{p,q}$.[1] Studying these samples, Erhardt et al. (2019) deduced a decadal-scale delay in the transition onsets in $Na^+$ and $\delta^{18}O$ records with respect to their counterparts in $Ca^{2+}$ and $\lambda$. In order to test if the data supports evidence for this lag to be systematic in a statistical sense, first the notion of a 'systematic lag' requires mathematical formalization. We

argue that a physical process that systematically delays one of the proxy variable transitions with respect to another must in the random experiment framework be associated with a population that exhibits a mean different from zero $\mu^{p,q} = E(\Delta T^{p,q}) \neq 0$. The outcomes of such a random experiment will systematically exhibit sample means different from zero in accordance with the population mean. Samples generated from a population with mean equal to zero may as well yield sample means that strongly differ from zero. However, their occurrence is not systematic but rather a coincidence. Given a limited number of

---

[1]In fact, the pairwise lags between the different proxies of the NGRIP ice core are certainly correlated and one could formulate the random experiment in terms of a multi-dimensional population where each component represents the time lag of a different proxy pair. Associating each time lag $\Delta T^{p,q}$ with its individual population technically corresponds to an investigation of the marginal distributions of $\mathcal{P}_{\Delta T}$.

observations, hypothesis tests provide a consistent, yet not unambiguous way to distinguish systematic from random features. If the mean of the observed sample $u^{p,q}(\mathbf{\Delta t^{p,q}}) = \frac{1}{n} \sum \Delta t_i^{p,q}$ indicates an apparent lag between the proxies $p$ and $q$, testing if the sample statistically contradicts a population that favours no ($\mu^{p,q} = 0$) or the opposite lag (or $\text{sign}(\mu^{p,q}) \neq \text{sign}(u^{p,q})$) can provide evidence at the significance level $\alpha$ for the observed mean lag to be systematic in the sense that $\text{sign}(\mu^{p,q}) = \text{sign}(u^{p,q})$. However, as long as the null hypothesis cannot be rejected, the observed average sample lag cannot be regarded as statistical

evidence for a systematic lag.

Before we introduce the tests deployed for this study, we discuss the particularity that the individual observations of the i.i.d. variables that comprise our samples are themselves subject to uncertainty and hence are represented by probability densities instead of scalar values. The common literature on hypothesis tests assumes that an observation of a random variable yields a scalar value.[2] Given a sample of $n$ scalar observations

$$\mathbf{x} = (x_1, x_2, ..., x_n). \tag{9}$$

the application of hypothesis tests to the sample is in general straight forward and has been abundantly discussed (e.g. Lehmann and Romano, 2006). In short, a test statistic $\phi_{\mathbf{x}} = \phi(\mathbf{x})$ is computed from the observed sample, where

$$\phi : \mathbb{R}^n \to \mathbb{R}, \quad \mathbf{x} \mapsto \phi(\mathbf{x}) \tag{10}$$

denotes the mapping from the space of $n$-dimensional samples to the space of the test statistic and $\phi_{\mathbf{x}}$ denotes the explicit value

of the function when applied to the observed sample $\mathbf{x}$. Subsequently, integration of the so-called null distribution over all values $\phi'$, which under the null hypothesis $H_0$ are more extreme than the observed $\phi_{\mathbf{x}}$, yields the test's $p$-value. In this study, an hypothesis on the lower limit of a parameter will be tested. In this one-sided left-tailed application of hypothesis testing, the $p$-value explicitly reads

$$p_{\mathbf{x}} = \int\limits_{-\infty}^{\phi_{\mathbf{x}}} \rho_\Phi^0(\phi') \, d\phi', \tag{11}$$

which defines the mapping:

$$p : \mathbb{R} \to [0,1], \quad \phi_{\mathbf{x}} \mapsto p(\phi_{\mathbf{x}}) = p_{\mathbf{x}}. \tag{12}$$

Analogous expressions may be given for one-sided right-tailed and two-sided tests. The null distribution $\rho_\Phi^0(\phi')$ is the theoretical distribution of the random test statistic $\Phi = \phi(\mathbf{X})$ under the assumption that the null hypothesis on the population $\mathcal{P}_X$ holds true. If $p_{\mathbf{x}}$ is less than a predefined significance level $\alpha$, the observed sample $\mathbf{x}$ is said to contradict the null hypothesis at

265 the significance level $\alpha$, and the null hypothesis should be rejected.

In contrast to this setting, the DO transition onset lags $\Delta t_i^{p,q}$ between the proxies $p$ and $q$, which are thought to have been generated from the population $\mathcal{P}_{\Delta T^{p,q}}$, are observed with uncertainty. In our case, the entries in the vector of observations are

---

[2]In fact, the problem that a random variable that assumes values in the real numbers can never be observed exactly is well-known and so-called fuzzy numbers may be employed to express the associated imprecision. However, this fundamental imprecision is different from the measurement uncertainty that we are facing, and thus the concept of fuzzy number is not applicable to our problem (Filzmoser and Viertl, 2004).

uncertain variables themselves, which are characterized by the previously introduced uncertainty distributions $\rho_{\Delta \hat{T}_i^{p,q}}(\Delta \hat{t}_i^{p,q})$.
The left panel in Fig. 4 illustrates this situation: from an underlying population $\mathcal{P}_X$ a sample $\mathbf{x} = (x_1, ..., x_6)$ is realized, with
$x_i$ denoting the true values of the individual realizations. The exact value of $x_i$ can, however, not be measured precisely due
to measurement uncertainties. Instead, an estimator $\hat{Y}_i$ is introduced together with the uncertainty distribution $\rho_{\hat{Y}_i}(\hat{y}_i)$ that
expresses the observers belief about how likely a specific value $\hat{y}_i$ for the estimator $\hat{Y}_i$ is to agree with the true value $x_i$. The $\hat{Y}_i$
correspond to the $\Delta \hat{T}_i^{p,q}$. For the $x_i$ there is no direct correspondence in the problem at hand, because this quantity cannot be
accessed in practice and is thus not denoted explicitly. We call the vector of estimators $\hat{\mathbf{Y}} = (\hat{Y}_1, ..., \hat{Y}_n)$ an uncertain sample
in the following. Omitting the $(p, q)$ notation, we denote an uncertain sample of time lags as

$$\Delta \hat{\mathbf{T}} = \left( \Delta \hat{T}_1, \Delta \hat{T}_2, ..., \Delta \hat{T}_n \right), \quad \text{with} \tag{13a}$$

$$\rho_{\Delta \hat{\mathbf{T}}}(\Delta \hat{\mathbf{t}}) = \prod_{i=1}^{n} \rho_{\Delta \hat{T}_i}(\Delta \hat{t}_i). \tag{13b}$$

Note that the uncertainty represented by the uncertain sample originates from the observation process - the sample no longer
carries the generic randomness of the population $\mathcal{P}_{\Delta T}$ it was generated from. The $\Delta \hat{T}_i$ are no longer identically but yet
independently distributed.

A simplistic approach to test hypotheses on an uncertain sample would be to average over the uncertainty distribution and
subsequently apply the test to the resulting expected sample

$$\mathrm{E}(\Delta \hat{\mathbf{T}}) = \left( \mathrm{E}(\Delta \hat{T}_1), ..., \mathrm{E}(\Delta \hat{T}_n) \right) \qquad = \left( \int \Delta \hat{t}_1 \, \rho_{\Delta \hat{T}_1}(\Delta \hat{t}_1) \, d\Delta \hat{t}_1, \, ... \, , \int \Delta \hat{t}_n \, \rho_{\Delta \hat{T}_n}(\Delta \hat{t}_n) \, d\Delta \hat{t}_n \right). \tag{14}$$

Averaging out uncertainties, however, essentially implies that the uncertainties are ignored and is thus always associated with
a loss of information. The need for a more thorough treatment, with proper propagation of the uncertainties, may be illustrated
with a simple example. Consider a sample $\mathbf{x} = (x_1, ..., x_n)$ that was generated from a population $\mathcal{P}_X$ and measured with very
high precision such that the $\rho_{\hat{Y}_i}(\hat{y}_i)$ can be assumed to be $\delta$-peaks around the values $\hat{y}_i$. Assume that, for some reason, after the
measurements have been carried out, the observer is unsure about the sign of the observed values. For example, in a voltage
measurement one might have confused plus and minus. In this case, the joint uncertainty distribution allocates 50% probability
to the value $(\hat{y}_1, ..., \hat{y}_n)$ and 50% probability to the value $(-\hat{y}_1, ..., -\hat{y}_n)$. The expected sample $\mathrm{E}(\hat{\mathbf{Y}}) = 0$ is obviously not
useful to test hypotheses on the population $\mathcal{P}_X$. In contrast, uncertainty propagation gives rise to an uncertainty distribution
of the $p$-value, which indicates the plausibility of a certain $p$-value in view of the data and in view of the limitations in the
transition onset detection. The propagation of the uncertainties from the level of observation to the test statistic and finally to
the $p$-value is illustrated in Fig. 4.

The uncertainty propagation relies on the fact that applying a function $f : \mathbb{R} \to \mathbb{R}$ to a real valued random (uncertain) variable
$X$ yields a new random (uncertain) variable $G = f(X)$, which is distributed according to

$$\rho_G(g) = \int \delta \left( f(x) - g \right) \rho_X(x) \, dx. \tag{15}$$

Analogously, the uncertain test statistic $\hat{\Phi} = \phi(\Delta \hat{\mathbf{T}})$ follows the distribution

$$\rho_{\hat{\Phi}}(\hat{\phi}) = \int \delta(\phi(\Delta \hat{\mathbf{t}}) - \hat{\phi}) \, \rho_{\Delta \hat{\mathbf{T}}}(\Delta \hat{\mathbf{t}}) \, d\Delta \hat{\mathbf{t}}. \tag{16}$$

Repeated application of Eq. 15 yields the uncertainty distribution of a given test's $p$-value $\hat{P} = p(\phi(\boldsymbol{\Delta}\hat{\mathbf{T}}))$:

$$\rho_{\hat{P}}(\hat{p}) = \int \delta\left(p(\hat{\phi}) - \hat{p}\right) \rho_{\hat{\Phi}}(\hat{\phi}) \, d\hat{\phi} = \int \int \delta\left(p(\hat{\phi}) - \hat{p}\right) \delta\left(\phi(\boldsymbol{\Delta}\hat{\mathbf{t}}) - \hat{\phi}\right) \rho_{\boldsymbol{\Delta}\hat{\mathbf{T}}}(\boldsymbol{\Delta}\hat{\mathbf{t}}) \, d\boldsymbol{\Delta}\hat{\mathbf{t}} \, d\hat{\phi}$$

$$= \int \delta\left(p(\phi(\boldsymbol{\Delta}\hat{\mathbf{t}})) - \hat{p}\right) \, d\boldsymbol{\Delta}\hat{\mathbf{t}}. \quad (17)$$

In the example shown in Fig. 4 the initial uncertainties in the observations translate into an uncertain $p$-value that features both,
probability for significance and probability for non-significance. This illustrates the need for a criterion to project the uncertain
$p$-value onto a binary decision space comprised of rejection and acceptance of the null hypothesis. We propose to consider the
following criteria to facilitate an informed decision:

    – *The hypothesis shall be rejected at the significance level $\alpha$ if and only if the expected $p$-value is less than $\alpha$, that is*

$$\int_0^1 \hat{p} \, \rho_{\hat{P}}(\hat{p}) \, d\hat{p} < \alpha. \quad (18)$$

– *The hypothesis shall be rejected at the significance level $\alpha$ if and only if the probability for $p$ to be less than $\alpha$ is greater than a predefined threshold $\eta$ (we propose $\eta = 90\%$), that is*

$$\pi(\hat{P} < \alpha) = \int_0^\alpha \rho_{\hat{P}}(\hat{p}) \, d\hat{p} > \eta. \quad (19)$$

While the $p$-value of a certain sample indicates its extremeness with respect to the null distribution, the expected $p$-value may
be regarded as a measure of the uncertain sample's extremeness. Given the measurement uncertainty, the quantity $\pi(\hat{P} < \alpha)$
indicates the informed estimate of the observer that the true value of the measured sample is in fact statistically significant with
respect to the null hypothesis. Thus, the first criterion assesses how 'strongly' the uncertain sample contradicts the null hypoth-
esis, while the second criterion evaluates the likelihood of the uncertain sample to contradict the null hypothesis. Depending
on $\eta$, in many cases both criteria will yield the same decision. If not, the specific situation determines which of the criteria is
more convenient. Under some circumstances one might want to guarantee that in fact the probability to achieve a significant
test result is high – e.g. when mistakenly attested significance is associated with high costs. In these cases the second criterion
is more appropriate even though it does not imply that the first criterion is fulfilled.

## 3.3   Hypothesis tests

We have introduced the notion of uncertain samples and its consequences for the application of hypothesis tests. Here, we
shortly introduce the tests used to test our null hypothesis that the observed tendency for delayed transition onsets in $Na^+$ and
$\delta^{18}O$ with respect to $Ca^{2+}$ and $\lambda$ has occurred by chance and that the corresponding populations $\mathcal{P}_{\Delta T}^{p,q}$ that characterize the
pairwise random lags $\Delta T^{p,q}$ do in fact not favour the tentative transition orders apparent from the observations. Mathematically,
this can be formulated as follows:

- Let $\rho_{\Delta T}^{p,q}(\Delta t)$ be the probability density associated with the popuplation of DO transition onset lags $\mathcal{P}_{\Delta t}^{p,q}$ between the proxy variables $p$ and $q$ and let the observations $\Delta \hat{\mathbf{T}}^{p,q}$ suggest a delayed transition of the proxy $q$ - that is, the corresponding uncertainty distributions $\rho_{\Delta \hat{T}_i^{p,q}}(\Delta \hat{t}_i^{p,q})$ indicate high probabilities for negative $\Delta \hat{T}_i$ across the sample according to Eq. 7. We then test the hypothesis $H_0$ : 'The mean value $\mu^{p,q} = \int \rho_{\Delta T}^{p,q}(\Delta t)\, d\Delta t$ of the population $\mathcal{P}_{\Delta T}^{p,q}$ is greater or equal than zero.'

We identified three tests that are suited for this task, namely the $t$-test, the Wilcoxon-signed-rank (WSR) test, and a bootstrap test. The WSR and the $t$-test are typically formulated in terms of paired observation $\{x_i, y_i\}$ that give rise to a sample of differences $\{d_i = x_i - y_i\}$ which correspond to the time lags $\{\Delta t_i^{p,q}\}$ of different DO events (Rice, 2007; Lehmann and Romano, 2006, e.g.). The null distributions of the tests rely on slightly different assumptions regarding the populations. Since we cannot guarantee the compliance of these assumptions, we apply the tests in combination to obtain a robust assessment.

### 3.3.1 $t$-test

The $t$-test (Student, 1908) relies on the assumption that the population of differences $\mathcal{P}_D$ is normally distributed with mean $\mu$ and standard deviation $\sigma$. For a random sample $\mathbf{D} = (D_1, ..., D_n)$ the test statistic

$$Z(\mathbf{D}) = \frac{U(\mathbf{D}) - \mu}{S(\mathbf{D})/\sqrt{n}} \tag{20}$$

follows a $t$-distribution $t_{n-1}(z)$ with $n-1$ degrees of freedom. Here, $U = \frac{1}{n}\sum D_i$ is the sample mean and $S = \frac{1}{n-1}\sum(U - D_i)^2$ is the samples' standard deviation. This allows to test whether an observed sample $\mathbf{d} = (d_1, ..., d_n)$ contradicts an hypothesis on the mean $\mu$. To compute the $p$-value for the hypothesis $H_0 : \mu \geq 0$ (left handed application) the null distribution is integrated from $-\infty$ to the observed value $z(\mathbf{d})$:

$$p_z(z(\mathbf{d})) = \int\limits_{-\infty}^{z(\mathbf{d})} t_{n-1}(z')dz'. \tag{21}$$

The resulting $p$-value must then be compared to the predefined significance level $\alpha$.

The $t$-test can be generalized for application to an uncertain sample of the form $\Delta \hat{\mathbf{T}} = (\Delta \hat{T}_1, ..., \Delta \hat{T}_n)$ as follows: Let $\rho_{\Delta \hat{\mathbf{T}}}(\Delta \hat{\mathbf{t}})$ denote the uncertainty distribution of $\Delta \hat{\mathbf{T}}$. Then according to Eq. 15 the distribution of the uncertain statistic $\hat{Z}(\Delta \hat{\mathbf{T}})$ reads

$$\rho_{\hat{Z}}(\hat{z}) = \int \delta\left(\frac{u(\Delta \hat{\mathbf{t}})}{s(\Delta \hat{\mathbf{t}})/\sqrt{n}} - \hat{z}\right) \rho_{\Delta \hat{\mathbf{T}}}(\Delta \hat{\mathbf{t}})\, d\Delta \hat{\mathbf{t}}. \tag{22}$$

Finally, the distribution of the uncertain $p$-value may again be computed according to Eq. 15

$$\rho_{\hat{P}_z}(\hat{p}_z) = \int \delta\left(p_z(\hat{z}) - \hat{p}_z\right) \rho_{\hat{Z}}(\hat{z})\, d\hat{z} \qquad = \int \delta\left(\int\limits_{-\infty}^{\hat{z}} t_{n-1}(z)\, dz - \hat{p}_z\right) \rho_{\hat{Z}}(\hat{z})\, d\hat{z} \tag{23}$$

and then be evaluated according to the two criteria formulated above.

### 3.3.2 Wilcoxon-signed-rank

Compared to the $t$-test, the WSR test (Wilcoxon, 1945) allows to relax the assumption of normality imposed on the generating population $\mathcal{P}_D$, and replaces it by the weaker assumption of symmetry with respect to its mean $\mu$ in order to test the null hypothesis $H_0 : \mu \geq 0$. The test statistic $W$ for this test is defined as

$$W(\mathbf{D}) = \sum_{i=1}^{n} R(|D_i|) \, \Theta(D_i), \tag{24}$$

where $R(|D_i|)$ denotes the rank of $|D_i|$ within the sorted set of the absolute values of differences $\{|D_i|\}$. The Heaviside function $\Theta(D_i)$ guarantees that exclusively $D_i > 0$ are summed. The derivation of the null distribution is a purely combinatoric problem and its explicit form can be be found in lookup tables. Because $W \in \mathbb{N}_{[0,n(n+1)/2]}$ we denote the null distribution by $\mathcal{P}_W^0(w)$ to signal that this is not a continuous density. Explicitly, the null distribution can be derived as follows: First, the assumption of symmetry around zero (for the hypothesis $H_0 : \mu \geq 0$ the relevant null distribution builds on $\mu = 0$) guarantees that the chance for $D_i$ to be positive is equal to $\frac{1}{2}$. Hence, the number of positive outcomes $m$ follows a symmetric binomial distribution $\pi(m) = \binom{n}{m} (\frac{1}{2})^n$. For $m$ positive observations, there are $\binom{n}{m}$ different sets of ranks $\{r_1, ..., r_m\}$ that they may assume, and which are again due to the symmetry of $\mathcal{P}_D$ equally likely. Hence, for a given number of positive outcomes $m$ the probability to obtain a test statistic $w$ is given by the share of those $\binom{n}{m}$ configurations that yield a rank sum equal to $w$. Summing these probabilities over all possible values of $m$ yields the null distribution for the test statistic $w$.

For a given sample $\mathbf{d}$ we test the hypothesis $H_0 : \mu \geq 0$ by computing the corresponding one-sided $p$-value $p_w$, which is given by the cumulative probability that the null distribution assigns to $w'$ values smaller than the observed $w(\mathbf{d})$:

$$p_w(w(\mathbf{d})) = \sum_{i=1}^{n} \mathcal{P}_W^0(w_i') \, \Theta(w(\mathbf{d}) - w_i'). \tag{25}$$

Since $W \in \mathbb{N}_{[0,n(n+1)/2]}$ it follows that $p_w$ assumes only discrete values in $[0,1]$ with the null distribution determining the mapping between these two sets.

The generalization of the WSR-test to the uncertain sample $\mathbf{\Delta \hat{T}}$ can be carried out almost analogously to the $t$-test. However, the fact that $W \in \mathbb{N}_{[0,n(n+1)/2]}$ makes it inconvenient to use a continuous probability density distribution. We denote the distribution for the uncertain $\hat{W}(\mathbf{\Delta \hat{T}})$ by

$$\mathcal{P}_{\hat{W}}(\hat{w}) = \int \delta \left( \sum_{i=1}^{n} R(|\Delta \hat{t}_i|) \, \Theta(\Delta \hat{t}_i) - \hat{w} \right) \rho_{\mathbf{\Delta \hat{T}}}(\mathbf{\Delta \hat{t}}) \, d\mathbf{\Delta \hat{t}}. \tag{26}$$

Given the one-to-one map from all $w \in \mathbb{N}_{[0,n(n+1)/2]}$ to the set of discrete potential values $p_w$ for $P_w$ in $[0,1]$ determined by equation Eq. 25, the probability to obtain $\hat{p}_w$ is already given by the probability to obtain the corresponding $\hat{w}$. Hence, we find

$$\mathcal{P}_{\hat{P}_w}(p_w(\hat{W}) = \hat{p}_w) = \mathcal{P}_{\hat{W}}(\hat{w}). \tag{27}$$

### 3.3.3 Bootstrap test

Given an observed sample of differences $\mathbf{d} = (d_1, ..., d_n)$, a bootstrap test constitutes a third option to test the compatibility of the sample with the hypothesis that the population of differences features a mean equal to or greater than zero: $H_0 := \mu_0 \geq 0$.

Guidance for the construction of a bootstrap hypothesis test can be found in Lehmann and Romano (2006) and Hall and Wilson (1991). The advantage of the bootstrap test lies in its independence from assumptions regarding the distributions' shape. Lehmann and Romano (2006) propose the test statistic

$$v = \sqrt{n}u, \tag{28}$$

with $u(\mathbf{d}) = \frac{1}{n}\sum_{i=1}^{n} d_i$ denoting the sample mean. In contrast to the above two tests, the bootstrap test constructs the null distribution directly from the observed data. In the absence of assumptions, the best available approximation of the population $\mathcal{P}_D$ is given by the empirical density

$$\mathcal{P}_D(d) \sim \frac{1}{n}\sum_{i=1}^{n} \delta(d - d_i). \tag{29}$$

The empirical density does, however, not necessarily comply with the null hypothesis and it thus has to be shifted accordingly:

$$\tilde{\rho}_D(d) = \sum_{i=1}^{n} \delta(d - d_i + u). \tag{30}$$

$\tilde{\rho}_D(d)$ corresponds to the borderline case of the null hypothesis $\mu = 0$. The null distribution for $v$ is then derived by resampling $m$ synthetic samples $\tilde{\mathbf{d}}_j = (\tilde{d}_1, ..., \tilde{d}_n)_j$ of size $n$ from $\tilde{\rho}_D(d)$ and computing $\tilde{v}_j = v(\tilde{\mathbf{d}}_j)$ for each of them. This corresponds to randomly drawing $n$ values from the set $\mathbf{d} - u$ with replacement and computing $v$ for the resampled vectors $m$ times, where the index $j$ labels the iteration of this process. The resulting set $\{\tilde{v}_j\}_j$ induces the data driven null distribution for the test statistic

$$\rho_V^0(v) = \frac{1}{m}\sum_{j=1}^{m} \delta(v - \tilde{v}_j). \tag{31}$$

Setting $m = 10000$ we obtain robust null distributions for the cases $n = 16$, and $n = 20$ relevant for this study. The $p$-value of this bootstrap test is then computed as before in a one-sided manner

$$p_v(v(\mathbf{d})) = \int_{-\infty}^{v(\mathbf{d})} \rho_V^0(v)\,dv = \frac{1}{m}\sum_{j=1}^{m} \Theta\left(v(\mathbf{d}) - \tilde{v}_j\right), \tag{32}$$

where the right hand side equals the fraction of resampled $\tilde{v}_j$ that are smaller than $v(\mathbf{d})$ of the original sample.

In the case where the sample of differences is uncertain, as for $\mathbf{\Delta\hat{T}} = (\Delta\hat{T}_1, ..., \Delta\hat{T}_n)$, the construction scheme for $\rho_V^0$ needs to be adjusted to reflect these uncertainties. In principle, each possible value $\mathbf{\Delta\hat{t}}$ for the uncertain $\mathbf{\Delta\hat{T}}$ is associated with its own null distribution $\rho_V^0(v, \mathbf{\Delta\hat{t}})$. In this sense, the value for the test statistic $v(\mathbf{\Delta t})$ should be compared to the corresponding $\rho_V^0(v, \mathbf{\Delta\hat{t}})$ to derive a $p$-value for this $\mathbf{\Delta\hat{t}}$. Eqs. 31 and 32 define a mapping from $\mathbf{\Delta\hat{t}}$ to its corresponding $p$-value. To compute the uncertainty distribution for the $p$-value, this map has to be evaluated for all potential $\mathbf{\Delta\hat{t}}$, weighted by the uncertainty distribution $\rho_{\mathbf{\Delta\hat{T}}}(\mathbf{\Delta\hat{t}})$:

$$\rho_{\hat{P}_v}(\hat{p}_v) = \int \delta(\hat{p}_v - p_v(\mathbf{\Delta\hat{t}}))\rho_{\mathbf{\Delta\hat{T}}}(\mathbf{\Delta\hat{t}})d\mathbf{\Delta\hat{t}}. \tag{33}$$

The three tests are applied in combination in order to compensate their individual deficits. If the population $\mathcal{P}_{\Delta T}$ was truly Gaussian, the $t$-test would be the most powerful test, i.e., its rejection region would be the largest across all tests on the population mean (Lehmann and Romano, 2006). Since normality of $\mathcal{P}_{\Delta T}$ cannot be guaranteed, the less powerful Wilcoxon-signed-rank test constitutes a meaningful supplement to the $t$-test, relying on the somewhat weaker assumption that $\mathcal{P}_{\Delta T}$ is symmetric around zero. Finally, the bootstrap test is non-parametric and in view of its independence from any assumptions adds a valuable contribution.

### 3.4 Comparison to the 'combined evidence' reported by Erhardt et al. (2019)

For the derivation of the transition lag uncertainty distributions $\rho_{\Delta \hat{T}_i^{p,q}}(\Delta \hat{t}_i^{p,q})$ of the i-th DO event between the proxies $p$ and $q$, we have directly adopted the methodology designed by Erhardt et al. (2019). However, our statistical interpretation of the resulting sets of uncertainty distributions $\{\rho_{\Delta \hat{T}_1}^{p,q}(\Delta \hat{t}_1), \ldots, \rho_{\Delta \hat{T}_n}^{p,q}(\Delta \hat{t}_n)\}$ derived from the set of DO events differs from the one proposed by Erhardt et al. (2019). In this section we explain the subtle yet important differences between the two statistical perspectives.

Given a pair of variables $(p, q)$, Erhardt et al. (2019) define what they call 'combined estimate' $\rho_{\Delta T^*}(\Delta t^*)$ as the product over all corresponding lag uncertainty distributions:

$$\rho_{\Delta T^*}(\Delta t^*) \propto \prod_{i=1}^{n} \rho_{\Delta \hat{T}_i}(\Delta t^*). \tag{34}$$

This implicitly assumes that all DO events share the exact same time lag $\Delta t^*$ between the variables $p$ and $q$. This is realized by inserting a single argument $\Delta t^*$ into the different distributions $\rho_{\Delta \hat{T}_i}(\cdot)$. Hence, the product on the right hand side of Eq. 34 in fact indicates the probability that all DO events assume the time lag $\Delta t^*$, provided that they all assume the same lag:

$$\rho_{\Delta T^*}(\Delta t^*) = \rho_{\Delta T^*}(\Delta t^* | \Delta \hat{t}_1 = \ldots = \Delta \hat{t}_n = \Delta t^*) = \frac{\prod \rho_{\Delta \hat{T}_i}(\Delta t^*)}{\int_{\Omega} \prod \rho_{\Delta \hat{T}_i}(\Delta \hat{t}_i) \, d\Delta \hat{t}_i}, \quad \Omega = \{\boldsymbol{\Delta \hat{t}} : \Delta \hat{t}_i = \Delta \hat{t}_j \, \forall i, j\}. \tag{35}$$

The denominator on the right hand side equals the probability that all DO events share a common time lag. Eq. 34 strongly emphasizes those regions where all uncertainty distributions $\rho_{\Delta \hat{T}_i}(\Delta \hat{t}_i)$ are simultaneously substantially larger than zero. The 'combined evidence' answers the question: Provided that all DO events exhibit the same lag between the transition onsets of $p$ and $q$, then how likely is it that this lag is given by $\Delta t^*$. Drawing on this quantity, (Erhardt et al., 2019) conclude that $\delta^{18}O$ and $Na^+$ 'on average' lag $Ca^{2+}$ and $\lambda$ by about one decade.

Thinking of the DO transition onset lags as i.i.d. random variables of a repeatedly executed random experiment takes into account the natural variability between different DO events and hence, it removes the restricting a priori assumption $\Delta \hat{t}_1 = \ldots = \Delta \hat{t}_n$. In our approach we have related the potentially systematic character of lags to the population mean. Since the sample mean is the best point-estimate of a population mean, we consider it to reasonably indicate potential leads and lags, whose significance should be tested in a second step. Thus, we ascribe the sample mean a similar role as Erhardt et al. (2019) ascribe to the 'combined estimate' and therefore, we present a comparison of these two quantities in Sec. 4.1.

The mean of an uncertain sample $\hat{U} = u(\boldsymbol{\Delta}\hat{\mathbf{T}})$ is again an uncertain quantity and its distribution reads

$$\rho_{\hat{U}}(\hat{u}) = \int \delta(\hat{u} - u(\boldsymbol{\Delta}\hat{\mathbf{t}}))\rho_{\boldsymbol{\Delta}\hat{\mathbf{T}}}(\boldsymbol{\Delta}\hat{\mathbf{t}}) \, d\boldsymbol{\Delta}\hat{\mathbf{t}}. \tag{36}$$

While the 'combined estimate' multiplies the distributions $\rho_{\Delta\hat{T}_i}(\Delta t^*)$, the uncertain sample mean convolutes them pairwise (see Appendix C). We thus expect the distributions for uncertain sample means to be broader than the corresponding distributions for the 'combined estimate'. This can be motivated by considering the simple example of two Gaussian variables $X$ and $Y$. According to the convolution their sample mean $U = \frac{X+Y}{2}$ is normally distributed with variance $\sigma_{x*y}^2 = \frac{\sigma_x^2 + \sigma_y^2}{4}$. In contrast, a combined estimate would yield a normal distribution with variance $\sigma_{xy}^2 = \frac{\sigma_x^2 + \sigma_y^2}{\sigma_x^2 \sigma_y^2}$. Thus, the convolution will appear broader for all $\sigma_x^2 \sigma_y^2 > 4$, which is the case for the distributions considered in this study.

## 4   Results

In the following we apply the above methodology to the different pairs of proxies that Erhardt et al. (2019) found to exhibit a decadal-scale time lag, based on an assessment of the 'combined estimate'; namely $(Ca^{2+}, Na^+)$, $(\lambda, Na^+)$, $(Ca^{2+}, \delta^{18}O)$ and $(\lambda, \delta^{18}O)$ from the NGRIP ice core, and $(Ca^{2+}, Na^+)$ from the NEEM ice core. For each individual proxies we estimate the uncertain transition onsets relative to the timing of the DO events as given by Rasmussen et al. (2014) (see Fig. 2). From these uncertain transition onsets, the uncertainty distributions for the sets of uncertain lags $\boldsymbol{\Delta}\hat{\mathbf{T}}^{p,q}$ between the proxies $p$ and $q$ are derived according to Eq. 7. As mentioned previously, we study the same selection of transitions evidenced in the multi-proxy records as Erhardt et al. (2019). This selection yields sample sizes of either 16 or 20 lags per pair of proxies, but not 23, which is the total number of DO events present in the Data.

We first study the uncertain sample means. As already mentioned, the sample mean is the best available point estimate for the population mean. Hence, sample means different from zero may be regarded as first indications for potential systematic lead-lag relationships and thus motivate the application of hypothesis tests. We compare the results obtained for the uncertain sample means with corresponding results for the 'combined estimate'. Both quantities indicate a tendency towards a delayed transition in $Na^+$ and $\delta^{18}O$. Accordingly, in the subsequent section we apply the generalized hypothesis tests introduced above to the uncertain samples of transition lags to test the null hypothesis that pairwise the apparent transition sequence is not systematically favoured, that is, that the populations have mean equal or greater than zero.

### 4.1   Uncertain sample mean and combined estimate

Based on their assessment of the 'combined estimate', Erhardt et al. (2019) concluded that on average, transitions in $Ca^{2+}$ and $\lambda$ started approximately one decade earlier than their counterparts in $Na^+$ and $\delta^{18}O$. Fig. 5 shows a reproduction of their results together with the uncertainty distributions of the sample means for all proxy pairs under study (($Ca^{2+}, \delta^{18}O$) and ($\lambda, \delta^{18}O$) are not shown in Erhardt et al. (2019)). For an uncertain sample of lags $\boldsymbol{\Delta}\hat{\mathbf{T}}^{p,q}$ between the proxies $p$ and $q$, the 'combined estimate' and the uncertain sample mean are computed according to Eq. 35 and Eq. 36, respectively. The reproduction of the 'combined estimate' deviates from the original publication by no more than 1 year with respect to the mean and the 5th and

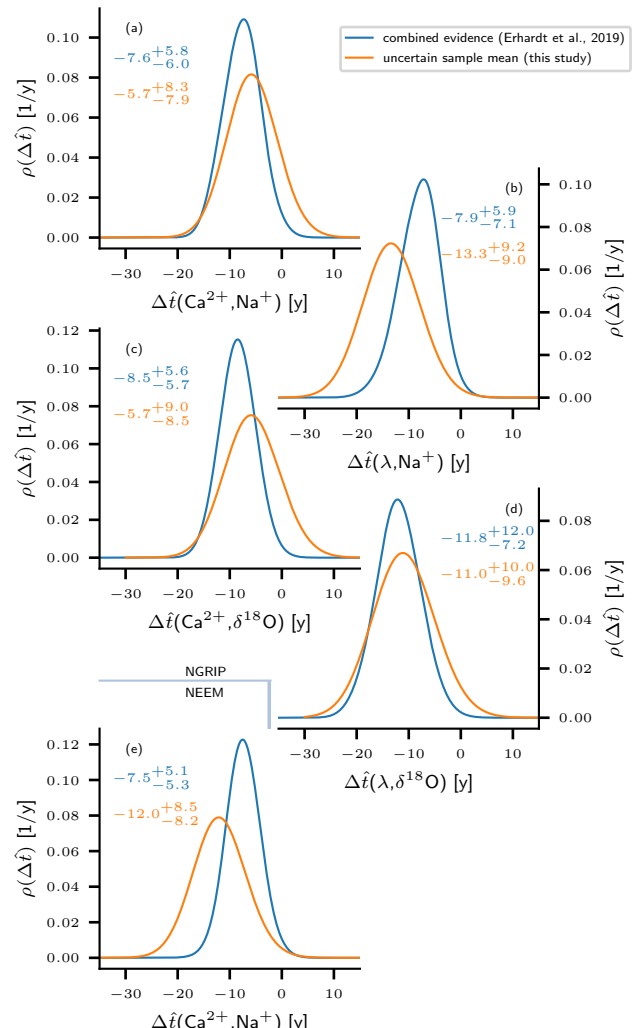

**Figure 5.** Comparison between the uncertain sample means (this study) and 'combined estimates' according to Erhardt et al. (2019). The probability densities for the 'combined estimate' are derived from the samples of uncertain time lags according to Eq. 34. Correspondingly, the uncertain sample means are computed according to Eq. 36. The numbers in the plots indicate the mean, the 5th and the 95th percentile of the respective quantity. Both computations use Gaussian kernel density estimates of the MCMC-sampled transition onsets lags. Panels (a-d) refer to proxy pairs from the NGRIP ice core and panel (e) shows results from the NEEM ice core. The distributions for both the combined estimate and the uncertain sample mean point towards a delayed transition onset in $\delta^{18}O$ and $Na^+$ with respect to $\lambda$ and $Ca^{2+}$.

95th percentiles across all pairs. These deviations might originate from the stochastic MCMC-sampling process used for the analysis.

With the sample mean being the best point estimator of the population mean, it serves as a suitable indicator for a potential population mean different from zero. The expectations

$$\mathrm{E}(\hat{U}) = \int \hat{u} \, \rho_{\hat{U}}(\hat{u}) \, d\hat{u} \tag{37}$$

for the sample means of all proxy pairs do in fact suggest a tendency towards negative values in all distributions, i.e., a delay of the $Na^+$ and $\delta^{18}O$ transition onsets with respect to $Ca^{2+}$ and $\lambda$. This indication is weakest for $(Ca^2, Na^+)$ and $(Ca^{2+}, \delta^{18}O)$

from NGRIP, since for these pairs we find non-zero probability for a positive sample mean. For the other pairs the indication is comparably strong, with the 95th percentiles of the uncertainty distributions for the sample mean still being less than zero. Overall, the results for the uncertain mean confirm the previously reported tendencies and in very rough terms, the distributions qualitatively agree with those for the 'combined estimate'. In agreement with the heuristic example from Sec. 3.4, we find the sample mean distributions to be broader than the 'combined estimate' distributions in all cases. The expected sample means

indicate less pronounced lags for $(Ca^2, Na^+)$ (panel (a)) and $(Ca^{2+}, \delta^{18}O)$ (panel (c)) from the NGRIP ice core compared to the expectations of the corresponding 'combined estimate'. In combination with the broadening of the distribution, this yields considerable probabilities for $U > 0$ of 12% and 14%, respectively, indicating a delayed transition of $Ca^{2+}$ in the sample mean with respect to $Na^+$ or $\delta^{18}O$. Contrarily, for $(\lambda, Na^+)$ (NGRIP, panel (b)) and $(Ca^{2+}, Na^+)$ (NEEM, panel(e)) the expected sample means point towards more distinct lags than reported by Erhardt et al. (2019) based on the 'combined estimate'. For

$(\lambda, \delta^{18}O)$ (NGRIP, panel (d)) the sample mean and the 'combined estimate' are very close. Note that the analysis of the uncertain sample values yields a more inconsistent picture with regard to the $(Ca^{2+}, Na^+)$ lag in the two different cores. While the distribution is shifted to less negative (less pronounced lag) for the NGRIP data, it tends to more negative values in the case of NEEM (stronger lag), suggesting a slight discrepancy between the cores.

    Both quantities, the uncertain sample mean and the 'combined estimate' point towards delayed transition onsets in $Na^+$

and $\delta^{18}O$ with respect to $Ca^{2+}$ and $\lambda$, with major fractions of their uncertainty densities being allocated to negative values. This motivates to test whether the observations significantly contradict the hypothesis of a population mean equal or greater than zero. Accordingly, the subsequent section presents the results obtained from the application of three different hypothesis test that target the population mean. As discussed in Sec. 3, the tests have been modified to allow for a rigorous uncertainty propagation and return uncertainty distribution for their corresponding $p$-values, rather than scalars.

**4.2  Statistical significance of the proposed lead-lag relations**

Above, we identified three tests for testing the hypothesis that the samples $\boldsymbol{\Delta\hat{T}}^{p,q}$ were actually generated from populations that on average feature no or even reversed time lags compared to what the sign of the corresponding uncertain sample mean suggests. Mathematically, this is equivalent to testing the hypothesis that the mean $\mu^{p,q}$ of the population $\mathcal{P}_{\Delta T}^{p,q}$ is greater or equal to zero: $H_0 : \mu^{p,q} \geq 0$. A rejection of this hypothesis would confirm that the assessed sample is very unlikely to stem

from a population with $\mu^{p,q} \geq 0$, and would thereby provide evidence for a systematic lag. Under the constraints indicated

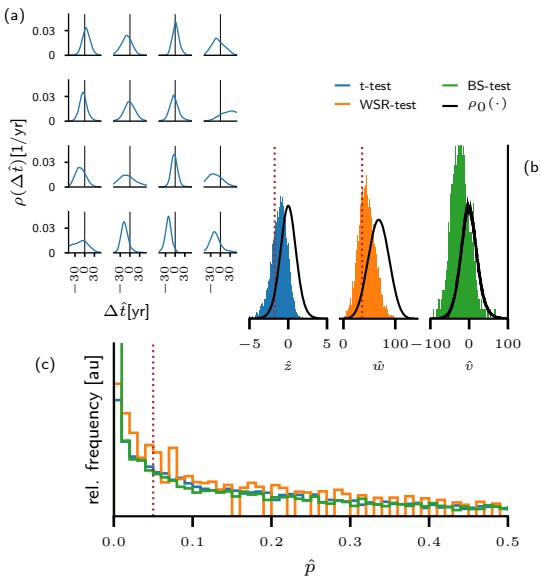

**Figure 6.** Exemplary application of the analysis to the proxy pair $(Ca^{2+}, Na^+)$ from the NGRIP ice core. Panel (a) shows 16 uncertain time lags $\Delta \hat{T}_i$ derived from the proxy data around DO events. The continuous densities have been obtained via a Gaussian kernel density estimate from the corresponding MCMC samples (see Sec. 3.1). In panel (b) the uncertain test statistics induced by the uncertain sample are shown for the $t$-test (blue), the WSR-test (orange) and a bootstrap test (green). The values that comprise the histograms are immediately derived from the MCMC samples. Panel (c) shows the empirical uncertainty distribution for the $p$-values of the three tests, following from the uncertain test statistics in panel (b). Dotted red lines seperate rejection from acceptance regions in panels (b) and (c). In case of the bootstrap test, the rejection regions cannot be defined consistently on the level of the test statistic, since each possible value $\mathbf{\Delta\hat{t}}$ for the uncertain $\mathbf{\Delta\hat{T}}$ induces its individual null distribution. The null distribution shown here is in fact the pooled distribution of resampled $\tilde{v}_j$ obtained from all MCMC-sampled values for $\mathbf{\Delta\hat{T}}$. For the other proxy pairs investigated in this study, corresponding plots would appear structurally similar.

above this would in turn yield evidence for an actual lead of the corresponding climatic process. We have chosen a significance level of $\alpha = 0.05$, which is a typical choice. Fig. 7 summarizes the final uncertainty distributions of the three tests for all proxy pairs under study. Corresponding values are given in Tab. 1.

Fig. 6 exemplarily illustrates the application of the three tests to the empirical densities obtained for $\mathbf{\Delta\hat{T}}(Ca^{2+}, Na^+)$

(NGIRP). In Fig. 6 the initial uncertainty in the observations – i.e., the uncertainty encoded by the distributions of transition onset lags – is propagated to an uncertain test statistic according to Eq. 16. In turn, the uncertain test statistic yields an uncertain $p$-value (see Eq. 17). Since the numerical computation is based on empirical densities as generated by the MCMC sampling, we show the corresponding histograms instead of continuous densities - for the $\rho_{\Delta \hat{T}_i}(\Delta \hat{t}_i)$ Gaussian kernel density estimates are presented only for the sake of visual clarity. On the level of the test statistics the red dashed line separates the acceptance

from the rejection region, based on the null distributions given in black. Qualitatively, the three tests yield the same results. The

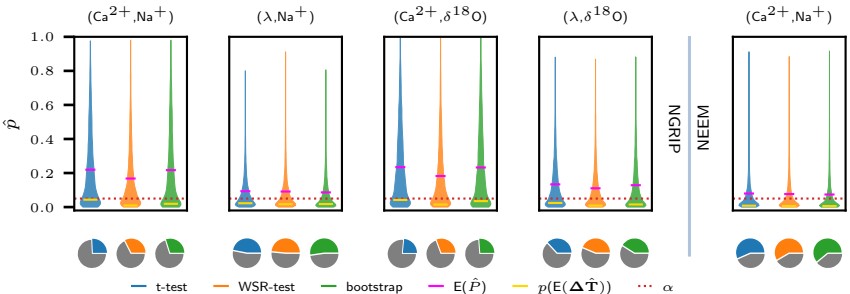

**Figure 7.** Results of the hypothesis tests applied to the uncertain samples of transition onset lags $\mathbf{\Delta\hat{T}}_i^{p,q}$. The violin plots show the Gaussian kernel density estimates of the empirical uncertainty distributions for $p$-values (see Fig. 6) obtained for all tests and for all proxy pairs investigated. Pink bars indicate the corresponding expected $p$-values $\mathrm{E}(\hat{P})$ and yellow bars indicate the $p$-values obtained from testing the expected samples $\mathrm{E}(\mathbf{\Delta\hat{T}})$. All expected $p$-values are above the significance level $\alpha = 0.05$ (red dotted line), while the expected samples apper to be significant consistently across all proxy pairs and all tests. The pie charts indicate the probability for the respective $p$-values to be less then $\alpha$.

histograms clearly indicate non-zero probabilities for the test statistic in both regions. Correspondingly, the histograms for the $p$-values stretch well across the significance threshold. The shapes of the histograms resemble an exponential decay towards higher $p$-values. This results from the non-linear mapping of the test statistics to the $p$-values. Despite the pronounced bulk of empirical $p$-values below the significance level, the probability for non-significant $p$-values is still well above 50% for the three tests (see Tab. 1). Also, the expected $p$-value exceeds the significance level for all tests. Hence, neither of the two criteria for rejecting the null hypothesis formulated in Sec. 3.2 is met for the proxy pair $(\mathrm{Ca}^{2+}, \mathrm{Na}^+)$. In contrast, if the observational uncertainties are averaged out on the level of the transition onset lags, all tests yield $p$-values below the significance level, which would indicate that the lags were indeed significant. Hence, the rigorous propagation of uncertainties qualitatively changes the statistical assessment of the uncertain sample of lags $\mathbf{\Delta\hat{T}}(\mathrm{Ca}^{2+}, \mathrm{Na}^+)$ (NGRIP). While the expected sample rejects the null hypothesis, rigorous uncertainty propagation leads to acceptance. This holds true for all tests.

Fig. 7 summarizes the results obtained for all proxy pairs under study. Qualitatively, our findings are the same for the other pairs as for the $(\mathrm{Ca}^{2+}, \mathrm{Na}^+)$ (NGRIP) case discussed in detailed above. All expected $p$-values, as indicated by the pink bars, are above the significance level. Also, the probability for significance is below 60% for all pairs and all tests as shown by the pie charts. For all proxy pairs and for all tests, the formulated decision criteria do thus not allow to reject the null hypothesis of a population mean greater or equal to zero. In contrast, all expected samples are significant across all tests with corresponding $p$-values indicated by the yellow bars. The proxy pairs with the lowest expected $p$-values and the highest probability for $\hat{P} < \alpha$ are $(\lambda, \mathrm{Na}^+)$ from NGRIP and $(\mathrm{Ca}^{2+}, \mathrm{Na}^+)$ from NEEM, as already suggested by the analysis of the uncertain sample mean. For the NGRIP ice core the delay of $\mathrm{Na}^+$ and $\delta^{18}\mathrm{O}$ with respect to $\mathrm{Ca}^{2+}$ has a very low probability to be significant (approximately one third). The pair $(\lambda, \delta^{18}\mathrm{O})$ ranges in between the latter two.

**Table 1.** Results from the application of the $t$-test, the WSR test and a bootstrap test to uncertain samples of DO transition onset lags $\mathbf{\Delta\hat{T}}^{p,q}$. $E(\hat{P})$ denotes the expected $p$-value, derived from the uncertainty-propagated $p$-value distribution. The probability for a significant test results associated with the same distribution is indicated by $\pi(\hat{P} < 0.05)$. For comparison, the $p$-values from the application of the tests to the expected sample $E(\mathbf{\Delta\hat{T}}) = \int \rho_{\mathbf{\Delta\hat{T}}}(\mathbf{\Delta\hat{t}})\mathbf{\Delta\hat{t}}\,d\mathbf{\Delta\hat{t}}$ are given in the bottom row.

| | NGRIP | | | | | | | | | | | | NEEM | | |
|---|---|---|---|---|---|---|---|---|---|---|---|---|---|---|---|
| | $(Ca^{2+}, Na^+)$ | | | $(\lambda, Na^+)$ | | | $(Ca^{2+}, \delta^{18}O)$ | | | $(\lambda, \delta^{18}O)$ | | | $(Ca^{2+}, Na^+)$ | | |
| | $t$-test | WSR | BS | $t$-test | WSR | BS | $t$-test | WSR | BS | $t$-test | WSR | BS | $t$-test | WSR | BS |
| $E(\hat{P})$ | 0.22 | 0.17 | 0.22 | 0.09 | 0.09 | 0.09 | 0.23 | 0.18 | 0.23 | 0.13 | 0.11 | 0.13 | 0.08 | 0.08 | 0.07 |
| $\pi(\hat{P} < 0.05)$ | 0.26 | 0.32 | 0.3 | 0.47 | 0.48 | 0.52 | 0.24 | 0.31 | 0.26 | 0.37 | 0.44 | 0.41 | 0.57 | 0.58 | 0.61 |
| $p(E(\mathbf{\Delta\hat{T}}))$ | 0.04 | 0.01 | 0.02 | 0.02 | 0.02 | 0.02 | 0.04 | 0.01 | 0.04 | 0.02 | 0.01 | 0.02 | 0.01 | 0.01 | 0.01 |

## 5 Discussion

Erhardt et al. (2019) have reported an average time lag between the transition onsets in $Na^+$ and $\delta^{18}O$ proxy values and their counterparts in $Ca^{2+}$ and $\lambda$ at the onset of DO events. This statement is based on the assessment of the 'combined estimate' derived from uncertain samples of time lags $\mathbf{\Delta\hat{T}}^{p,q}$. The samples were obtained by applying a well-suited Bayesian transition onset detection scheme to high resolution time series of the different proxies. The 'combined estimate' indicates leads of the $Ca^{2+}$ and $\lambda$ transition onsets with respect to $Na^+$ and $\delta^{18}O$ by approximately one decade, with the 90% confidence interval ranging from 0 to approximately 15 years. The 'combined estimate' implicitly assumes that for a given proxy pair all DO events share a common time lag ($\Delta\hat{T}_i^{p,q} = \Delta\hat{T}_j^{p,q}$).

We argue that the variability across different DO events cannot be ignored in the assessment of the data. Although the DO events are likely to be caused by the same physical mechanism, changing boundary conditions and other natural climate fluctuations will lead to deviations in the exact timings of the different processes involved in triggering the individual DO events. Fig. 2 clearly shows that the different events exhibit different time lags. Provided that the DO events were driven by the same process, physically they constitute different realizations and they exhibit great variability also in other variables such as the amplitude of the temperature change (Kindler et al., 2014) or the waiting times with respect to the previous event (Ditlevsen et al., 2007; Boers et al., 2018). The random experiment framework introduced in this study allows to relax the constraint of a common time lag $\Delta t^*$ shared across all events, and reflects the fact that natural variability will cause different expressions of the same mechanism across different DO events. Moreover, this framework relates potential systematic leads and lags in the physical process that drives DO events to a corresponding non-zero mean of a population of lags between proxy variables. This allows for the physically meaningful formulation of a statistical hypothesis and a corresponding null hypothesis. By applying different hypothesis tests we have followed a well-established line of statistical inference. Motivated by the apparent transition onset delays in $Na^+$ and $\delta^{18}O$ with respect to the transitions in $\lambda$ and $Ca^{2+}$, as reported by Erhardt et al. (2019) and confirmed here on the level of uncertain sample means, we tested the null hypothesis that the corresponding populations do not favor the

proposed transition sequence. Rejection of this hypothesis would have provided evidence that the observed lag tendency is an imprint of the underlying physical process and therefore a systematic feature.

Generalized versions of three different hypothesis tests consistently fail to reject the null hypothesis under rigorous propagation of the observational uncertainties originating from the MCMC-based transition onset detection. This holds true for all proxy pairs. The fact that the tests rely on different assumptions on the population's shape, but nonetheless qualitatively yield the same results, makes our assessment robust. We conclude that the possibility that the observed tendencies towards advanced transitions in $Ca^{2+}$ and $\lambda$ have occurred simply by chance cannot be ruled out. If the common physical interpretation of the studied proxies holds true, our results imply that the hypothesis that the trigger of the DO events is associated directly with the North Atlantic sea-ice cover rather than the atmospheric circulation - be it on synoptic or hemispheric scale - cannot be ruled out. We emphasize that our results should not be misunderstood as evidence against the alternative hypothesis of a systematic lag. In the presence of a systematic lag ($\mu < 0$) the ability of hypothesis tests to reject the null hypothesis of no systematic lag (($H_0 : \mu = 0$)) depends on the sample size $n$, the ratio between the mean lag $|\mu|$, the variance of the population, and on the precision of the measurement. Neither of these quantities is favourable in our case and thus, it is certainly possible that the null hypothesis cannot be rejected despite the alternative being true.

Our main purpose was the consistent treatment of observational uncertainties and we have largely ignored the vibrant debate on the qualitative interpretation of the proxies. Surprisingly, we could not find any literature on the application of hypothesis tests to uncertain samples of the kind discussed here. The theory of fuzzy $p$-values is in fact concerned with uncertainties either in the data or in the hypothesis. It is, however, not applicable to measurement uncertainties that are quantifiable in terms of probability density functions (Filzmoser and Viertl, 2004). We have proposed to propagate the uncertainties to the level of the $p$-values and to then consider the expected $p$-values and the share of $p$-values which indicate significance, in order to decide between rejection and acceptance. The $p$-value measures the extremeness of a sample with respect to the null distribution and we hence regard the expected $p$-value to be a suitable measure for the uncertain samples' extremeness.

The probability of the uncertain sample to be significant at a given level is also a reasonable indicator, which can be invoked in addition. In cases of high cost of a wrongly rejected null hypothesis, one might want to have a high degree of certainty that the uncertain sample actually contradicts the null hypothesis and hence a high probability for the uncertain $p$-value to be smaller than $\alpha$. In contrast, if the observational uncertainties are averaged out beforehand, crucial information is lost. The expected sample may either be significant or not, but the uncertainty about the significance can no longer be accurately quantified.

The potential of the availability of data from different sites has probably not been fully leveraged in this study. Naively, one could think of the NEEM and NGRIP ($Ca^{2+}$, $Na^+$) lag records as two independent observations of the same entity. However, the discrepancy in the corresponding sample mean uncertainty distributions opens up the question how changes in the climatic features such as sea-ice cover and atmospheric circulation are actually recorded by the proxies at different sites, and how important regional geographic differences are. Proxy-enabled modeling studies as presented by Sime et al. (2019) could shed further light on the question to what extent the NEEM and NGRIP sites record the same signal after an abrupt change of the climatic conditions. Also, a comparison of the NGRIP and NEEM records on an individual event level could provide

further insights how to combine these records statistically. There might be ways to further exploit the advantage of having two recordings of the same signal.

## 6 Conclusions

We have presented a statistical reinterpretation of the high-resolution proxy records provided and analyzed by Erhardt et al. (2019). The probabilistic transition onset detection also designed by Erhardt et al. (2019) very conveniently quantifies the uncertainty in the transition onset estimation by returning probability densities instead of scalar estimates. While the statistical quantities 'combined estimate' (Erhardt et al., 2019) and 'uncertain sample mean' (this study) indicate a tendency for a pairwise delayed transition onset in $Na^+$ and $\delta^{18}O$ proxy values with respect to $Ca^{2+}$ and $\lambda$, a more rigorous treatment of the involved uncertainties shows that these tendencies are not statistically significant. That is, at the significance level $\alpha = 5\%$ they do not contradict the null-hypothesis that no or the reversed transition sequence is in fact physically favoured. Thus, a pairwise systematic lead-lag relation cannot be evidenced for any of the proxies studied here. We have shown that if uncertainties on the level of transition onset lags are averaged out beforehand, the samples of lags indeed appear to be significant, which underpins the importance of rigorous uncertainty propagation in the analysis of paleoclimate proxy data. We have focused on the quantitative uncertainties and have largely ignored qualitative uncertainty stemming from the climatic interpretation of the proxies. However, if the common proxy interpretations hold true, our findings suggest that, for example, the hypothesis of an atmospheric trigger - either of hemispheric or synoptic scale - for the DO events should not be favoured over the hypothesis that a change in the North Atlantic sea-ice cover initiates the DO events.

Even though we find that the uncertainty of the transition onset detection combined with the small sample size prevents the deduction of statistically unambiguous statements on the temporal order of events, we think that multi-proxy analysis is a promising approach to investigate the sequential order at the beginning of DO events. In this study, we refrained from analyzing the lags between the different proxies in a combined approach and focused on the marginal populations. However, a combined statistical evaluation - that is, treating the transition onsets of all proxy variables as a four-dimensional random variable - merits further investigation. Also, we propose to statistically combine measurements from NEEM and NGRIP (and potentially further ice cores) of the same proxy pairs. Finally, hierarchical models may be invoked to avoid switching from a Bayesian perspective in the transition onset estimation to a frequentist perspective in the statistical interpretation of the uncertain samples.

Furthermore, the interpretation of proxy variables requires further refinement. Especially the interpretation of $Na^+$ as a sea ice proxy remains debated. Potentially, other proxies could be included in subsequent analyses.

Finally, effort in conducting modelling studies should be sustained. Especially proxy-enabled modeling bears the potential to improve comparability between model results and paleoclimate records. Together, these lines of research are promising to further constrain the sequence of events that have caused the abrupt climate changes assocaited with the DO events during the last glacial.

*Code and data availability.* 10-year resolution time series of Na$^+$ and Ca$^{2+}$ concentrations and $\delta^{18}$O values from the NGRIP ice core shown in Fig. 1 are retrieved from PANGAEA (Erhardt et al., 2018, https://doi.org/10.1594/PANGAEA.896743). The high-resolution Na$^+$

and Ca$^{2+}$ concentration time series centered around DO transitions which were used to derive the time lags between the transition onsets of the two proxies can be found in the same PANGAEA archive. The code used to generate the empirical densities of transition onsets is available at https://github.com/terhardt/DO-progression (last access: 21 October 2020). The code used to carry out the statistical analysis of the sample of empirical transition onset distributions is available from the authors upon request and will be published once the manuscript is accepted.

*Acknowledgements.* We thank Norbert Marwan for very helpful comments and discussions. This is TiPES contribution 60; the TiPES (Tip-

630 ping Points in the Earth System) project has received funding from the European Union's Horizon 2020 research and innovation program under grant agreement 178 No. 820970. NB acknowledges funding by the Volkswagen foundation.

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

## Appendix A: Numerical treatment of high dimensional probability densities

In Sec.3.1 we introduced the probabilistic transition onset detection designed by Erhardt et al. (2019). Given a single time series the formulation of a stochastic ramp model induces a posterior probability density for the set of model parameters $\Theta$ in a Bayesian sense

$$\pi(\Theta|\mathcal{D}) = \frac{\pi(\mathcal{D}|\Theta)\,\pi(\Theta)}{\pi(\mathcal{D})}. \tag{A1}$$

However, a classical numerical representation of this density on a discretized grid is inconvenient. Due to its high dimension-
780 ality for a reasonable grid spacing the number of data points easily overloads the computational power of ordinary computers. E.g. representing each dimension with a minimum of 100 points would amount to a total of $10^{12}$ data points. On top of that, the application of any methods to such a grid is computationally very costly. Here, the MCMC sampler constitutes an efficient solution. By sampling a representative set $\{\theta_j\}_j$ from the posterior probability density it may be used to construct an empirical density in the sense of Eq. 4. For sake of simplicity in the main text we have formulated the methods in terms of continuous
probability densities, although all computations in fact rely on empirical densities obtained from MCMC samples. Here, we show that all steps in the derivation of the methods can be performed equivalently under stringent use of the empirical density. With regards to hypothesis tests, the use of empirical densities for the uncertain transition lag samples $\mathbf{\Delta T}_i^{p,q}$ essentially boils down to an application of the tests to each individual value comprised in the respective empirical density.

For a given proxy and a given DO event, in a first step the MCMC algorithm samples from the joint posterior probability
density for the models parameter configuration $\theta = (t_0, \tau, y_0, \Delta y, \alpha, \sigma)$, giving rise to the empirical density $\bar{\rho}_\Theta(\theta) = \frac{1}{m}\sum\delta(\theta - \theta_j)$. Integration over the nuisance parameters then yields the marginal empirical density for the transition onset

$$\bar{\rho}_{T_0}^{p,i}(t_0^{p,i}) = \frac{1}{m}\sum_{j=1}^{m}\delta(t_0^{p,i} - t_{0,j}^{p,i}), \tag{A2}$$

where the index $i$ indicates the DO event and $p$ denotes the proxy variable while $j$ runs over the MCMC sampled values. We use bars to mark empirical densities in contrast to continuous densities. The uncertainty distribution for the lag $\Delta T_i^{p,q}$ between
795 the variables $p$ and $q$ as defined by Eq. 7 may then be approximated as follows (ommitting the index $i$):

$$\begin{aligned}
\rho_{\Delta T}^{p,q}(\Delta t^{p,q}) &= \int\int \delta(t_0^p - t_0^q - \Delta t^{p,q})\rho_{T_0}^p(t_0^p)\,\rho_{T_0}^q(t_0^q)\,dt_0^p\,dt_0^q \\
&\simeq \int\int \delta(t_0^p - t_0^q - \Delta t^{p,q})\bar{\rho}_{T_0}^p(t_0^p)\,\bar{\rho}_{T_0}^q(t_0^q)\,dt_0^p\,dt_0^q = \int\int \delta(t_0^p - t_0^q - \Delta t^{p,q}) \\
&\quad \frac{1}{m}\sum_{j=1}^{m}\delta(t_0^p - t_{0,j}^p)\,\frac{1}{m}\sum_{k=1}^{m}\delta(t_0^p - t_{0,k}^q)\,dt_0^p\,dt_0^q = \frac{1}{m^2}\sum_{j,k=1}^{m}\delta(t_{0,j}^p - t_{0,k}^q - \Delta t^{p,q}) = \bar{\rho}_{\Delta T}^{p,q}(\Delta t^{p,q}). \tag{A3}
\end{aligned}$$

Thus, the empirical uncertainty distribution for the time lag is induced by the set of all possible differences between members of the two MCMC samples for the respective transition onsets

$$\{\Delta t_j^{p,q}\}_{j\in[1,m^2]} = \{t_{0,k}^p - t_{0,l}^q\}_{p,q\in[1,m]}. \tag{A4}$$

For this study $m = 6000$ values have been sampled with the MCMC algorithm for each transition under study. This yields $m^2 = 36 \cdot 10^6$ potential values for the empirical $\Delta T$ uncertainty distribution. To keep the computation efficient, the sets of lags were restricted to combinations $k = l$ and thus to 6000 empirical values. We thus approximate

$$\bar{\rho}_{\Delta T}^{p,q}(\Delta t^{p,q}) \simeq \frac{1}{m} \sum_{j=1}^{m} \delta(t_{0,j}^{p} - t_{0,j}^{q} - \Delta t^{p,q}). \tag{A5}$$

This drastic reduction of values certainly requires justification, which we give later by comparing final results of the analysis to those obtained from control runs. The control runs analogously construct the empirical densities for the transition onset lags from 6000 out of the $36 \cdot 10^6$ possible values, but use randomly shuffled versions of the original sets of transition onset times for the variables $p$ and $q$:

$$\bar{\rho}_{\Delta T}^{p,q,ctrl}(\Delta t^{p,q}) \simeq \frac{1}{m} \sum_{j=1}^{m} \delta(t_{0,s(j)}^{p} - t_{0,s'(j)}^{q} - \Delta t^{p,q}). \tag{A6}$$

Here, $s$ and $s'$ denote randomly chosen permutations of the set $\{1, 2, ...., m\}$.

As in the main text, in the following we denote uncertain quantities with a hat. For a given proxy pair the starting point for the statistical analysis however, is the uncertain sample $\mathbf{\Delta \hat{T}}^{p,q} = (\Delta \hat{T}_1^{p,q}, ..., \Delta \hat{T}_n^{p,q})$, which is characterized by the $n$ dimensional uncertainty distribution $\rho_{\mathbf{\Delta \hat{T}}^{p,q}}(\mathbf{\Delta \hat{t}}^{p,q}) = \prod \rho_{\Delta \hat{T}_i^{p,q}}(\Delta \hat{t}_i^{p,q})$. Its empirical counterpart is given by

$$\bar{\rho}_{\mathbf{\Delta \hat{T}}^{p,q}}(\mathbf{\Delta \hat{t}}^{p,q}) = \prod_{i=1}^{n} \bar{\rho}_{\Delta \hat{T}_i^{p,q}}(\Delta \hat{t}_i^{p,q}) = \frac{1}{m^n} \prod_{i=1}^{n} \sum_{j=1}^{m} \delta(\Delta \hat{t}_i^{p,q} - \Delta t_{i,j}^{p,q}). \tag{A7}$$

This empirical density is comprised of $m^n$ possible values for the n dimensional random vector $\mathbf{\Delta \hat{T}}^{p,q}$ and again, a substantial reduction of the representing set is required for practical computation. Defining the reduced empirical density for $\mathbf{\Delta \hat{T}}^{p,q}$ as

$$\tilde{\rho}_{\mathbf{\Delta \hat{T}}^{p,q}}(\mathbf{\Delta \hat{t}}^{p,q}) = \frac{1}{m} \sum_{j=1}^{m} \prod_{i=1}^{n} \delta(\Delta \hat{t}_i^{p,q} - \Delta t_{i,j}^{p,q}) = \frac{1}{m} \sum_{j=1}^{m} \delta(\mathbf{\Delta \hat{t}}^{p,q} - \mathbf{\Delta t}_j^{p,q}) \tag{A8}$$

constrains the set that determines $\tilde{\rho}_{\mathbf{\Delta \hat{T}}^{p,q}}(\mathbf{\Delta \hat{t}}^{p,q})$ to $m$ values, where those values from different DO events with the same MCMC index $j$ are combined:

$$\mathbf{\Delta t}_j^{p,q} = (\Delta t_{1,j}^{p,q}, ..., \Delta t_{n,j}^{p,q}). \tag{A9}$$

Again, the validity is checked by randomly permuting the sets $\{\Delta t_{i,j}^{p,q}\}$ for the individual DO events with respect to the index $j$ before the set reduction in the control runs.

Having found a numerically manageable expression for the empirical uncertainty distribution of the sample $\mathbf{\Delta \hat{T}}^{p,q}$ it remains to be shown how the hypothesis tests can be formulated on this basis. If $\{\mathbf{\Delta t}_j\}_j$ denotes the set of $n$ dimensional vectors forming the empirical uncertainty distribution for the sample of lags obtained from $n$ DO events, then the naive intuition holds true and the corresponding set $\{\phi_j = \phi(\mathbf{\Delta t}_j)\}_j$ represents the empirical uncertainty distribution of the test statistic and correspondingly $\{p_\phi(\phi_j)\}_j$ characterizes the uncertain $p$-value. In the following, we examplarily derive this relation for the $t$-test - the derivations for the WSR and the bootstrap test are analogue.

Recall the statistic of the $t$-test

$$z(\mathbf{d}) = \frac{u(\mathbf{d}) - \mu}{s(\mathbf{d})/\sqrt{(n)}}.$$ (A10)

The empirical uncertainty distribution for a sample $\mathbf{\Delta\hat{T}}$ induces a joint uncertainty distribution for the samples mean and standard deviation

$$\bar{\rho}_{\hat{U},\hat{S}}(\hat{u},\hat{s}) = \int \delta\left(u - \frac{1}{n}\sum_{i=1}^{n}\Delta\hat{t}_i\right)\delta\left(s - \frac{1}{n-1}\sum_{i=1}^{n}(u-\Delta\hat{t}_i)^2\right)\frac{1}{m}\sum_{j=1}^{m}\prod_{i=1}^{n}\delta(\Delta\hat{t}_i - \Delta t_{i,j})\,d\Delta\hat{t}_1\,...\,d\Delta\hat{t}_n$$

$$= \frac{1}{m}\sum_{j=1}^{m}\delta\left(\hat{u} - \frac{1}{n}\sum_{i=1}^{n}\Delta t_{i,j}\right)\delta\left(\hat{s} - \frac{1}{n-1}\sum_{i=1}^{n}(\hat{u}-\Delta t_{i,j})^2\right).$$ (A11)

Let $u_j = \frac{1}{n}\sum_{i=1}^{n}\Delta t_{i,j}$, and $s_j = \frac{1}{n-1}\sum_{i=1}^{n}(u_j-\Delta t_{i,j})^2$. Then, the empirical uncertainty distribution for $(\hat{U},\hat{S})$ can be written as

$$\bar{\rho}_{\hat{U},\hat{S}}(\hat{u},\hat{s}) = \frac{1}{m}\sum_{j=1}^{m}\delta(\hat{u} - u_j)\,\delta(\hat{s} - s_j)$$ (A12)

The $(u_j, s_j)$ that form the empirical uncertainty distribution are simply the mean and standard deviation of those $\mathbf{\Delta t}_j = (\Delta t_{1,j}, \Delta t_{2,j}, ..., \Delta t_{n,j})$ that form the vector valued empirical uncertainty distribution for $\mathbf{\Delta\hat{T}}$. From $\bar{\rho}_{\hat{U},\hat{S}}(\hat{u},\hat{s})$, the empirical uncertainty distribution for the uncertain test statistic $\hat{Z}$ can be computed as follows:

$$\bar{\rho}_{\hat{Z}}(\hat{z}) = \int \delta\left(\hat{z} - \frac{\hat{u}-\mu}{\hat{s}/\sqrt{(n)}}\right)\rho_{\hat{U},\hat{S}}(\hat{u},\hat{s})\,d\hat{u}\,d\hat{s} = \frac{1}{m}\sum_{j=1}^{m}\delta\left(\hat{z} - \underbrace{\frac{u_j-\mu}{s_j/\sqrt{(n)}}}_{=z_j}\right).$$ (A13)

This shows, that for a given empirical uncertainty distribution for a sample of time lags $\bar{\rho}_{\mathbf{\Delta\hat{T}}}(\mathbf{\Delta\hat{t}}) = \frac{1}{m}\sum_{i=1}^{m}\delta\left(\mathbf{\Delta\hat{t}} - \mathbf{\Delta t}_j\right)$, the corresponding distribution for the test statistic $\hat{Z} = z(\mathbf{\Delta\hat{T}})$ is formed by the set $\{z(\mathbf{\Delta t}_j)|j \in [1,m]\}$ where each $\mathbf{\Delta t}_j$ is a vector in $n$ dimensions. The uncertain (left-handed) $p$-value remains to be derived from $\bar{\rho}_{\hat{Z}}(\hat{z})$:

$$\bar{\rho}_{\hat{P}_z}(\hat{p}_z) = \int \delta\left(\hat{p}_z - \int_{-\infty}^{\hat{z}} t_{n-1}(z)\,dz\right)\bar{\rho}_{\hat{Z}}(\hat{z})\,d\hat{z} \qquad = \frac{1}{m}\sum_{j=1}^{m}\delta\left(\hat{p}_z - \underbrace{\int_{-\infty}^{z_j} t_{n-1}(z')\,dz'}_{=p_{z,j}}\right).$$ (A14)

Finally, the practical computation of the uncertain $p$-values boils down to an application of the test to all members of the set $\mathbf{\Delta t}_j$ that originates from the MCMC sampling used to approximate the posterior probability density for the ramp parameter configuration $\Theta$. For the WSR test the expression

$$\bar{\rho}_{\hat{P}_w}(\hat{p}_w) = \frac{1}{m}\sum_{j=1}^{m}\delta(\hat{p}_w - p_{w,j}) \quad \text{with} \quad p_{w,j} = p_w(\mathbf{\Delta t}_j)$$ (A15)

can be derived analogously. The bootstrap test bears the particularity that each $\mathbf{\Delta t}_j$ induces its own null distribution. Yet, the application of the test to each individual $\mathbf{\Delta t}_j$ induces a set of $p_{v,j} = p_v(\mathbf{\Delta t}_j)$ that determines the empirical density

$$\bar{\rho}_{\hat{P}_v}(\hat{p}_v) = \frac{1}{m}\sum_{j=1}^{m}\delta(\hat{p}_v - p_{v,j}).$$ (A16)

## Appendix B: Results of the analysis for the control group

As explained in Sec. A, we drastically reduce the cardinality of the sets that form the empirical densities $\bar{\rho}_{\Delta\hat{T}^{p,q}}(\Delta\hat{t}^{p,q})$ at two points in the analysis. First, for the representation of the uncertain time lag $\Delta\hat{T}_i^{p,q}$ between the proxies $p$ and $q$ at a given DO event, only 6000 out of the $6000^2$ possible values are utilized. Second, the set of vectors considered in the representation of $\bar{\rho}_{\Delta\hat{T}^{p,q}}(\Delta\hat{t}) = \frac{1}{6000}\sum_{j=1}^{6000}\delta(\Delta\hat{t}^{p,q} - \Delta t_j^{p,q})$ is comprised of only 6000 out of the $6000^{16}$ theoretically available vectors. To cross-check the robustness of the results obtained within the limits of this approximation, we applied our analysis to a control group of 9 alternative realizations of the empirical uncertainty density for $\Delta\hat{T}^{p,q}$ for each proxy pair. The control group uncertainty densities are constructed as follows: First, the empirical uncertainty distributions for the event specific lags $\Delta\hat{T}_i^{p,q}$ are obtained via Eq. A6. In a second step, the joint empirical uncertainty distribution for $\Delta\hat{T}^{p,q}$ is constructed from randomly shuffled empirical sets $\Delta t_{i,s_i(j)}^{p,q}$ of each DO event:

$$\tilde{\rho}_{\Delta\hat{T}^{p,q}}^{ctrl}(\Delta\hat{t}^{p,q}) = \frac{1}{m}\sum_{j=1}^{m}\prod_{i=1}^{n}\delta(\Delta\hat{t}_i^{p,q} - \Delta t_{i,s_i(j)}^{p,q}). \tag{B1}$$

Here $s_i$ denotes an event specific permutation of the index set $\{1, ..., 6000\}$. Thus the empirical $\Delta t_{i,j}^{p,q}$ recombine between events and give rise to a new set of 6000 vectors that constitute 6000 empirical realizations of the uncertain $\Delta\hat{T}^{p,q}$.

The results obtained from the control runs show only minor deviations from the results presented in the main text and thus confirm the validity of the reduction of the corresponding sets. Tab. B1 summarizes the results obtained by application of the hypothesis tests to the control group.

## Appendix C: Computation of the uncertain sample mean

In the main text, we stated that the uncertain sample mean is given by the pairwise convolution of the individual uncertainty distributions that describe the uncertain sample members. Here, we show how the uncertain sample mean can be computed if the individual uncertainty distributions are known.

Consider $n$ random variables which are independently, yet not identically distributed:

$$\mathbf{X} = (X_1, ..., X_n) \quad \text{with} \quad X_i \sim \rho_{X_i}(x_i)\, dx_i \tag{C1}$$

in analogy to the

$$\Delta\hat{T}^{p,q} = (\Delta\hat{T}_1^{p,q}, ..., \Delta\hat{T}_n^{p,q}) \quad \text{with} \quad \Delta\hat{T}_i^{p,q} \sim \rho_{\Delta\hat{T}_i^{p,q}}(\Delta\hat{t}_i^{p,q}) \tag{C2}$$

from the main text. Further, let

$$U = \frac{1}{n}\sum_{i=1}^{n}X_i \tag{C3}$$

**Table B1.** Results obtained from the application of hypothesis tests to the control group. Reported are the mean $p$-values $\mathrm{E}(p(\mathbf{\Delta\hat{T}}))$ together with the probability of the uncertain sample to be smaller than the significance level $\pi(p(\mathbf{\Delta\hat{T}}) < 0.05)$ and the $p$-values of the expected samples $p(\mathrm{E}(\mathbf{\Delta\hat{T}}))$ for all three tests. All results were derived from the corresponding empirical densities $\bar{\rho}_{\mathbf{\Delta\hat{T}}^{p,q}}(\mathbf{\Delta\hat{t}}^{p,q})$. The results from the original analysis are given as well by the $p - q - 0$ run for each proxy variable.

| proxies | run | $\mathrm{E}(\hat{P})$ | | | $\pi(\hat{P} < 0.05)$ | | | $p(\mathrm{E}(\mathbf{\Delta\hat{T}}))$ | | |
|---|---|---|---|---|---|---|---|---|---|---|
| | | z | w | bs | z | w | bs | z | w | bs |
| NGRIP:$Ca^{2+}$-$Na^+$ | 0 | 0.219 | 0.168 | 0.217 | 0.258 | 0.324 | 0.299 | 0.044 | 0.009 | 0.02 |
| NGRIP:$Ca^{2+}$-$Na^+$ | 1 | 0.218 | 0.166 | 0.215 | 0.246 | 0.316 | 0.292 | 0.044 | 0.009 | 0.019 |
| NGRIP:$Ca^{2+}$-$Na^+$ | 2 | 0.219 | 0.165 | 0.216 | 0.258 | 0.324 | 0.294 | 0.044 | 0.009 | 0.018 |
| NGRIP:$Ca^{2+}$-$Na^+$ | 3 | 0.22 | 0.166 | 0.217 | 0.254 | 0.322 | 0.295 | 0.044 | 0.009 | 0.02 |
| NGRIP:$Ca^{2+}$-$Na^+$ | 4 | 0.219 | 0.166 | 0.217 | 0.255 | 0.32 | 0.296 | 0.044 | 0.009 | 0.02 |
| NGRIP:$Ca^{2+}$-$Na^+$ | 5 | 0.218 | 0.166 | 0.216 | 0.254 | 0.319 | 0.293 | 0.044 | 0.009 | 0.019 |
| NGRIP:$Ca^{2+}$-$Na^+$ | 6 | 0.219 | 0.167 | 0.217 | 0.255 | 0.319 | 0.299 | 0.044 | 0.009 | 0.021 |
| NGRIP:$Ca^{2+}$-$Na^+$ | 7 | 0.219 | 0.167 | 0.217 | 0.252 | 0.319 | 0.295 | 0.044 | 0.009 | 0.019 |
| NGRIP:$Ca^{2+}$-$Na^+$ | 8 | 0.219 | 0.164 | 0.217 | 0.257 | 0.316 | 0.3 | 0.044 | 0.009 | 0.02 |
| NGRIP:$Ca^{2+}$-$Na^+$ | 9 | 0.218 | 0.165 | 0.216 | 0.261 | 0.32 | 0.302 | 0.044 | 0.009 | 0.02 |
| NGRIP:$\lambda$-$Na^+$ | 0 | 0.093 | 0.091 | 0.086 | 0.469 | 0.484 | 0.524 | 0.023 | 0.017 | 0.017 |
| NGRIP:$\lambda$-$Na^+$ | 1 | 0.092 | 0.091 | 0.085 | 0.467 | 0.482 | 0.516 | 0.023 | 0.017 | 0.014 |
| NGRIP:$\lambda$-$Na^+$ | 2 | 0.093 | 0.092 | 0.086 | 0.462 | 0.489 | 0.519 | 0.023 | 0.017 | 0.015 |
| NGRIP:$\lambda$-$Na^+$ | 3 | 0.092 | 0.09 | 0.085 | 0.465 | 0.482 | 0.516 | 0.023 | 0.017 | 0.016 |
| NGRIP:$\lambda$-$Na^+$ | 4 | 0.093 | 0.09 | 0.086 | 0.471 | 0.488 | 0.529 | 0.023 | 0.017 | 0.014 |
| NGRIP:$\lambda$-$Na^+$ | 5 | 0.093 | 0.092 | 0.086 | 0.468 | 0.492 | 0.522 | 0.023 | 0.017 | 0.015 |
| NGRIP:$\lambda$-$Na^+$ | 6 | 0.092 | 0.089 | 0.085 | 0.47 | 0.488 | 0.521 | 0.023 | 0.017 | 0.013 |
| NGRIP:$\lambda$-$Na^+$ | 7 | 0.092 | 0.091 | 0.085 | 0.461 | 0.486 | 0.515 | 0.023 | 0.017 | 0.016 |
| NGRIP:$\lambda$-$Na^+$ | 8 | 0.093 | 0.091 | 0.086 | 0.477 | 0.486 | 0.525 | 0.023 | 0.017 | 0.015 |
| NGRIP:$\lambda$-$Na^+$ | 9 | 0.093 | 0.091 | 0.086 | 0.475 | 0.488 | 0.524 | 0.023 | 0.017 | 0.015 |
| NGRIP:$Ca^{2+}$-$\delta^{18}O$ | 0 | 0.234 | 0.182 | 0.233 | 0.235 | 0.306 | 0.262 | 0.042 | 0.015 | 0.037 |
| NGRIP:$Ca^{2+}$-$\delta^{18}O$ | 1 | 0.234 | 0.182 | 0.232 | 0.231 | 0.294 | 0.257 | 0.042 | 0.015 | 0.035 |
| NGRIP:$Ca^{2+}$-$\delta^{18}O$ | 2 | 0.234 | 0.18 | 0.232 | 0.226 | 0.3 | 0.254 | 0.042 | 0.015 | 0.039 |
| NGRIP:$Ca^{2+}$-$\delta^{18}O$ | 3 | 0.234 | 0.182 | 0.233 | 0.236 | 0.314 | 0.261 | 0.042 | 0.015 | 0.036 |
| NGRIP:$Ca^{2+}$-$\delta^{18}O$ | 4 | 0.234 | 0.181 | 0.232 | 0.234 | 0.308 | 0.261 | 0.042 | 0.015 | 0.031 |
| NGRIP:$Ca^{2+}$-$\delta^{18}O$ | 5 | 0.233 | 0.181 | 0.231 | 0.23 | 0.304 | 0.253 | 0.042 | 0.015 | 0.032 |
| NGRIP:$Ca^{2+}$-$\delta^{18}O$ | 6 | 0.234 | 0.181 | 0.232 | 0.228 | 0.306 | 0.253 | 0.042 | 0.015 | 0.037 |
| NGRIP:$Ca^{2+}$-$\delta^{18}O$ | 7 | 0.234 | 0.18 | 0.232 | 0.235 | 0.31 | 0.261 | 0.042 | 0.015 | 0.033 |
| NGRIP:$Ca^{2+}$-$\delta^{18}O$ | 8 | 0.234 | 0.183 | 0.232 | 0.236 | 0.313 | 0.263 | 0.042 | 0.015 | 0.034 |
| NGRIP:$Ca^{2+}$-$\delta^{18}O$ | 9 | 0.234 | 0.182 | 0.232 | 0.231 | 0.307 | 0.257 | 0.042 | 0.015 | 0.035 |

**Table B2.** Continuation of Tab.B1.

| proxies | run | $E(\hat{P})$ | | | $\pi(\hat{P} < 0.05)$ | | | $p(E(\mathbf{\Delta}\hat{\mathbf{T}}))$ | | |
|---|---|---|---|---|---|---|---|---|---|---|
| | | z | w | bs | z | w | bs | z | w | bs |
| NGRIP:$\lambda$-$\delta^{18}$O | 0 | 0.133 | 0.11 | 0.129 | 0.369 | 0.436 | 0.414 | 0.024 | 0.009 | 0.017 |
| NGRIP:$\lambda$-$\delta^{18}$O | 1 | 0.134 | 0.111 | 0.129 | 0.37 | 0.441 | 0.416 | 0.024 | 0.009 | 0.016 |
| NGRIP:$\lambda$-$\delta^{18}$O | 2 | 0.133 | 0.11 | 0.127 | 0.379 | 0.44 | 0.422 | 0.024 | 0.009 | 0.017 |
| NGRIP:$\lambda$-$\delta^{18}$O | 3 | 0.135 | 0.112 | 0.13 | 0.38 | 0.435 | 0.42 | 0.024 | 0.009 | 0.017 |
| NGRIP:$\lambda$-$\delta^{18}$O | 4 | 0.134 | 0.111 | 0.129 | 0.378 | 0.442 | 0.419 | 0.024 | 0.009 | 0.018 |
| NGRIP:$\lambda$-$\delta^{18}$O | 5 | 0.133 | 0.109 | 0.128 | 0.373 | 0.437 | 0.416 | 0.024 | 0.009 | 0.018 |
| NGRIP:$\lambda$-$\delta^{18}$O | 6 | 0.133 | 0.111 | 0.128 | 0.384 | 0.446 | 0.426 | 0.024 | 0.009 | 0.017 |
| NGRIP:$\lambda$-$\delta^{18}$O | 7 | 0.133 | 0.109 | 0.128 | 0.376 | 0.445 | 0.416 | 0.024 | 0.009 | 0.017 |
| NGRIP:$\lambda$-$\delta^{18}$O | 8 | 0.134 | 0.11 | 0.129 | 0.381 | 0.443 | 0.424 | 0.024 | 0.009 | 0.018 |
| NGRIP:$\lambda$-$\delta^{18}$O | 9 | 0.134 | 0.11 | 0.129 | 0.376 | 0.441 | 0.418 | 0.024 | 0.009 | 0.019 |
| NEEM:$Ca^{2+}$-$Na^+$ | 0 | 0.08 | 0.076 | 0.074 | 0.566 | 0.584 | 0.61 | 0.008 | 0.007 | 0.006 |
| NEEM:$Ca^{2+}$-$Na^+$ | 1 | 0.08 | 0.076 | 0.074 | 0.57 | 0.581 | 0.61 | 0.008 | 0.007 | 0.006 |
| NEEM:$Ca^{2+}$-$Na^+$ | 2 | 0.079 | 0.075 | 0.073 | 0.571 | 0.587 | 0.614 | 0.008 | 0.007 | 0.005 |
| NEEM:$Ca^{2+}$-$Na^+$ | 3 | 0.079 | 0.076 | 0.073 | 0.573 | 0.586 | 0.615 | 0.008 | 0.007 | 0.006 |
| NEEM:$Ca^{2+}$-$Na^+$ | 4 | 0.08 | 0.077 | 0.074 | 0.572 | 0.584 | 0.612 | 0.008 | 0.007 | 0.005 |
| NEEM:$Ca^{2+}$-$Na^+$ | 5 | 0.08 | 0.076 | 0.074 | 0.571 | 0.579 | 0.608 | 0.008 | 0.007 | 0.005 |
| NEEM:$Ca^{2+}$-$Na^+$ | 6 | 0.08 | 0.077 | 0.074 | 0.565 | 0.577 | 0.609 | 0.008 | 0.007 | 0.006 |
| NEEM:$Ca^{2+}$-$Na^+$ | 7 | 0.08 | 0.077 | 0.074 | 0.57 | 0.583 | 0.612 | 0.008 | 0.007 | 0.006 |
| NEEM:$Ca^{2+}$-$Na^+$ | 8 | 0.079 | 0.075 | 0.073 | 0.57 | 0.58 | 0.614 | 0.008 | 0.007 | 0.006 |
| NEEM:$Ca^{2+}$-$Na^+$ | 9 | 0.078 | 0.075 | 0.072 | 0.567 | 0.576 | 0.608 | 0.008 | 0.007 | 0.006 |

denote the mean of the sample of random variables, which is in turn a random variable by itself. In order to compute the distribution $\rho_U(u)\, du$ we introduce the variable $V = nU$ and the sequence of variables

$$V_j = \sum_{i=1}^{j} X_i, \tag{C4}$$

such that $V_n = V$. From C4 it follows that

$$V_{j+1} = V_j + X_{j+1} \tag{C5}$$

and hence

$$\rho_{V_{j+1}}(v_{j+1})\,dv_{j+1} = \int\limits_{-\infty}^{\infty}\int\limits_{-\infty}^{\infty}\rho_{V_j}(v_j)\,\rho_{X_{j+1}}(x_{j+1})\,\delta(v_{j+1}-v_j-x_{j+1})\,dx_{j+1}\,dv_j\,dv_{j+1}$$

$$= \int\limits_{-\infty}^{\infty}\rho_{V_j}(v_j)\,\rho_{X_{j+1}}(v_{j+1}-v_j)\,dv_j\,dv_{j+1}. \quad \text{(C6)}$$

Self-iteration of C6 yields

$$\rho_{V_{j+1}}(v_{j+1})\,dv_{j+1}$$

$$= \int\limits_{-\infty}^{\infty}\int\limits_{-\infty}^{\infty}\underbrace{\rho_{V_{j-1}}(v_{j-1})\,\rho_{X_j}(v_j-v_{j-1})\,dv_{j-1}}_{=\rho_{V_j}(v_j)}\,\rho_{X_{j+1}}(v_{j+1}-v_j)\,dv_j\,dv_{j+1}$$

$$=\ldots$$

$$= \int\limits_{-\infty}^{\infty}\ldots\int\limits_{-\infty}^{\infty}\prod_{i=1}^{j+1}\rho_{X_i}(v_i-v_{i-1})\,dv_{i-1}\,dv_{j+1}, \quad \text{(C7)}$$

where $v_0 = 0$. With $V_n/n = U$ the distribution for the uncertain sample mean reads

$$\rho_{V_n}(v_n)\,dV_n = \rho_{V_n}(nu)\,n\,du = \rho_U(u)\,du \quad \text{(C8)}$$

and thus

$$\rho_U(u)\,du = \int\limits_{-\infty}^{\infty}\prod_{i=1}^{n}\rho_{X_i}(v_i-v_{i-1})\,dv_{i-1}\,n\,du, \quad \text{(C9)}$$

with $v_0 = 0$ and $v_n = nu$.