# Peer review of "Significance of uncertain phasing between the onsets of stadial-interstadial transitions in different Greenland ice-core proxies"

_Climate of the Past, 2020_

## Referee Comment (RC1) · Anonymous Referee #1 · 14 Dec 2020

In their manuscript Riechers and Boers present a method for the statistical analysis of the results presented in Erhardt et al. (2019) with the goal to determine the average phasing of the onset of Greenland Interstadials (GIS). To do so, they present three different approaches with the same goal: To account for between-event variability. This additional layer of variability, as pointed out by the Authors (as well as more implicitly by Erhardt et al., 2019), was assumed to be non-existent to derive the averaged estimates in the original study. The exemplary application of their method highlights the importance of accounting for this additional layer of variability and underlines the difficulties of investigating changes in a very tightly coupled system under uncertainty.

Overall, the paper is very well structured and written and the presented methods provide an interesting toolset for a range of questions. The derivations of the methods are presented very thoroughly, and all necessary technical information is present. For the broader paleo-climate community (which is the audience of CP) the method descriptions could be supplemented with more explanations to aid intuition and prolonged use of the innovative approaches.

Unfortunately, and this is my biggest concern, the authors focus the manuscript entirely on their methods and only provide one brief albeit very intriguing example at the end. This is exceptionally disappointing as the analysis could easily be extended to the other records presented in Erhardt et al (2019), especially given the simple calculations needed for the presented methods. The answer of the authors to this concern will likely be that this analysis will follow in a later paper. However, this raises the question whether the manuscript in its current form qualifies for a journal that is not focused on statistical methods but the climate of the past. In the present form, the paper is basically a method collection and is better suited for a different i.e., method focused journal. I thus strongly urge the authors to also add the results for the other records presented in the original study. This would not only demonstrate the viability of their method but would also further our understanding of the climate dynamics during the onsets of an GIS. Additionally, this would allow the authors to put their results into perspective when compared to the vast amount of research (other than Erhardt et al .2019) that exists on the records and mechanisms DO events which presently is sorely lacking from both discussion and conclusions. With these additions, the resulting paper would be made much more valuable to the broader paleo-climate community.

Throughout the text there are a number of inaccuracies such as wrongly stated ages, confusion of fluxes and concentrations, the nature of ice core records and MCMC. Even though each on their own might seem minor, I will warn the authors that they could be

interpreted as negligence. I thus strongly encourage the authors to seek out the input of experts in ice-core records (of which there are plenty in the TiPES project) to give the manuscript a thorough once-over to avoid potential pitfalls.

**Specific remarks**

*L18:* additional the

*Figure 2:* The vertical lines are colored blue. This is probably an accident. Please also add the full list of references for the datasets shown in the Figure as well as the age-scale to either the caption of the text.

*L64ff:* What is high? Both for the statements about the record resolution as well as about the variability choice of a relative term is only useful if it is clear what the resolution or frequency is high in relation to. It would be much better to state at least orders of magnitude for these instead.

*Table 1:* The caption is a bit misleading: In the data that you use in your study only the DO events given in bold are contained. Furthermore, the statement "the stochastic, MCMC-based method successfully detected empirical density distributions for transition onset" is inaccurate on multiple levels: To begin with the method is probabilistic, not stochastic, secondly, the method does not detect empirical density distributions but provides those for the transitions. Following the description of the model in Erhardt et al., 2019, it seems likely, that the investigated transitions where chosen because they could be described well enough with the ramp model. This seems especially likely as the very short sub-events are sometimes poorly defined in the ice core record and/or often exhibit too short stable levels before or after. This is also stated in the text in multiple occasions. The ages stated in the Table are numerically identical with the ones provided in Rasmussen et al. (2014), that means that the age reference is than in fact not 1950 but the year 2000. Please check this throughout the manuscript to make sure

that the correct ages are used at all times. It is also advisable to avoid the use "BP" to avoid confusion with Radiocarbon ages.

*L78ff:* Please elaborate if you extended or changed the original approach by Erhardt et al. or used it as is. Judging from the code/data availability statement the latter seems to be true. Should that be the case this needs a clear statement to distinguish prior from original work.

*L85:* Consider not using the variable name here at is it only fully introduced and used much later in the manuscript.

*L95:* Please consider to not cite the pangea reference separately as it is technically only a supplement to the 2019 study by Erhardt et al. Having both mentioned separately seems contrary to the notion of a supplement.

*L98:* Ice core record is technically not compressed in greater depth but rather extended in the horizontal due to glacial flow which in turn leads to a thinning and has nothing to do with compression due to hydrostatic pressure. Please use the correct term "thinning". Deposition rates and concentrations are two fundamentally different things, from what I gather you are using concentration records only. Please correct "deposition rates" to concentrations.

*L107f:* Because the algorithm is based in MCMC it by design (and necessity, as the problem has no analytical solution) returns samples from the posterior distribution, not a probability density function. The probability density functions are later only approximated using kernel density estimates.

*L114ff:* I appreciate the authors desire to advertise their approaches to a wider audience. However, the provided example is completely irrelevant in the context of the readership of this journal. Furthermore, it is somewhat contradictory because if it were true, then the statistical approaches outlined in the manuscript are hopefully in fact not new but long solved in the medical context. Please find a better suited example and

elaborate why the problem of inferring a population mean from an uncertain measurement is not yet solved? And if it is solved, under which assumptions is it solved and how do your assumptions differ?

*Figure 2:* How is it possible, that the pdf for $\Delta t$ is unimodal if one of the pdfs for the transition onsets is bimodal? Please elaborate.

*L123:* See comment above about the product of MCMC.

*L129:* See comment above about the statement of failure of the MCMC algorithm. Please either remove the statement or elaborate.

*L147f:* The investigation has basically been already performed already in the original publication (see Figures A1 and A2 in Erhardt et al. 2019). Furthermore, the test data the Authors use here violates an important and explicit assumption of the algorithm: autocorrelation of the noise (i.e. an autocorrelation time larger than zero) and are thus rendering the tests invalid. I appreciate the intention of the authors here, but the oversight of the white-nose vs red-noise assumption is disappointing at best. I suggest the authors remove this section entirely.

*L184:* Please elaborate on the statement why hierarchical distributional models cannot be invoked here and better define the term.

*L196:* As a side note: Summing does not require to keep all summands in memory.

*L195ff:* This paragraph makes an interesting point as the samples of $\Delta t$ for each of the transitions are interchangeable, they could technically be reshuffled to simulate more samples. Could you elaborate on the uncertainty that this sub-sampling adds to your methods and how much the results are dependent on the individual realization? If the results are not stable it might hint at the fact that the 6000 samples from what is a 16-dimensional distribution might not be enough to fully capture all of the uncertainty.

*L205-213:* In this short section, the authors provide the arguably most elegant way of drawing inference on the population from the underlying data. Even though this section is a little bit hidden and Eq. (9) is not straight forward to understand, the resulting convolution of the individual posteriors for $\Delta t_i$ provides a posterior for the average lag of the DO events. My judgment of this section being the most elegant stems from the fact that it is not dependent on any additional assumption such as the presence of an infinite number of DO events or normality of any of the distributions – It rather only answers the question which average lags are consistent with the observed 16 DO events, which is arguably the question the authors set out to answer. Comparing this to the estimates that Erhardt et al. call the "combined evidence" the difference stems from the fact that Erhardt et al. assume that all DO events exhibit an archetypical lag, i.e. one that does not vary between events or verbatim: "[. . .] this implicitly assumes that the timing differences for all interstadial onsets in the parameters investigated here are the result of the same underlying process or, in other words, are similar between the interstadial onsets" The method presented here relaxes this assumption by realizing that the averaging can be expressed as summation and thus as a convolution of the posterior densities. In comparison to the convolution described here, it is very important to note, that both the t-distribution approach as well as the bootstrap approach described later aim at something subtle, yet fundamentally different: The convolution provides the probability of a mean lag given the observations. The other two however assess the distribution of this mean under their respective assumptions! I will take the liberty to encourage the authors to treat this section entirely separately from the other approaches t and to extent it a little bit to emphasize its difference to the other methods. As a side note/word of caution: The accuracy of the numerical convolution of the kernel density estimates is very much dependent on the chosen discretization and especially range of values that it is performed for. This can easily be tested when comparing distributions where the convolution is known to their numerical convolution (such as the Normal distribution). I suggest the authors do some experiments and present these in the appendix.

*L215ff:* The "refinement" that the authors present by using the definition of the t-distribution and a change in variables to estimate the mean of the underlying distribution hinges on the assumption that this is a normal distribution. Even though this assumption seems inconspicuous at first sight, the authors should provide evidence that this assumption is both justified as well as not violated by the samples from the ramp-fit. Depending on the justification of the assumption or consistency with the data the results that are based on the assumptions will not be valid or should at least be interpreted with care. I suggest the authors spent some time in this section (and all other sections) to clearly (and in words) state the underlying assumptions, their implications and justifications.

*L243ff:* The authors try to justify their choice in the t-distribution based estimation by a presenting a bootstrapping version of the same thing. However, I do not think that this can be used to do so. Looking at the results the agreement between $\rho(\mu)$ and $\rho_{bs}(\mu)$ made me wonder why that might be the case: In fact, no matter what randomly generated data the approaches are use on, the results always very closely agree with each other. This is also seemingly independent from the data being normally distributed or not. This seems slightly odd to me however the reason might be, that both methods fundamentally do the same thing: they aim to estimate the mean and the standard error of the mean from the sample and provide a distribution about mean given this standard error. To assure the reader of the validity of their approach, the authors should spend some time elaborating on this and clarify the rationale of why the bootstrap of the mean should be different than the t-distribution and what we can actually learn from having both if they yield seemingly identical results. Furthermore, the authors should add how many bootstrap samples where generated as this is an important information for the reproducibility of their study.

*L261ff:* After going through a lot of trouble to derive ways to estimate the distribution of the mean from uncertain observations the authors opt to throw all of this overboard and to start from scratch to come up with a way to test whether this mean is different from zero. I am a little bit puzzled why the authors present, what basically amounts to calculating a p-value for each of the MCMC samples of 16 $\Delta t_i$ rather than investigating

the distribution of the mean that they just derived. I am sure that the results would likely be not much different. It also remains unclear to me, whether this "propagation of uncertainty to the p-value" is actually a valid approach: Essentially each of the 6000 p-values that is calculated constitutes the p-value of the test of that one sample is from a distribution different to zero. The meaning of the distribution of these p-values over many repeated samples is not straight forward. This starts with the observation that for a non-significant distance the resulting p-values will be uniformly distributed, so the distribution of p-values that the authors present needs to be interpreted within that context. Though the approach the authors present might seem convenient and maybe even clever it leaves me with more questions than answers. And with this I am not arguing against the conclusions the authors arrive with using this method, as anything but a non-significant lead would be surprising given the assumptions of the calculations. What is also missing from the otherwise extensive presentation is the alternative view, that the 6000 MCMC samples each present 16 observations of the mean and could be tested accordingly as 6000*16 observations. I am sure that there is a good reason to not do this, but this alternative should at least be mentioned and discussed in the context of the other methods. I suggest the authors either rework this section entirely to better justify and clarify their approach and to include an investigation of the derived distributions of the mean or complete refrain from presenting hypothesis tests in this context. Should the authors decide to keep this section, they need to make sure to state the Null Hypothesis (and the alternative hypothesis, depending how closely they follow Fisher) correctly and explicitly.

*L496ff:* The authors observe that the distribution that results from the convolution is narrower than the one obtained by the other methods. This is interesting albeit not surprising, given the conceptual difference of the convolution to the other methods. The brief explanation that the authors give here is quite difficult to follow, could the authors elaborate?

*L406f:* I think this point deserves a moment of attention: Despite the added layer of

uncertainty for the posterior distribution of $U_{\Delta t}$ and the additional assumptions going into $\mu_{\Delta t}$ both still put around 4/5 of the probability on lead of Ca over Na. Yes, this is not 90/100, but in IPCC parlance it is still likely that the transitions are led by a transition in Ca.

*L443ff:* The calculation of the number of events being consistent with the lead of Ca over Na again is a very good addition to the discussion and a great extension of the order statistics shown in Fig 5 of Erhardt et al. (2019). The authors interpretation of the analysis is however somewhat strongly formulated given the large uncertainties of the estimates: The postulate that if an atmospheric circulation change (effecting only Ca) would trigger the sea ice retreat of the DO events (in turn effecting Na) than all of the events should show a lead of Ca over Na at their onset. This is not wrong but would only ever occur in a scenario where we would observe these atmospheric and sea ice changes directly and without error, not through a set of proxy records and could reasonably exclude any influence of internal climate variability. All in all, that seems to comprise quite a high bar. I suggest the authors to tone down the interpretation of this otherwise very enlightening analysis.

*L468f:* How do the authors arrive at the conclusion that the observations cannot be used to investigate the transitions with Na leading? To put it sarcastically: If that is not possible, then why is the reverse, investigating the transitions with a Ca lead?

*471ff:* In the conclusions the authors very carefully and thoroughly present and interpret their results. Overall, I tend to agree with their conclusions based on the methods and the result that they presented.

*480ff:* The statement on the ability of the presented results to serve as evidence is somewhat unjustified. I do agree that on the base of the presented data the Null Hypothesis of a zero or larger lead cannot be rejected but in reverse that does not mean that the same evidence cannot be used at a later stage (combined with prior knowledge and more evidence).

*486ff:* The possible existence of a process other than the processes that directly influence Ca or Na is an important note here and is likely the best explanation for what is visible in the data. The authors could spend a little more time on this point.

––––––––––––––––––––––––––––

---

## Referee Comment (RC2) · Anonymous Referee #2 · 4 Jan 2021

Summary

This is an interesting study that re-investigates the temporal order of abrupt changes in different Greenland ice core proxies leading up to Dansgaard-Oeschger events. Compared to a previous paper by Erhardt et al, the authors use different methods and an alternative interpretation of the timing differences as a random variable that can be different from event to event. With this, they arrive at a different conclusion, namely

that due to the uncertainties in estimating the onsets of the abrupt transitions, we cannot exclude that there was no systematic lag of the abrupt transitions in either of the Greenland proxies. The paper is quite heavy on the mathematical and methodological aspects. To be more appealing to CP readers, I hope with the following comments to encourage the authors to better motivate why their approach can indeed be beneficial in resolving the problem at hand, or similar problems that readers might encounter in their own research.

General Comments

1. The authors do not specifically discuss how their approach is different to Erhardt et al 2019. While the manuscript gives the impression that previous studies completely disregarded uncertainties and used expectation values, this is not true for Erhardt et al, who included the uncertainties in a rigorous manner in their own right. The difference lies in the interpretation of the estimated onsets as random variables. Erhardt et al consider the uncertain samples as measurements of the same fixed quantity (a fixed time lag of Ca and Na at DO onsets), which allows them to simply multiply the individual MCMC posteriors to obtain a single posterior distribution that represents the measurement uncertainty of the fixed time lag. In contrast, the present study interprets the time lag to be a random variable, varying in between DO events. Thus, they cannot multiply the individual posteriors. In the discussion, the authors contrast their approach with the results one would obtain when completely discarding the uncertainties in the onset determination. It is not very surprising that including uncertainty yields hypothesis tests which are no longer significant. Instead, it would be more relevant to highlight the contrasting results of their approach to Erhardt et al.

2. The authors speak of a rigorous propagation of uncertainties to p-values, among other things. To achieve this, they introduce probability distributions of the mean and other test statistics, which are formally represented using delta distributions arising from the empirical sample. In practice, these distributions are all computed by summing a random sample from the individual MCMC posteriors, which is essentially bootstrapping if I understand correctly. Can the authors comment on how the results of this work would be different if they simply summed the individual MCMC posteriors (which would be equivalent to bootstrapping as well), and then looked at the arising distribution of the lag, determining the probability of a lag >=0? This would be much simpler and the obvious alternative approach to the multiplication of the posteriors by Erhardt et al. It would probably also give a non-significant result regarding a Ca2+ lead.

3. I am wondering in what way the hypothesis tests introduced in Sec. 3.6-3.7 are necessary and add to the results? When reading the manuscript, I was surprised that hypothesis tests were introduced after distributions for the sample or population mean had already been given, which would directly allow to test the hypothesis of a mean >=0?

4. Why did the authors restrict the analysis to Na and Ca, and omitted an analysis of the offsets of d18O and lambda? This would be an important consistency test, and might even be more relevant since lambda has a very direct meaning as a proxy, and d18O is the most important of all proxies.

Specific Comments

L15ff: I think the conclusions should be stated differently. In my interpretation, the analysis does not contradict a lead or lag. Rather, as a result of the large uncertainties in estimating the individual onset timings, it cannot be excluded that there are in fact no leads or lags. Similarly, on the grounds of the uncertainties in the individual event timings, there is not enough evidence to conclude that atmospheric reorganization systematically preceded sea ice retreat for all events.

L34-35: The interstadials lasted up to 10 millennia, with 1.5 millennia being the average.

L45-49: While I understand that this discussion is primarily about the abrupt onset of DO events, it might be worthwhile to mention that there is evidence that sea ice

and atmospheric proxies already start to gradually change much earlier than d18O in stadials (see Sadatzki et al, Sci Adv 5, 2019 and Lohmann, GRL 46, 2019).

L130: Why does the transition detection fail for some events?

L150: How would this depend on the autocorrelation of the noise? The Na and Ca records might have different noise structure and thus there is the potential for systematic biases indeed.

Eq. 7: Maybe the authors can elaborate specifically in the text what this approximation does. Since the order of MCMC samples for each event is arbitrary, by associating the i-th MCMC sample for every event, it seems like it is just a random sampling of m=6000 points in the joint space. If this is the case, why not simply sample randomly in the first place, and why not choose many more than m=6000 points? Or is this rather done in order to simplify the notation?

L278ff: I am not sure why the authors say that they are only given "relative" data, since they estimate the onsets in the two proxies. Furthermore, since until this point the data was already given exclusively as onset timing differences, I wonder why it is necessary to introduce "paired samples" now? This is a bit confusing to the reader.

L348: I might have missed this somewhere, but how is the distribution of the test statistic under the null hypothesis constructed?

L396: Could the authors explain more explicitly why the sample and population mean distributions are so different, and what this means for the interpretation of the results? Which distribution should be preferred?

L439: "...might simply be a stochastic feature." This is bit unclear. Maybe it would be better to write "...might simply occur by chance due to the small sample size."

L441ff: Here the authors introduce another simple method to address the likelihood of a systematic Ca2+ lead. I think it would be more coherent if this were moved to the Results Section. Furthermore, I find the conclusions from this simple calculation too

confident, and in spirit contradictory to the interpretation of the other results. The probability of obtaining exactly n=16 events with a Ca lead will always be very small when there is a relatively large measurement uncertainty of the individual lags (spanning both positive and negative values), even if all individual posteriors would be clearly centered at negative values. Just like the probability of flipping 16 out of 16 heads is still very low for a strongly biased coin. This does not allow one to contradict the hypothesis that all events would follow the same pattern with a Ca2+ lead. For the data at hand, all but two events show posterior distributions centered at a negative value. Maybe the authors could write instead that from the MCMC posteriors it seems unlikely that all DO events occurred with a preceding abrupt change in Ca2+. However, this might merely reflect the fact that due to the large uncertainty in estimating the onsets, the individual MCMC posteriors have significant support for positive lags as well. Arguing like this would also be much more in line with the authors' earlier statements that they cannot infer an absence of causality from their non-significant tests.

Technical Comments

Figure 1 and Table 1: Just to be sure, can the authors confirm that the time scale they use really is years BP (before the year 1950 AD), and not years b2k (before the year 2000), which is what is commonly used in GICC05?

L18: ...holds true, the we conclude...

L98: Instead of "compression" rather say: ...due to the thinning of the annual layers in the core.

Figure 3: Maybe it would be good to choose a different color for the null hypothesis in panel b. Otherwise it gives the impression that it corresponds in some way to the blue distribution in panel a.

Eq. 14 and Eq. 16: I am wondering whether "u" in the second delta function should be replaced by the empirical mean sum(y_i)/n?

L235: Maybe "marginal distribution" would correspond better to the nature of this distribution?

Eq. 17: What is the notation $u_j$ and $s_j$?

Eq. 20: I am unfamiliar with this definition of a p-value. Is the integration not simply over all phi < phi_0 (for a one-sided test)? The definition here could lead to rather strange results for very asymmetric and long-tailed distributions.

L387: missing delta in the sum.

Figure 4: Larger fonts and panels would be nice for better visibility.

Eq. 35: It would be good if the authors could point to where the individual probabilities in the product come from.

---

## Author Comment (AC1) · 1 Feb 2021

We thank the referees for their very thorough and careful review. Following the constructive criticism and valuable feedback, we would like to propose several changes to the manuscript. We are convinced that these changes will substantially improve the quality and clarity of our manuscript and that they will address the referees' objections, questions and suggestions. Since both reviews share several general and important

aspects, we would first like to reply to these in a combined way here, and propose a substantial restructuring of the manuscript before we address the specific comments of this referee further below.

1. Most importantly, both referees urge us to expand the analysis to the full possible set of pairs of proxy variables to make the manuscript more appealing for CP-readers. After careful reconsideration, we agree with this criticism and would integrate the analysis of all pairs of proxies that are reported to show a clear lag-behaviour by Erhardt et al. in a revised version of the manuscript. We would not include those proxies, which Erhardt et al. find to transition simultaneously.

2. We acknowledge the comment made by referee 1, that the manuscript appears as a method collection and agree that there is a lack of guidance and motivation throughout the method section in the current version of the manuscript. In particular, both referee reports indicate that the role of the hypothesis tests hasn't been made sufficiently clear. In response to this, we would restructure the manuscript and focus on the key question the manuscript aims to answer: Can we rule out that the lag-tendency observed from a sample of 16 DO events arises by chance, with an underlying population mean of the lag equal to zero? This question can be directly answered in terms of the extended significance tests we carry out, and we therefore plan to put considerably more focus on this in a revised manuscript.

In the revised manuscript, we would then take the following, natural line of inference that should be immediately obvious to the reader:

1. Establishment of the statistical framework in terms of random experiments: The DO transition onset lags constitute an n=16 sample (for other proxy-pairs this number can deviate) drawn from an underlying population. Here, we would integrate a detailed comparison with the assumptions made by Erhardt et al. to derive what they call 'combined evidence', as requested by both referees.

2. Introduction of uncertainty propagation in the statistical inference in the case of uncertain samples.

3. Testing whether or not the observed sample contradicts the null-hypothesis of a population mean equal to (or greater than) zero, given the uncertainty of the sample. A population mean significantly different from zero can be interpreted as a systematic lag. Here, 'systematic' does not directly imply a causal relation but is a necessary condition for the inference of causality.

4. Comparison of the uncertain sample mean with the combined evidence reported by Erhardt et al.

We would introduce this streamlined methodological framework in the 'Methods' and show the results of its application to all proxy-pairs under study in the 'Results' section.

We would like to propose to remove the following from our manuscript in order to increase its stringency and readability:

- Remove the derivation of distributions for the population mean, because it is not required to answer the stated question.

- To simplify the presentation, if you agree, we would refrain from carrying out all the derivations for empirical densities and use continuous probability density functions instead. The equivalence between empirical densities (as provided by the MCMC) and continuous probability density functions in all our derivations would be shown in the appendix. While the formulation in terms of empirical densities has not explicitly been criticized by the referees, we noticed that it does not contribute to the understanding of the physical and statistical reasoning of the manuscript.

- Remove the discussion on the probability for n events to be lead by calcium. Originally, we used this derivation to reject a causal relation, but the objection of

referee 2 made us reconsider our method and we found argumentative inconsistencies.

These changes will result in a streamlined version of the manuscript and make it significantly easier for the reader to follow the main line of thought without getting distracted by a lot of technical details. We think that tightening of the methods together with the incorporation of the additional proxy pairs will yield a convenient balance for publication in CP.

As both referees proposed to treat all 16 * 6000 (6000 MCMC-samples for each of the 16 DO events) values as equal observations of the same quantity and gather all values to one single empirical probability distribution, we would add a section to explains why this cannot be done (for an explanation see General comment 2 of review 2 and similarly L.243 of review 1).

Point by point answer to referee 1:

*'In their manuscript Riechers and Boers present a method for the statistical analysis of the results presented in Erhardt et al. (2019) with the goal to determine the average phasing of the onset of Greenland Interstadials (GIS). To do so, they present three different approaches with the same goal: To account for between-event variability. This additional layer of variability, as pointed out by the Authors (as well as more implicitly by Erhardt et al., 2019), was assumed to be non-existent to derive the averaged estimates in the original study. The exemplary application of their method highlights the importance of accounting for this additional layer of variability and underlines the difficulties of investigating changes in a very tightly coupled system under uncertainty. Overall, the paper is very well structured and written and the presented methods provide an interesting toolset for a range of questions. The derivations of the methods are presented very thoroughly, and all necessary technical information is present. For the broader paleo-climate community (which is the audience of CP) the method descriptions could be supplemented with more explanations to aid intuition and prolonged use*

*of the innovative approaches.'*

We thank the referee for this positive feedback. A clear focus on the main line of reasoning will facilitate the readers' understanding in a revised manuscript.

*'Unfortunately, and this is my biggest concern, the authors focus the manuscript entirely on their methods and only provide one brief albeit very intriguing example at the end. This is exceptionally disappointing as the analysis could easily be extended to the other records presented in Erhardt et al (2019), especially given the simple calculations needed for the presented methods. The answer of the authors to this concern will likely be that this analysis will follow in a later paper. However, this raises the question whether the manuscript in its current form qualifies for a journal that is not focused on statistical methods but the climate of the past. In the present form, the paper is basically a method collection and is better suited for a different i.e., method focused journal. I thus strongly urge the authors to also add the results for the other records presented in the original study. This would not only demonstrate the viability of their method but would also further our understanding of the climate dynamics during the onsets of an GIS. Additionally, this would allow the authors to put their results into perspective when compared to the vast amount of research (other than Erhardt et al .2019) that exists on the records and mechanisms DO events which presently is sorely lacking from both discussion and conclusions. With these additions, the resulting paper would be made much more valuable to the broader paleo-climate community.'*

We thank the referee for this comment, which motivated us to substantially reconsider the way in which we present our results. We addressed this in our introductory statement.

*'Throughout the text there are a number of inaccuracies such as wrongly stated ages, confusion of fluxes and concentrations, the nature of ice core records and MCMC. Even though each on their own might seem minor, I will warn the authors that they could be interpreted as negligence. I thus strongly encourage the authors to seek out the input*

[Figure]

*of experts in ice-core records (of which there are plenty in the TiPES project) to give
the manuscript a thorough once-over to avoid potential pitfalls.'*

We will follow this advice and reach out to our collaborators in this regard. In the revised
manuscript we will correct these inaccuracies.

Specific Remarks

Figure 2: *'The vertical lines are colored blue. This is probably an accident. Please
also add the full list of references for the datasets shown in the Figure as well as the
age-scale to either the caption of the text.'*

We retrieved the data directly from the supplement to Erhardt et al. (2019) who were
the first to publish the shown data for Ca2+ and Na+ concentrations. We will also add
the original reference for the different datasets in a revised manuscript. . We would
change the color of vertical lines (and tilted connecting lines) to light gray. We would
add the time scale to those Figure captions where it is missing.

L64ff: *'What is high? Both for the statements about the record resolution as well
as about the variability choice of a relative term is only useful if it is clear what the
resolution or frequency is high in relation to. It would be much better to state at least
orders of magnitude for these instead.'*

We would add total numbers to clarify what is meant by 'high'.

Table 1: *'The caption is a bit misleading: In the data that you use in your study only
the DO events given in bold are contained. Furthermore, the statement "the stochastic,
MCMC-based method successfully detected empirical density distributions for transi-
tion onset" is inaccurate on multiple levels: To begin with the method is probabilistic,
not stochastic, secondly, the method does not detect empirical density distributions but
provides those for the transitions. Following the description of the model in Erhardt et
al., 2019, it seems likely, that the investigated transitions where chosen because they
could be described well enough with the ramp model. This seems especially likely as*

*the very short sub-events are sometimes poorly defined in the ice core record and/or often exhibit too short stable levels before or after. This is also stated in the text in multiple occasions. The ages stated in the Table are numerically identical with the ones provided in Rasmussen et al. (2014), that means that the age reference is than in fact not 1950 but the year 2000. Please check this throughout the manuscript to make sure that the correct ages are used at all times. It is also advisable to avoid the use "BP" to avoid confusion with Radiocarbon ages.'*

The reviewer is right, the caption is indeed ambiguous. We would clarify in a revision that only data from events printed in bold is further investigated in the manuscript. Further, we would change the sentence 'Bold print indicates those events for which the stochastic, MCMC-based method successfully detected empirical density distributions for transition onset.' to 'Bold print indicates those events for which application of the probabilistic MCMC-based method yields a convenient sample from the posterior probability distribution of the ramp-fit parameters. For other events, the rejection rate in the MCMC sampling procedure exceeded a critical threshold of 70

L78ff: *'Please elaborate if you extended or changed the original approach by Erhardt et al. or used it as is. Judging from the code/data availability statement the latter seems to be true. Should that be the case this needs a clear statement to distinguish prior from original work.'*

We used the ramp-fit algorithm as provided by Erhardt et al. 2019 and we will add a sentence to unambiguously clarify this.

L85: *'Consider not using the variable name here at is it only fully introduced and used much later in the manuscript.'*

A revised version of the manuscript would not comprise the computation of the probability for $n_{Ca2+}$ events to be lead by calcium.

L95: *'Please consider to not cite the pangea reference separately as it is technically*

*only a supplement to the 2019 study by Erhardt et al. Having both mentioned sepa-
rately seems contrary to the notion of a supplement.'*

We would change all references (Erhardt et al., 2018) which refer to the data stored on
the Pangea website to (Erhardt et al., 2019) which refers to the article.

L98: *'Ice core record is technically not compressed in greater depth but rather ex-
tended in the horizontal due to glacial flow which in turn leads to a thinning and has
nothing to do with compression due to hydrostatic pressure. Please use the correct
term "thinning". Deposition rates and concentrations are two fundamentally different
things, from what I gather you are using concentration records only. Please correct
"deposition rates" to concentrations.'*

Thank you, 'compression' will be replaced by 'thinning' and 'deposition rates' will be
corrected to 'concentrations'

L.106: 'stochastic' would be replaced by 'probabilistic' and accordingly everywhere in
the manuscript, when regarding the ramp-fit algorithm.

L107f: *'Because the algorithm is based in MCMC it by design (and necessity, as the
problem has no analytical solution) returns samples from the posterior distribution, not
a probability density function. The probability density functions are later only approxi-
mated using kernel density estimates.'*

We would change the sentence 'First, we introduce the stochastic transition onset de-
tection algorithm that by design returns an uncertain transition onset $t_0$ in form of a
posterior probability density distribution.' to 'First, we introduce the probabilistic transi-
tion onset detection algorithm that by design returns an uncertain transition onset $t_0$ in
form of an empirical posterior probability density distribution.'

L114ff: *'I appreciate the authors desire to advertise their approaches to a wider au-
dience. However, the provided example is completely irrelevant in the context of the
readership of this journal. Furthermore, it is somewhat contradictory because if it were*

*true, then the statistical approaches outlined in the manuscript are hopefully in fact not new but long solved in the medical context. Please find a better suited example and elaborate why the problem of inferring a population mean from an uncertain measurement is not yet solved? And if it is solved, under which assumptions is it solved and how do your assumptions differ?'*

It is true that the given example is of no relevance for the readership of CP and hence we would delete the paragraph. We were surprised ourselves that we could not find any literature on this specific issue of hypothesis testing with a sample comprised of uncertain individual measurement. We suspect that in most relevant cases, the uncertainty associated with the individuals of the sample is small compared to the uncertainty that arises from the spread of the individuals within the sample.

Figure 2: *'How is it possible, that the pdf for $\Delta t$ is unimodal if one of the pdfs for the transition onsets is bimodal? Please elaborate.'*

Computing the distribution of $\Delta t$ from the two individual distributions for the transition onsets of calcium and sodium corresponds to a convolution of the latter two. Convoluting the bimodal distribution for the calcium transition onset with a kernel as broad as the distribution for the sodium transition onset merges the two peaks of the bimodal distribution into a unimodal distribution.

L123: *'See comment above about the product of MCMC.'*

Again, we would change 'posterior probability distributions to 'empirical posterior probability distributions'.

L129: *'See comment above about the statement of failure of the MCMC algorithm. Please either remove the statement or elaborate.'*

We would introduce the notion of the rejection rate from the MCMC-sampling procedure here.

L147f: *'The investigation has basically been already performed already in the original*

*publication (see Figures A1 and A2 in Erhardt et al. 2019). Furthermore, the test data the Authors use here violates an important and explicit assumption of the algorithm: autocorrelation of the noise (i.e. an autocorrelation time larger than zero) and are thus rendering the tests invalid. I appreciate the intention of the authors here, but the oversight of the white-nose vs red-noise assumption is disappointing at best. I suggest the authors remove this section entirely.'*

In a revised manuscript we would add an investigation of how the auto-correlation of the noise influences the performance of the algorithm, thus making our performance test two-dimensional as proposed by referee 2 (L.150). The white noise used up to now corresponds to auto-correlated noise in the limit of the auto-correlation tending to zero and hence still constitutes a valid test case for the algorithm. What is shown in Erhardt et al. (2019) is not a systematic investigation of the influence of the signal to noise ratio on the quality of the returned empirical posterior probability density. First, the signal-to-noise ratio itself is not controlled but only estimated from the data in a way not further defined in the publication. Second, there is no direct way to control how the inferred empirical posterior probability density relates to the true value. In our investigation, both are guaranteed.

L184: *'Please elaborate on the statement why hierarchical distributional models cannot be invoked here and better define the term.'*

We in fact think that the problem could, in principle, be tackled with hierarchical distributional models, and plan to investigate this in future work. We'll change this accordingly.

L196: *'As a side note: Summing does not require to keep all summands in memory.'*

The sum is required for the empirical density distribution - the sum is technically not executed. In order to propagate the uncertainty inherent to the distribution in Equation (6) would indeed make it necessary to store all $6000^16$ values. We would add a clarifying comment in a revised manuscript.

L195ff: *'This paragraph makes an interesting point as the samples of $\Delta t$ for each of the transitions are interchangeable, they could technically be reshuffled to simulate more samples. Could you elaborate on the uncertainty that this sub-sampling adds to your methods and how much the results are dependent on the individual realization? If the results are not stable it might hint at the fact that the 6000 samples from what is a 16-dimensional distribution might not be enough to fully capture all of the uncertainty.'*

The uncertainty arising from subsampling is investigated in Appendix C. Table C1 provides an overview of results obtained from randomly generated alternative subsamples. The results are robust, which shows that 6000 samples are sufficient to represent the density in 16 dimensions.

L205-213: *'In this short section, the authors provide the arguably most elegant way of drawing inference on the population from the underlying data. Even though this section is a little bit hidden and Eq. (9) is not straight forward to understand, the resulting convolution of the individual posteriors for $\Delta t\,i$ provides a posterior for the average lag of the DO events. My judgment of this section being the most elegant stems from the fact that it is not dependent on any additional assumption such as the presence of an infinite number of DO events or normality of any of the distributions – It rather only answers the question which average lags are consistent with the observed 16 DO events, which is arguably the question the authors set out to answer. Comparing this to the estimates that Erhardt et al. call the "combined evidence" the difference stems from the fact that Erhardt et al. assume that all DO events exhibit an archetypical lag, i.e. one that does not vary between events or verbatim: "[. . .] this implicitly assumes that the timing differences for all interstadial onsets in the parameters investigated here are the result of the same underlying process or, in other words, are similar between the interstadial onsets" The method presented here relaxes this assumption by realizing that the averaging can be expressed as summation and thus as a convolution of the posterior densities. In comparison to the convolution described here, it is very important to note, that both the t-distribution approach as well as the bootstrap approach*

*described later aim at something subtle, yet fundamentally different: The convolution provides the probability of a mean lag given the observations. The other two however assess the distribution of this mean under their respective assumptions! I will take the liberty to encourage the authors to treat this section entirely separately from the other approaches t and to extent it a little bit to emphasize its difference to the other methods. As a side note/word of caution: The accuracy of the numerical convolution of the kernel density estimates is very much dependent on the chosen discretization and especially range of values that it is performed for. This can easily be tested when comparing distributions where the convolution is known to their numerical convolution (such as the Normal distribution). I suggest the authors do some experiments and present these in the appendix.'*

The referee correctly pointed out the differences between our approach and the one chosen by Erhardt et al. As already mentioned above, we will emphasize and clarify these differences in a revised manuscript. However, regarding the DO time lags as an outcome of a random experiment draws on the notion of a population. In this framework, it is not our main priority to compute the uncertain sample mean, but instead test whether the given sample of 16 uncertain lags contradicts a population mean equal to zero. A population mean equal to zero is identified with the absence of a systematic lags, which serves as our null hypothesis. The explanatory power of the sample mean U (termed average lag by the referee) lies in the fact that it is the best estimate (point estimate in case of a sample without uncertainty) of the population mean. Since the distributions of the population mean are not required for the line of inference proposed for a revised manuscript, we would exclude them and build the analysis solely on the hypothesis tests. We will test whether the chosen discretization has any effect on the kernel density estimates.

L215ff: *'The "refinement" that the authors present by using the definition of the t-distribution and a change in variables to estimate the mean of the underlying distribution hinges on the assumption that this is a normal distribution. Even though this*

*assumption seems inconspicuous at first sight, the authors should provide evidence that this assumption is both justified as well as not violated by the samples from the ramp-fit. Depending on the justification of the assumption or consistency with the data the results that are based on the assumptions will not be valid or should at least be interpreted with care. I suggest the authors spent some time in this section (and all other sections) to clearly (and in words) state the underlying assumptions, their implications and justifications.'*

We would not present the probability distribution of the population mean anymore in a revised manuscript.

L243ff: *'The authors try to justify their choice in the t-distribution based estimation by a presenting a bootstrapping version of the same thing. However, I do not think that this can be used to do so. Looking at the results the agreement between (μ) and bs (μ) made me wonder why that might be the case: In fact, no matter what randomly generated data the approaches are use on, the results always very closely agree with each other. This is also seemingly independent from the data being normally distributed or not. This seems slightly odd to me however the reason might be, that both methods fundamentally do the same thing: they aim to estimate the mean and the standard error of the mean from the sample and provide a distribution about mean given this standard error. To assure the reader of the validity of their approach, the authors should spend some time elaborating on this and clarify the rationale of why the bootstrap of the mean should be different than the t-distribution and what we can actually learn from having both if they yield seemingly identical results. Furthermore, the authors should add how many bootstrap samples where generated as this is an important information for the reproducibility of their study.'*

We would refrain from presenting the probability distribution for the population that is based on the bootstrapping approach. The referee seems to be right that both methods (bootstrapping and using the t-distribution) produce similar (although not the same) results regardless of the process that generated the data. The fact that the

results of the two approaches converge for large sample sizes n can be explained as follows: For large samples, the probability distribution for the sample mean tends to a Gaussian distribution with standard deviation of sigma / sqrt(n) and mean $\mu$, with sigma and $\mu$ being the standard deviation and the mean of the population regardless of the population's shape (central limit theorem). In this case, as we are bootstrapping from the cdf induced by the observed sample, $\mu$ is given by u and sigma is given by s the sample mean and standard deviation. Hence the bootstrapped distribution of sample means converges to N(u, s/sqrt(n)) for large sample sizes. The t-distribution with n-1 degrees of freedom with (z= u-$\mu$/(s/sqrt(n)) converges to the same gaussian distribution as n increases. In our case, n might still be considered small enough for the approaches to yield different results, but the uncertainties in u and s finally blur these differences. Both approaches rely on strong assumptions – the first that the original population's cdf is approximated reasonably well by the cdf induced by the sample and the second that the original population is Gaussian.

L261ff: *'After going through a lot of trouble to derive ways to estimate the distribution of the mean from uncertain observations the authors opt to throw all of this overboard and to start from scratch to come up with a way to test whether this mean is different from zero. I am a little bit puzzled why the authors present, what basically amounts to calculating a p-value for each of the MCMC samples of 16 $\Delta t\,i$ rather than investigating the distribution of the mean that they just derived. I am sure that the results would likely be not much different. It also remains unclear to me, whether this "propagation of uncertainty to the p-value" is actually a valid approach: Essentially each of the 6000 p-values that is calculated constitutes the p-value of the test of that one sample is from a distribution different to zero. The meaning of the distribution of these p-values over many repeated samples is not straight forward. This starts with the observation that for a non-significant distance the resulting p-values will be uniformly distributed, so the distribution of p-values that the authors present needs to be interpreted within that context. Though the approach the authors present might seem convenient and maybe even clever it leaves me with more questions than answers. And with this I*

*am not arguing against the conclusions the authors arrive with using this method, as anything but a non-significant lead would be surprising given the assumptions of the calculations. What is also missing from the otherwise extensive presentation is the alternative view, that the 6000 MCMC samples each present 16 observations of the mean and could be tested accordingly as 6000\*16 observations. I am sure that there is a good reason to not do this, but this alternative should at least be mentioned and discussed in the context of the other methods. I suggest the authors either rework this section entirely to better justify and clarify their approach and to include an investigation of the derived distributions of the mean or complete refrain from presenting hypothesis tests in this context. Should the authors decide to keep this section, they need to make sure to state the Null Hypothesis (and the alternative hypothesis, depending how closely they follow Fisher) correctly and explicitly.'*

The key question we want to address is whether the measured lags between the different proxy variables significantly contradict a population mean equal to (or greater than) zero. If a population mean equal to zero cannot be ruled out, a systemic lead-lag relation cannot be evidenced. For this aim, hypothesis tests constitute the convenient and scientifically well-established tool. Given that we do not present the probability distribution of the population mean, the tests will be key in the analysis. If the lags of the individual DO events were free of any uncertainty, then a single p-value could be computed for the sample (for each test). The uncertainty of the sample immediately translates into an uncertain p-value. Given that we could not find any literature on this specific problem, we propose two possible interpretations of this uncertain p-value. Under different assumptions on the shape of the population (and no assumptions in case of the bootstrap test) the corresponding tests fail to reject the null hypothesis of a population mean equal or greater than zero, for both interpretations. It is the fact that even under different assumptions we achieve the same result that makes our analysis robust. We would keep the section on hypothesis tests but would make an effort to better explain the role of these tests within the framework of our analysis.

*'This starts with the observation that for a non-significant distance the resulting p-values will be uniformly distributed, so the distribution of p-values that the authors present needs to be interpreted within that context.'*

In our case, the distribution of p-values arises from the uncertainty in the individual lag measurements within the n=16 sample. The distribution indicates the probability that the observed sample corresponds to a certain p-value with respect to the null-hypothesis. The uniform distribution of p-values mentioned by the referee indicates that the probability to realize an n-sample with some p-value p from the population is uniform, given the null hypothesis holds true. Therefore, we do not think comparing the derived p-value distributions to a uniform distribution is meaningful.

*'Essentially each of the 6000 p-values that is calculated constitutes the p-value of the test of that one sample is from a distribution different to zero.'*

We agree: The uncertain n=16 sample of DO time lags is represented by 6000 vectors in 16 dimensions (as generated by the MCMC). For each of these vectors a p-value is calculated, for the test whether this vector contradicts a population mean equal to zero. Since the 6000 vectors represent the uncertainty of the sample, the 6000 p-values represent the corresponding uncertainty of the p-value. This enables us to deduce the probability for the sample to significantly contradict a population mean equal to 0.

*'What is also missing from the otherwise extensive presentation is the alternative view, that the 6000 MCMC samples each present 16 observations of the mean and could be tested accordingly as 6000\*16 observations.'*

We are not sure if we fully understand this comment. We assume that the referee proposes to treat all 6000 * 16 values acquired from the MCMC-sampling as equally meaningful observations of the same quantity and put all of them together into one pot. As already mentioned, we would incorporate an explanation why this is not valid in a revised manuscript. Our objection to this approach draws on the following: There is no physical quantity that would be represented by such a lumped / gathered distribution.

[Figure]

Physically, 16 observations have been made for 16 different DO-events, each of which is uncertain and hence represented by 6000 MCMC-samples. Together, these 16 * 6000 values must be regarded as an empirical probability density in 16 dimensions and not in one dimension. Disregarding this would have severe consequences for further inference. For example: Assume that one tries to observe the outcome of a repeated random experiment. In the first attempt to observe it one is uncertain whether the observation was either 1 or 2. In the second runone observes 2 or 3. Gathering these possible observations together results in a set of observations 1,2,2,3 which corresponds to mean u=2 and a standard deviation of 2/3. However, equation 14 yields four possible vectors (u,s) which are (u=1.5, s=0.5), (u=2,s=2), (u=2, s=0) and again (u=1.5, s=0.5). All four vectors carry the same probability weight. From this, one may compute the expectations  = 2 and <s> = $\frac{3}{4}$ of the uncertain quantities u and s .

L396ff (the referee mistakenly wrote 496) : *'The authors observe that the distribution that results from the convolution is narrower than the one obtained by the other methods. This is interesting albeit not surprising, given the conceptual difference of the convolution to the other methods. The brief explanation that the authors give here is quite difficult to follow, could the authors elaborate?'*

As mentioned above, we would not present the probability distributions for the population mean in a revised manuscript. However, for sake of clarity: Consider a certain sample with mean u and standard deviation s. According to Equation (12) the u and s induce a probability distribution for the population mean $\mu$ centered around u. If the sample is now taken to be uncertain, uncountable many combinations (u,s) induce distributions for $\mu$ centered around the corresponding u. According to equation (15) they all contribute to the population mean distribution under uncertainty. Hence, this distribution must be broader than the distribution for the sample mean. Or the other way around, any sample mean is associated with a broad range of possible population means that define a population which potentially has generated the sample.

L406f: *'I think this point deserves a moment of attention: Despite the added layer of*

*uncertainty for the posterior distribution of U $\Delta t$ and the additional assumptions going into $\mu \Delta t$ both still put around 4/5 of the probability on lead of Ca over Na. Yes, this is not 90/100, but in IPCC parlance it is still likely that the transitions are led by a transition in Ca.'*

The referee mentions a very relevant point here. However, we disagree with the inference that the referee proposes. We have inferred a probability of  âĚŸ for the population mean to be less than zero. As far as we understand, the referee interprets this results as an indication that 'it is likely that the transitions are led by a transition in Ca'. In this sentence 'the transitions' apparently refers to all transitions. However, the population mean itself does not allow to make any statement on the probability of a single DO time lag randomly generated from the population to be less than zero - nor about the probability that the true values of individual observed time lags were in fact less than zero. In a revised manuscript we would not present the probability distribution of the population mean but we would still present the probability distribution of the sample mean. Also from the latter one cannot deduce the probability that the true values of all observed time lags were negative. We have tried to make such statements in the last part of our analysis, but the given arguments were inconsistent and would therefore not be shown in a revised manuscript.

L443ff: *'The calculation of the number of events being consistent with the lead of Ca over Na again is a very good addition to the discussion and a great extension of the order statistics shown in Fig 5 of Erhardt et al. (2019). The authors interpretation of the analysis is however somewhat strongly formulated given the large uncertainties of the estimates: The postulate that if an atmospheric circulation change (effecting only Ca) would trigger the sea ice retreat of the DO events (in turn effecting Na) than all of the events should show a lead of Ca over Na at their onset. This is not wrong but would only ever occur in a scenario where we would observe these atmospheric and sea ice changes directly and without error, not through a set of proxy records and could reasonably exclude any influence of internal climate variability. All in all, that seems to*

*comprise quite a high bar. I suggest the authors to tone down the interpretation of this otherwise very enlightening analysis.'*

After thorough review of this section we found inconsistencies in the reasoning. A stringent discussion of this issue is beyond the scope of this answer. We refrain from presenting this approach in the revised manuscript and will return to this in future work.

L468f: *'How do the authors arrive at the conclusion that the observations cannot be used to investigate the transitions with Na leading? To put it sarcastically: If that is not possible, then why is the reverse, investigating the transitions with a Ca lead?'*

Sorry, we think there's a misunderstanding here. The question is which are the specific DO events that potentially are led by sodium. In the previous paragraph we argued that the most likely configuration is one where 10 events are led by calcium while 6 are led by sodium. However, since these are all probabilistic statements, it makes no sense to indicate which specific events these are.

480ff: *'The statement on the ability of the presented results to serve as evidence is somewhat unjustified. I do agree that on the base of the presented data the Null Hypothesis of a zero or larger lead cannot be rejected but in reverse that does not mean that the same evidence cannot be used at a later stage (combined with prior knowledge and more evidence).'*

In our opinion, the chosen wording does not conflict with the demands of the referee. We state that the results 'cannot serve as evidence for atmospheric changes to trigger sea ice retreat during DO events' – and this statement does not deny that a review of the investigated data in combination with additional data or other methods will support the hypothesis of an atmospheric trigger. We will nevertheless clarify this point in a revised manuscript.

486ff: *'The possible existence of a process other than the processes that directly influence Ca or Na is an important note here and is likely the best explanation for what is*

*visible in the data. The authors could spend a little more time on this point.'*

We thank the referee for this comment. It is true that the manuscript has a strong emphasize on the methods, while the possible physical mechanisms at work during DO-events are treated only to a limited extent. We would elaborate on this aspect and include this possibility as a potential explanation of the data already in the introduction. There we would add a sentence like: 'Previous studies have found a tendency for calcium to transition before sodium. This may be interpreted as an indication for the atmosphere to trigger a change in the sea ice extent. However, it may also be that both – atmosphere and sea ice – respond to some other trigger, with the atmosphere simply responding faster. If a change in the calcium records was to be the trigger for the change in the sodium concentrations, a lag between the transitions should consistently be detected. We show that this is not the case and therefore argue that the second interpretation is the more plausible one. However, we confirm the tendency of a delayed sodium transition which we interpret as a slower reaction of the sea ice to the original trigger.'

References: Erhardt, T. et al. Decadal-scale progression of the onset of Dansgaard-Oeschger warming events. Clim. Past 15, 811–825 (2019).

---

## Author Comment (AC2) · 1 Feb 2021

General Points

We thank the referees for their very thorough and careful review. Following the constructive criticism and valuable feedback, we would like to propose several changes to the manuscript. We are convinced that these changes will substantially improve the quality and clarity of our manuscript and that they will address the referees' objections,

questions and suggestions. Since both reviews share several general and important aspects, we would first like to reply to these in a combined way here, and propose a substantial restructuring of the manuscript before we address the specific comments of this referee further below.

1. Most importantly, both referees urge us to expand the analysis to the full possible set of pairs of proxy variables to make the manuscript more appealing for CP-readers. After careful reconsideration, we agree with this criticism and would integrate the analysis of all pairs of proxies that are reported to show a clear lag-behaviour by Erhardt et al. in a revised version of the manuscript. We would not include those proxies, which Erhardt et al. find to transition simultaneously.

2. We acknowledge the comment made by referee 1, that the manuscript appears as a method collection and agree that there is a lack of guidance and motivation throughout the method section in the current version of the manuscript. In particular, both referee reports indicate that the role of the hypothesis tests hasn't been made sufficiently clear. In response to this, we would restructure the manuscript and focus on the key question the manuscript aims to answer: Can we rule out that the lag-tendency observed from a sample of 16 DO events arises by chance, with an underlying population mean of the lag equal to zero? This question can be directly answered in terms of the extended significance tests we carry out, and we therefore plan to put considerably more focus on this in a revised manuscript.

In the revised manuscript, we would then take the following, natural line of inference that should be immediately obvious to the reader:

1. Establishment of the statistical framework in terms of random experiments: The DO transition onset lags constitute an n=16 sample (for other proxy-pairs this number can deviate) drawn from an underlying population. Here, we would in-

tegrate a detailed comparison with the assumptions made by Erhardt et al. to derive what they call 'combined evidence', as requested by both referees.

2. Introduction of uncertainty propagation in the statistical inference in the case of uncertain samples.

3. Testing whether or not the observed sample contradicts the null-hypothesis of a population mean equal to (or greater than) zero, given the uncertainty of the sample. A population mean significantly different from zero can be interpreted as a systematic lag. Here, 'systematic' does not directly imply a causal relation but is a necessary condition for the inference of causality.

4. Comparison of the uncertain sample mean with the combined evidence reported by Erhardt et al. We would introduce this streamlined methodological framework in the 'Methods' and show the results of its application to all proxy-pairs under study in the 'Results' section.

We would like to propose to remove the following from our manuscript in order to increase its stringency and readability:

- Remove the derivation of distributions for the population mean, because it is not required to answer the stated question.

- To simplify the presentation, if you agree, we would refrain from carrying out all the derivations for empirical densities and use continuous probability density functions instead. The equivalence between empirical densities (as provided by the MCMC) and continuous probability density functions in all our derivations would be shown in the appendix. While the formulation in terms of empirical densities has not explicitly been criticized by the referees, we noticed that it does not contribute to the understanding of the physical and statistical reasoning of the manuscript.

- Remove the discussion on the probability for n events to be lead by calcium. Originally, we used this derivation to reject a causal relation, but the objection of referee 2 made us reconsider our method and we found argumentative inconsistencies.

These changes will result in a streamlined version of the manuscript and make it significantly easier for the reader to follow the main line of thought without getting distracted by a lot of technical details. We think that tightening of the methods together with the incorporation of the additional proxy pairs will yield a convenient balance for publication in CP.

As both referees proposed to treat all 16 * 6000 (6000 MCMC-samples for each of the 16 DO events) values as equal observations of the same quantity and gather all values to one single empirical probability distribution, we would add a section to explains why this cannot be done (for an explanation see General comment 2 of review 2 and similarly L.243 of review 1).

**Point by point answer to referee 2:**

General Comments

1. *'The authors do not specifically discuss how their approach is different to Erhardt et al 2019. While the manuscript gives the impression that previous studies completely disregarded uncertainties and used expectation values, this is not true for Erhardt et al, who included the uncertainties in a rigorous manner in their own right. The difference lies in the interpretation of the estimated onsets as random variables. Erhardt et al consider the uncertain samples as measurements of the same fixed quantity (a fixed time lag of Ca and Na at DO onsets), which allows them to simply multiply the individual MCMC posteriors to obtain a single posterior distribution that represents the measurement uncertainty of the fixed time lag. In contrast, the present study interprets the time lag to be a random variable, varying in between DO events. Thus, they cannot*

*multiply the individual posteriors. In the discussion, the authors contrast their approach with the results one would obtain when completely discarding the uncertainties in the onset determination. It is not very surprising that including uncertainty yields hypothesis tests which are no longer significant. Instead, it would be more relevant to highlight the contrasting results of their approach to Erhardt et al.'*

We agree that this should have been described more clearly and would discuss the differences in the underlying assumptions explicitly in the Methods section of a revised manuscript. Additionally, in Figure 4 we would include the 'combined evidence' according to Erhardt et al. instead of the probability distribution of the population mean, and provide a detailed explanation of the differences. While Erhardt et al. assume that the time lag between the calcium and the sodium transition was a physical constant, such as the speed of light, we argue that climate variability will cause this time lag to differ from event to event. Assuming that the DO mechanism remains unchanged between events, it is very convenient to treat the 16 observed time lags as the result of a random experiment that was performed 16 times. Even if the results of the hypothesis tests were not surprising given the uncertainties in the observations as the referee states, they still contradict the conclusions of Erhardt et al. who state: 'They [probability density estimates] clearly show that, on average, both the reduction in terrestrial aerosol concentration and the increase in annual layer thickness precede the reduction in sea-salt aerosol for all stages of the transition'. Our tests indicate that one cannot differentiate the observed tendency for a sodium lag from the result of a random experiment with population mean equal to zero. In this way, our results from the hypothesis tests add valuable information to the debate on the succession of events at DO onsets.

2. *'The authors speak of a rigorous propagation of uncertainties to p-values, among other things. To achieve this, they introduce probability distributions of the mean and other test statistics, which are formally represented using delta distributions arising from the empirical sample. In practice, these distributions are all computed by sum-*

*ming a random sample from the individual MCMC posteriors, which is essentially boot-strapping if I understand correctly. Can the authors comment on how the results of this work would be different if they simply summed the individual MCMC posteriors (which would be equivalent to bootstrapping as well), and then looked at the arising distribution of the lag, determining the probability of a lag >=0? This would be much simpler and the obvious alternative approach to the multiplication of the posteriors by Erhardt et al. It would probably also give a non-significant result regarding a Ca2+ lead.'*

We are not sure if we fully understand this comment. The referee proposes to 'sum the individual MCMC posteriors and look at the arising distribution of the lag'. We interpret this as follows: For each DO-event the MCMC samples 6000 lags from the corresponding posterior distribution. The referee proposes to put all these 16*6000 values together in one pot and regard them as observations of the same quantity. Our objection to this approach draws on the following: There is no physical quantity that would be represented by such a lumped / gathered distribution. Physically, 16 observations have been made for 16 different DO-events, each of which is uncertain and hence represented by 6000 MCMC-samples. Together, these 16 * 6000 values must be regarded as an empirical probability density in 16 dimensions and not in one dimension. Disregarding this would have severe consequences for further inference. For example: Assume that one tries to observe the outcome of a repeated random experiment. In the first attempt to observe it one is uncertain whether the observation was either 1 or 2. In the second runone observes 2 or 3. Gathering these possible observations together results in a set of observations 1,2,2,3 which corresponds to mean u=2 and a standard deviation of 2/3. However, equation 14 yields four possible vectors (u,s) which are (u=1.5, s=0.5), (u=2,s=2), (u=2, s=0) and again (u=1.5, s=0.5). All four vectors carry the same probability weight. From this, one may compute the expectations  = 2 and <s> = $\frac{3}{4}$ of the uncertain quantities u and s . The referee states: 'In practice, these distributions are all computed by summing a random sample from the individual MCMC posteriors, which is essentially bootstrapping if I understand correctly.' Here, as well we are not sure if we understand this statement correctly. In our

understanding a random sample from the posterior lag distribution for a single DO event corresponds to the set of 6000 values, sampled by the MCMC from the corresponding posterior probability distribution. The empirical density distribution that is induced by this MCMC-sample approximates the true posterior distribution for the DO-event in the sense of equation (3). The MCMC-sampling procedure that gives rise to the sample of 6000 values is not equivalent to bootstrapping. The MCMC algorithm is used to sample from a continuous probability density, because the distribution is too complex to use it directly for inference. Bootstrapping is used to generate synthetic samples from a finite size sample by 'drawing with replacement', not to sample from a continuous probability distribution.

3. *'I am wondering in what way the hypothesis tests introduced in Sec. 3.6-3.7 are necessary and add to the results? When reading the manuscript, I was surprised that hypothesis tests were introduced after distributions for the sample or population mean had already been given, which would directly allow to test the hypothesis of a mean >=0?'*

We hope this was clarified in the explanation of our proposed restructuring. In brief, we think that the hypothesis tests provide the most straightforward way to answer the question whether the observed sample of lags is inconsistent with an underlying population mean lag larger than or equal to 0. Our results show that, under consideration of the uncertainties present, it is not inconsistent with that null hypothesis, which adds an important layer of information to the debate on the trigger of DO events.

4. *'Why did the authors restrict the analysis to Na and Ca, and omitted an analysis of the offsets of d18O and lambda? This would be an important consistency test, and might even be more relevant since lambda has a very direct meaning as a proxy, and d18O is the most important of all proxies.'*

This point was addressed in our first statement, we'll add the other variables in the revised manuscript

Specific comments:

L15ff: *'I think the conclusions should be stated differently. In my interpretation, the analysis does not contradict a lead or lag. Rather, as a result of the large uncertainties in estimating the individual onset timings, it cannot be excluded that there are in fact no leads or lags. Similarly, on the grounds of the uncertainties in the individual event timings, there is not enough evidence to conclude that atmospheric reorganization systematically preceded sea ice retreat for all events.'*

We agree and would adjust the abstract in the sense of the comment by the referee.

L34-35:*'The interstadials lasted up to 10 millennia, with 1.5 millennia being the average.'*

Thank you, we would change this to: 'The abrupt warming is followed by gradual cooling over centuries to millennia, before the climate abruptly transitions back to cold conditions'.

L130: *'Why does the transition detection fail for some events?'*

The study only takes into account those events for which the rejection rate during the MCMC-sampling process was below 70

L150: *'How would this depend on the autocorrelation of the noise? The Na and Ca records might have different noise structure and thus there is the potential for systematic biases indeed.'*

We must admit that we did not consider this. We would add the same investigation as carried out for the influence of the noise amplitude for the correlation coefficient of the noise. This would also meet the demands formulated by referee 1 regarding this issue.

Eq. 7: *'Maybe the authors can elaborate specifically in the text what this approximation does. Since the order of MCMC samples for each event is arbitrary, by associating the*

*i-th MCMC sample for every event, it seems like it is just a random sampling of m=6000 points in the joint space. If this is the case, why not simply sample randomly in the first place, and why not choose many more than m=6000 points? Or is this rather done in order to simplify the notation?'*

The referee is correct. Associating the i-th individual member of every event's MCMC-sample is a random choice and is indeed the simplest in terms of notation. The analysis in Appendix C shows that 6000 vectors from the 16 dimensional probability density distribution suffice to capture the distributions features. Randomizing the choice of these vectors does not influence the results. Note that this would not appear in the manuscript's main text as we would refrain from using the empirical density notation in the revised methods section.

L278ff: *'I am not sure why the authors say that they are only given "relative" data, since they estimate the onsets in the two proxies. Furthermore, since until this point the data was already given exclusively as onset timing differences, I wonder why it is necessary to introduce "paired samples" now? This is a bit confusing to the reader.'*

We agree that the sentence is not very helpful and would delete it. In fact, the lags constitute relative data in the sense that there is no absolute time for the onset of the DO event. The timings of calcium onset and the sodium onset only receive meaning when compared to each other. Further, a measurement of the timing of the $Ca2+$ transition from the i-th DO event cannot be compared to the timing of the $Na+$ transition of the j-th DO event, that's why the data is paired.

L348: *'I might have missed this somewhere, but how is the distribution of the test statistic under the null hypothesis constructed?'*

The distribution of the test statistic under the null-hypothesis has a very lengthy expression and therefore was not explicitly given. It is true that Eq.(29) is missing some explanation here. We would add the functional form of $P(w'_i)$.

L396: *'Could the authors explain more explicitly why the sample and population mean distributions are so different, and what this means for the interpretation of the results? Which distribution should be preferred?'*

Note that we would not show the population mean's distribution anymore in the revised manuscript.. For sake of clarity, we nevertheless respond here: Assuming that all DO events followed the same physical mechanism, we regard the observed time lags as the outcome of a repeated random experiment with the randomness being due to climate variability. A random experiment is fully characterized by its population. The population mean indicates systematic leads or lags and hence is the decisive quantity. The explanatory power of the sample mean U (termed average lag by the referee) is limited to the fact that it is the best estimate (point estimate in case of a sample without uncertainty) of the population mean; conclusions based on the sample lag hence need to be accompanied with information on how likely it would be to obtain this sample mean from a population characterized by a null hypothesis To give a simple example: If you want to judge whether a coin is biased, it is not sufficient to toss the coin 16 times (with head assigned a value of 0 and tail assigned 1) and report on a mean of, say, 0.3, arguing that it is different from 0.5, the expected outcome if the coin is unbiased. Rather, one would have to compute the probability of obtaining a mean of 0.3 or smaller under the assumption that the coin is unbiased. If this probability is small, one can (at a reported significance level) argue that the coin is biased. Also for the reasons given in the introductory statement, we hence prefer applying hypothesis tests to the uncertain sample over deriving the distribution for the population mean. The population mean distribution must necessarily be broader than the sample mean distribution. If the observation was free of any uncertainty, the sample mean would be a scalar. The t-distribution would induce a probability distribution for the potential population mean around this sample mean. Now, with the sample mean being uncertain itself, the certain sample mean probability distribution for the population is convoluted with the sample mean distribution and the convolution of two functions is always broader than either of the functions being convoluted with one another.

L441ff: *'Here the authors introduce another simple method to address the likelihood of a systematic Ca2+ lead. I think it would be more coherent if this were moved to the Results Section. Furthermore, I find the conclusions from this simple calculation too confident, and in spirit contradictory to the interpretation of the other results. The probability of obtaining exactly n=16 events with a Ca lead will always be very small when there is a relatively large measurement uncertainty of the individual lags (spanning both positive and negative values), even if all individual posteriors would be clearly centered at negative values. Just like the probability of flipping 16 out of 16 heads is still very low for a strongly biased coin. This does not allow one to contradict the hypothesis that all events would follow the same pattern with a Ca2+ lead. For the data at hand, all but two events show posterior distributions centered at a negative value. Maybe the authors could write instead that from the MCMC posteriors it seems unlikely that all DO events occurred with a preceding abrupt change in Ca2+. However, this might merely reflect the fact that due to the large uncertainty in estimating the onsets, the individual MCMC posteriors have significant support for positive lags as well. Arguing like this would also be much more in line with the authors' earlier statements that they cannot infer an absence of causality from their non-significant tests.'*

The comment of the referee made us review the reasoning invoked in this section. In fact, we find inconsistencies and would therefore not present this at all. We will try to reconcile the inconsistencies and derive further statements on a potential causal relationship between the two transitions at a later stage.

Figure 1 and Table 1: *'Just to be sure, can the authors confirm that the time scale they use really is years BP (before the year 1950 AD), and not years b2k (before the year 2000), which is what is commonly used in GICC05?'*

In Figure 1 we used the data provided by Erhardt et al. 2019 who use BP (before 1950) as age reference. This is the only data source we are aware of that provides d18O data with 10y resolution. In Table 1, however, our indication is fact wrong. These are ages

in b2k (before 2000).

L18: *'...holds true, the we conclude...'*

Will be corrected.

L98: *'Instead of "compression" rather say: ...due to the thinning of the annual layers in the core.'*

Will be corrected.

Figure 3: *'Maybe it would be good to choose a different color for the null hypothesis in panel b. Otherwise it gives the impression that it corresponds in some way to the blue distribution in panel a.'*

This is true and will be adjusted.

Eq. 14 and Eq. 16: *'I am wondering whether "u" in the second delta function should be replaced by the empirical mean $\sum(y_i)/n$?'*

Mathematically this should be the same, since the first 'delta function' forces $\sum(y_i)/n$ to assume the value u. However, we would change this, since the formulation including $\sum(y_i)/n$ seems more instructive.

L235: *'Maybe "marginal distribution" would correspond better to the nature of this distribution?'*

In view of the proposed changes, this is obsolete. Here the term 'expected' was chosen on purpose, since we consider the distribution $\rho(\mu)$ as a function of the two variables u and s. Since u and s are uncertain, we can compute an expectation of $\rho(\mu, u, s)$ by averaging over (u,s) with the weight $\rho(u, s)$.

Eq. 17: *'What is the notation $u_j$ and $s_j$?'*

They are missing an $^{emp}$ – definition is then given in the previous line.

Eq. 20: *'I am unfamiliar with this definition of a p-value. Is the integration not simply over all $\phi < \phi_0$ (for a one-sided test)? The definition here could lead to rather strange results for very asymmetric and long-tailed distributions.'*

The definition corresponds to the 2-sided test. Equation (20) is the mathematical formulation of what the text states: 'Given a statistic of a certain sample realization $\phi(x-0) = \phi_0$ , the p-value is indicates the cumulative probability for obtaining a more extreme $\phi_1(X_1)$ from a second sample $X_1$ , provided $H_0$ was true.'

L387:*' missing delta in the sum.'*

Will be corrected.

Figure 4: *'Larger fonts and panels would be nice for better visibility.'*

We will adjust the figure to improve readability.

Eq. 35: *'It would be good if the authors could point to where the individual probabilities in the product come from.'*

Given the already extensive mathematical formulation of our methods, we tried to keep it short at this point. Given the comment we would now add more explanation to this formula.

---

## Author Response (AR1)

**General Points**

We thank the referees for their very thorough and careful review. Following the constructive criticism and valuable feedback, we substantially changed the manuscript. We are convinced that these changes improved the quality and clarity of our manuscript and that they address the referees' objections, questions and suggestions. We do not provide a mark-up version of the changes, because there apart from the introduction all sections have been rewritten at least in parts.

Since both reviews share several general and important aspects, we would first like to elaborate on the changes that have been implemented in response to these general points in a combined way here, before we address the specific comments of this referee further below.

1. Most importantly, both referees urge us to expand the analysis to the full possible set of pairs of proxy variables to make the manuscript more appealing for CP-readers. After careful reconsideration, we agreed with this criticism and integrated the analysis of all pairs of proxies that are reported to show a clear lag-behaviour by Erhardt et al. in the revised version of the manuscript. We did not include those proxies, which Erhardt et al. find to transition simultaneously.

2. We acknowledge the comment made by referee 1, that the manuscript appears as a method collection and agree that there was a lack of guidance and motivation throughout the method section in the original version of the manuscript. In particular, both referee reports indicate that the role of the hypothesis tests hasn't been made sufficiently clear. In response to this, we would restructured the manuscript and focussed on the key question the manuscript aims to answer: Can we attribute the observed lag-tendencies to a systematic mechanism or can we not distinguish them from randomly occurred lag-tendencies? This question is directly answered in terms of the extended significance tests. Hence, we left aside all additional statistical considerations presented in the original manuscript and put the focus solely on the hypothesis tests in the presence of uncertainty.

In the revised manuscript, we now take the following, natural line of inference that should be immediately obvious to the reader:
1. Establishment of the statistical framework in terms of random experiments: The DO transition onset lags constitute samples drawn from underlying populations.
2. Introduction of uncertainty propagation in the statistical inference in the case of uncertain samples.
3. testing whether or not the observed sample contradicts the null-hypothesis of a population mean equal to (or greater than) zero, given the uncertainty of the sample. A population mean significantly different from zero can be interpreted as a systematic lag.
4. detailed comparison with the statistical perspective assumed by Erhardt et al. as requested by both referees.

We now introduce this streamlined methodological framework in the 'Methods' and show the results of its application to all proxy-pairs under study in the 'Results' section.

We removed the following from our manuscript in order to increase its stringency and readability:
- derivation of distributions for the population mean, because it is not required to answer the stated key question.

- we refrained from carrying out all the derivations for empirical densities and used continuous probability density functions instead. The equivalence between empirical densities (as provided by the MCMC) and continuous probability density functions in all our derivations is now shown in the appendix. While the formulation in terms of empirical densities has not explicitly been criticized by the referees, we noticed that it does not contribute to the understanding of the physical and statistical reasoning of the manuscript.
- We removed the discussion on the probability for n events to be lead by calcium. Originally, we used this derivation to reject a causal relation, but the objection of referee 2 made us reconsider our method and we found argumentative inconsistencies.

These changes result in a streamlined version of the manuscript and make it significantly easier for the reader to follow the main line of thought without getting distracted by a lot of technical details. We think that tightening of the methods together with the incorporation of the additional proxy pairs now yields a convenient balance for publication in CP.

As both referees proposed to treat all 16 * 6000 (6000 MCMC-samples for each of the 16 DO events) values as equal observations of the same quantity and gather all values to one single empirical probability distribution, we would added a paragraph in the discussion section that explains why this cannot be done (l.571).

**Point by point answer to referee 1:**

In their manuscript Riechers and Boers present a method for the statistical analysis of the results presented in Erhardt et al. (2019) with the goal to determine the average phasing of the onset of Greenland Interstadials (GIS). To do so, they present three different approaches with the same goal: To account for between-event variability. This additional layer of variability, as pointed out by the Authors (as well as more implicitly by Erhardt et al., 2019), was assumed to be non-existent to derive the averaged estimates in the original study. The exemplary application of their method highlights the importance of accounting for this additional layer of variability and underlines the difficulties of investigating changes in a very tightly coupled system under uncertainty. Overall, the paper is very well structured and written and the presented methods provide an interesting toolset for a range of questions. The derivations of the methods are presented very thoroughly, and all necessary technical information is present. For the broader paleo-climate community (which is the audience of CP) the method descriptions could be supplemented with more explanations to aid intuition and prolonged use of the innovative approaches.

We thank the referee for this positive feedback. The clear focus on the main line of reasoning in the revised manuscript will facilitate the readers' understanding.

Unfortunately, and this is my biggest concern, the authors focus the manuscript entirely on their methods and only provide one brief albeit very intriguing example at the end. This is exceptionally disappointing as the analysis could easily be extended to the other records presented in Erhardt et al (2019), especially given the simple calculations needed for the presented methods. The answer of the authors to this concern will likely be that this analysis will follow in a later paper. However, this raises the question whether the manuscript in its current form qualifies for a journal that is not focused on statistical methods but the climate of the past. In the present form, the paper is basically a method collection and is better suited for a different i.e., method focused journal. I thus strongly urge the authors to also add the results for the other records presented in the original study. This would not only

demonstrate the viability of their method but would also further our understanding of the climate dynamics during the onsets of an GIS. Additionally, this would allow the authors to put their results into perspective when compared to the vast amount of research (other than Erhardt et al .2019) that exists on the records and mechanisms DO events which presently is sorely lacking from both discussion and conclusions. With these additions, the resulting paper would be made much more valuable to the broader paleo-climate community.

We thank the referee for this comment, which motivated us to substantially reconsider the way in which we present our results. We addressed this in our introductory statement.

Throughout the text there are a number of inaccuracies such as wrongly stated ages, confusion of fluxes and concentrations, the nature of ice core records and MCMC. Even though each on their own might seem minor, I will warn the authors that they could be interpreted as negligence. I thus strongly encourage the authors to seek out the input of experts in ice-core records (of which there are plenty in the TiPES project) to give the manuscript a thorough once-over to avoid potential pitfalls.

We paid special attention to accurately handle dates and units of the climate proxies under study. Also, we carefully reviewed that all quantities are correctly indicated.

Figure 2: The vertical lines are colored blue. This is probably an accident. Please also add the full list of references for the datasets shown in the Figure as well as the age-scale to either the caption of the text.

Figure1: We retrieved the data directly from the supplement to Erhardt et al. (2019) who were the first to publish the shown data for Ca2+ and Na+ concentrations. We also added the original reference for the d18O data and the GICC05 age scale in the revised manuscript. We changed the color of vertical lines (and tilted connecting lines) to light gray. We added the time scale to those Figure captions where it was missing. The time scale in Fig1. is indeed BP (before 1950), the ages of the DO events as indicated by Rasmussen (2014) have been converted accordingly.

L64ff: What is high? Both for the statements about the record resolution as well as about the variability choice of a relative term is only useful if it is clear what the resolution or frequency is high in relation to. It would be much better to state at least orders of magnitude for these instead.

l.73: We added total numbers (7 years of higher) or clarify what is meant by 'high'.

Table 1: The caption is a bit misleading: In the data that you use in your study only the DO events given in bold are contained. Furthermore, the statement "the stochastic, MCMC-based method successfully detected empirical density distributions for transition onset" is inaccurate on multiple levels: To begin with the method is probabilistic, not stochastic, secondly, the method does not detect empirical density distributions but provides those for the transitions. Following the description of the model in Erhardt et al., 2019, it seems likely, that the investigated transitions where chosen because they could be described well enough with the ramp model. This seems especially likely as the very short sub-events are sometimes poorly defined in the ice core record and/or often exhibit too short stable levels before or after. This is also stated in the text in multiple occasions. The ages stated in the Table are numerically identical with the ones provided in Rasmussen et al. (2014), that means that the age reference is than in fact not 1950 but the year 2000. Please check this throughout the manuscript to make sure that the correct ages are used at all times. It is also advisable to avoid the use "BP" to avoid confusion with Radiocarbon ages.

Tab.1 was removed from the manuscript. Fig. 2 in the revised manuscript now takes a similar role. The caption for Fig. 2 was rewritten entirely to avoid the ambiguity pointed out be the referee.

L78ff: Please elaborate if you extended or changed the original approach by Erhardt et al. or used it as is. Judging from the code/data availability statement the latter seems to be true. Should that be the case this needs a clear statement to distinguish prior from original work.

We used the ramp-fit algorithm as provided by Erhardt et al. 2019. This is now clarified in l.72.

('We use the same data and the same probabilistic transition onset detection method as provided by Erhardt et al. (2019).')

L85: Consider not using the variable name here at is it only fully introduced and used much later in the manuscript.

The revised version of the manuscript does not comprise the computation of the probability for $n_{\{Ca2+\}}$ events to be lead by calcium.

L95: Please consider to not cite the pangea reference separately as it is technically only a supplement to the 2019 study by Erhardt et al. Having both mentioned separately seems contrary to the notion of a supplement.

We changed all references (Erhardt et al., 2018) which refer to the data stored on the Pangea website to (Erhardt et al., 2019) which refers to the article in the main text. In the data availability statement we still cite the Pangea reference but we are absolutely willing to change this, if requested.

L98: Ice core record is technically not compressed in greater depth but rather extended in the horizontal due to glacial flow which in turn leads to a thinning and has nothing to do with compression due to hydrostatic pressure. Please use the correct term "thinning". Deposition rates and concentrations are two fundamentally different things, from what I gather you are using concentration records only. Please correct "deposition rates" to concentrations.

Thank you, 'compression' was replaced by 'thinning' and 'deposition rates' were corrected to 'concentrations'.

L107f: Because the algorithm is based in MCMC it by design (and necessity, as the problem has no analytical solution) returns samples from the posterior distribution, not a probability density function. The probability density functions are later only approximated using kernel density estimates.

We introduce the transition onset detection method designed by Erhardt et al. under 3.1 (l.149). In our understanding, the formulation of a linear ramp fit is part of this method and the MCMC is used to handle the posterior probability distributions, which are induced by the stochastic linear ramp model.

We hope this is conveniently explained in Sec. 3.1 and in Appendix B, accordingly.

L114ff: I appreciate the authors desire to advertise their approaches to a wider audience. However, the provided example is completely irrelevant in the context of the readership of this journal. Furthermore, it is somewhat contradictory because if it were true, then the statistical approaches outlined in the manuscript are hopefully in fact not new but long solved in the medical context. Please find a better

suited example and elaborate why the problem of inferring a population mean from an uncertain measurement is not yet solved? And if it is solved, under which assumptions is it solved and how do your assumptions differ?

It is true that the given example is of no relevance for the readership of CP and hence we removed the paragraph. We were surprised ourselves that we could not find any literature on this specific issue of hypothesis testing with a sample comprised of uncertain individual measurement. We suspect that in most relevant cases, the uncertainty associated with the individuals of the sample is small compared to the uncertainty that arises from the spread of the individuals within the sample.

Figure 2: How is it possible, that the pdf for Δt is unimodal if one of the pdfs for the transition onsets is bimodal? Please elaborate.

Computing the distribution of \Delta t from the two individual distributions for the transition onsets of calcium and sodium corresponds to a convolution of the latter two. Convoluting the bimodal distribution for the calcium transition onset with a kernel as broad as the distribution for the sodium transition onset merges the two peaks of the bimodal distribution into a unimodal distribution.

L123: See comment above about the product of MCMC.

This was clarified in Sec. 3.1 and in particular in l. 191 where we explain, that throughout the main text we refer to continuous probability densities instead of empirical probability densities. The derivation of the methods in terms of empirical densities as induced by the MCMC sampler is given in Appendix A.

L129: See comment above about the statement of failure of the MCMC algorithm. Please either remove the statement or elaborate.

For sake of comparability we included the same DO events previously investigated by Erhardt et al. - all of these events are successfully fitted by the ramp-fit method. We removed the comment on the success criterion of the MCMC sampler, but included an explanation on the data selection (l.129)

L147f: The investigation has basically been already performed already in the original publication (see Figures A1 and A2 in Erhardt et al. 2019). Furthermore, the test data the Authors use here violates an important and explicit assumption of the algorithm: autocorrelation of the noise (i.e. an autocorrelation time larger than zero) and are thus rendering the tests invalid. I appreciate the intention of the authors here, but the oversight of the white-nose vs red-noise assumption is disappointing at best. I suggest the authors remove this section entirely.

We revisited the robustness test carried out by Erhardt et al. and agree with the referee, that this legitimates the application of the Bayesian transition detection method. Furthermore, the focus of this study is the application of hypothesis tests to uncertain data samples. Hence, we removed all considerations regarding the performance of the ramp-fit method and rely on the assessment provided by Erhardt et al.

L184: Please elaborate on the statement why hierarchical distributional models cannot be invoked here and better define the term.

After reconsideration, we believe that hierarchical models could in fact be applied to the data set. We corrected the manuscript accordingly (l.601).

L196: As a side note: Summing does not require to keep all summands in memory.

Working with empirical density distributions requires to store all values comprised in the representative

set of the distribution. The sum is not executed in the sense, that is returns a single value. The computational problems are now explained in more detail in Appendix A and B.

L195ff: This paragraph makes an interesting point as the samples of Δt for each of the transitions are interchangeable, they could technically be reshuffled to simulate more samples. Could you elaborate on the uncertainty that this sub-sampling adds to your methods and how much the results are dependent on the individual realization? If the results are not stable it might hint at the fact that the 6000 samples from what is a 16-dimensional distribution might not be enough to fully capture all of the uncertainty.

The uncertainty arising from subsampling is investigated in Appendix B. Tables B1 and B2 provide an overview of results obtained from randomly generated alternative subsamples. The results are robust, which shows that 6000 samples are sufficient to represent the respective densities.

L205-213: In this short section, the authors provide the arguably most elegant way of drawing inference on the population from the underlying data. Even though this section is a little bit hidden and Eq. (9) is not straight forward to understand, the resulting convolution of the individual posteriors for Δt i provides a posterior for the average lag of the DO events. My judgment of this section being the most elegant stems from the fact that it is not dependent on any additional assumption such as the presence of an infinite number of DO events or normality of any of the distributions – It rather only answers the question which average lags are consistent with the observed 16 DO events, which is arguably the question the authors set out to answer. Comparing this to the estimates that Erhardt et al. call the "combined evidence" the difference stems from the fact that Erhardt et al. assume that all DO events exhibit an archetypical lag, i.e. one that does not vary between events or verbatim: "[. . .] this implicitly assumes that the timing differences for all interstadial onsets in the parameters investigated here are the result of the same underlying process or, in other words, are similar between the interstadial onsets" The method presented here relaxes this assumption by realizing that the averaging can be expressed as summation and thus as a convolution of the posterior densities. In comparison to the convolution described here, it is very important to note, that both the t-distribution approach as well as the bootstrap approach described later aim at something subtle, yet fundamentally different: The convolution provides the probability of a mean lag given the observations. The other two however assess the distribution of this mean under their respective assumptions! I will take the liberty to encourage the authors to treat this section entirely separately from the other approaches t and to extent it a little bit to emphasize its difference to the other methods. As a side note/word of caution: The accuracy of the numerical convolution of the kernel density estimates is very much dependent on the chosen discretization and especially range of values that it is performed for. This can easily be tested when comparing distributions where the convolution is known to their numerical convolution (such as the Normal distribution). I suggest the authors do some experiments and present these in the appendix.

The referee correctly pointed out the differences between our approach and the one chosen by Erhardt et al. As already mentioned above, we emphasized and clarify these differences in the revised manuscript in Sec. 3.4 and in the discussion.

Also, the referee is correct in the sense that the computation of the uncertain sample mean (termed average lag by the referee) adds valuable information without invoking any additional assumptions. Additionally, the study aims to judge whether or not this sample mean can be attributed to a systematic mechanism. Therefore, the hypothesis testing is key to the revised manuscript.

The other methods have been removed from the manuscript to improve the stringency of the argumentation.

L215ff: The "refinement" that the authors present by using the definition of the t-distribution and a change in variables to estimate the mean of the underlying distribution hinges on the assumption that this is a normal distribution. Even though this assumption seems inconspicuous at first sight, the authors should provide evidence that this assumption is both justified as well as not violated by the samples from the ramp-fit. Depending on the justification of the assumption or consistency with the data the results that are based on the assumptions will not be valid or should at least be interpreted with care. I suggest the authors spent some time in this section (and all other sections) to clearly (and in words) state the underlying assumptions, their implications and justifications.

We do not present the probability distribution of the population mean anymore in the revised manuscript.

L243ff: The authors try to justify their choice in the t-distribution based estimation by a presenting a bootstrapping version of the same thing. However, I do not think that this can be used to do so. Looking at the results the agreement between ρ(μ) and ρ bs (μ) made me wonder why that might be the case: In fact, no matter what randomly generated data the approaches are use on, the results always very closely agree with each other. This is also seemingly independent from the data being normally distributed or not. This seems slightly odd to me however the reason might be, that both methods fundamentally do the same thing: they aim to estimate the mean and the standard error of the mean from the sample and provide a distribution about mean given this standard error. To assure the reader of the validity of their approach, the authors should spend some time elaborating on this and clarify the rationale of why the bootstrap of the mean should be different than the t-distribution and what we can actually learn from having both if they yield seemingly identical results. Furthermore, the authors should add how many bootstrap samples where generated as this is an important information for the reproducibility of their study.

Since the revised manuscript focuses on the hypothesis test, we refrained from presenting the probability distribution for the population that is based on the bootstrapping approach. The referee seems to be right that both methods (bootstrapping and using the t-distribution) produce similar (although not the same) results regardless of the process that generated the data. This is – at least partially – due to the central limit theorem.

L261ff: After going through a lot of trouble to derive ways to estimate the distribution of the mean from uncertain observations the authors opt to throw all of this overboard and to start from scratch to come up with a way to test whether this mean is different from zero. I am a little bit puzzled why the authors present, what basically amounts to calculating a p-value for each of the MCMC samples of 16 Δt i rather than investigating the distribution of the mean that they just derived. I am sure that the results would likely be not much different. It also remains unclear to me, whether this "propagation of uncertainty to the p-value" is actually a valid approach: Essentially each of the 6000 p-values that is calculated constitutes the p-value of the test of that one sample is from a distribution different to zero. The meaning of the distribution of these p-values over many repeated samples is not straight forward. This starts with the observation that for a non-significant distance the resulting p-values will be uniformly distributed, so the distribution of p-values that the authors present needs to be interpreted within that context. Though the approach the authors present might seem convenient and maybe even clever it leaves me with more questions than answers. And with this I am not arguing against the conclusions the authors arrive with using this method, as anything but a non-significant lead would be surprising given the assumptions of the calculations. What is also missing from the otherwise extensive presentation is the alternative view, that the 6000 MCMC samples each present 16 observations of the

mean and could be tested accordingly as 6000*16 observations. I am sure that there is a good reason to not do this, but this alternative should at least be mentioned and discussed in the context of the other methods. I suggest the authors either rework this section entirely to better justify and clarify their approach and to include an investigation of the derived distributions of the mean or complete refrain from presenting hypothesis tests in this context. Should the authors decide to keep this section, they need to make sure to state the Null Hypothesis (and the alternative hypothesis, depending how closely they follow Fisher) correctly and explicitly.

The starting point of this study was the observation by Erhadt et al, that transitions in Ca2+ and the annual layer thickness start on average about one decade earlier than their counterparts in Na+ and d18O. We set out to answer the question, if this observed lag can be attributed to an underlying mechanism or whether it may have arisen simply by chance. For this aim, we regard the observed lags (for a given pair of proxies) as observations of independent, identically distributed random variables that have been generated in series of a repeated random experiment. As a next step, we identify the potential systematic mechanism with a bias in the population of this experiment, which finally allows us to test the null-hypothesis of an unbiased population (or even reversely biased with respect to the observation). We have structured the revised version of the manuscript around this natural line of inference and hope, that the revised version now explains well the role and the meaning of the hypothesis tests. We have made an effort to clarify the null hypothesis and the assumptions underlying the applied tests in Sec. 3.3.

Technically, the main obstacle for this line of inference was the uncertainty comprised in samples that were meant to be tested. Hence, in Sec. 3.2. we establish a general framework for the application of hypothesis test to uncertain samples, since we could not find any such framework in the literature. The referee is right – in the case of the t-test and WSR test, this boils down to applying the tests to the individual members of the MCMC sample. However, this is not exactly the case for the bootstrap test. Since the propagation of uncertainty is not straight forward, we have treated this aspect with caution and provide detailed derivations. An interpretation of the uncertain p-value is provided in Sec. 3.2.

'This starts with the observation that for a non-significant distance the resulting p-values will be uniformly distributed, so the distribution of p-values that the authors present needs to be interpreted within that context.'

In our case, the distribution of p-values arises from the uncertainty in the individual lag measurements within the samples. The distribution indicates the probability that the observed sample corresponds to a certain p-value with respect to the null-hypothesis. The uniform distribution of p-values mentioned by the referee indicates that the probability to realize an n-sample with some p-value p from the population is uniform, given the null hypothesis holds true. Therefore, we do not think comparing the derived p-value distributions to a uniform distribution is meaningful.

'What is also missing from the otherwise extensive presentation is the alternative view, that the 6000 MCMC samples each present 16 observations of the mean and could be tested accordingly as 6000*16 observations.'

Remark: In the original version of the manuscript only the lag between calcium and sodium was discussed. For this case, uncertain lags could be derived for 16 events.

We are not sure if we fully understand this comment. We assume that the referee proposes to treat all 6000 * 16 values acquired from the MCMC-sampling as equally meaningful observations of the same quantity and put all of them together into one pot. As already mentioned, we incorporated an explanation why this is not valid in the revised manuscript. Our objection to this approach draws on the

following:

There is no physical quantity that would be represented by such a pooled / gathered distribution. Physically, 16 observations have been made for 16 **different** DO-events, each of which is uncertain and hence represented by 6000 MCMC-samples. Together, these 16 * 6000 values must be regarded as an empirical probability density in 16 dimensions and not in one dimension. Disregarding this would have severe consequences for further inference.

For example: Assume that one tries to observe the outcome of a repeated random experiment. In the first attempt to observe it one is uncertain whether the observation was either 1 or 2. In the second runone observes 2 or 3. Gathering these possible observations together results in a set of observations {1,2,2,3} which corresponds to mean u=2 and a standard deviation of 2/3. However, equation 14 yields four possible vectors (u,s) which are (u=1.5, s=0.5), (u=2,s=2), (u=2, s=0) and again (u=1.5, s=0.5). All four vectors carry the same probability weight. From this, one may compute the expectations  = 2 and <s> = ¾ of the uncertain quantities u and s .

L496ff: The authors observe that the distribution that results from the convolution is narrower than the one obtained by the other methods. This is interesting albeit not surprising, given the conceptual difference of the convolution to the other methods. The brief explanation that the authors give here is quite difficult to follow, could the authors elaborate?

As mentioned above, we do not present the probability distributions for the population mean in the revised manuscript. However, for sake of clarity: Consider a certain sample with mean u and standard deviation s. According to Equation (12) [original manuscript] the u and s induce a probability distribution for the population mean $\mu$ centered around u. If the sample is now taken to be uncertain, uncountably many combinations (u,s) induce distributions for $\mu$ centered around the corresponding u. According to equation (15)[original manuscript] they all contribute to the population mean distribution under uncertainty. Hence, this distribution must be broader than the distribution for the sample mean. Or the other way around, any sample mean is associated with a broad range of possible population means that define a population which potentially has generated the sample.

L406f: I think this point deserves a moment of attention: Despite the added layer of uncertainty for the posterior distribution of U $\Delta$t and the additional assumptions going into $\mu$ $\Delta$t both still put around 4/5 of the probability on lead of Ca over Na. Yes, this is not 90/100, but in IPCC parlance it is still likely that the transitions are led by a transition in Ca.

The referee mentions a very relevant point here.  Due to the substantial changes, this question is not discussed any more in the revised manuscript.

L443ff: The calculation of the number of events being consistent with the lead of Ca over Na again is a very good addition to the discussion and a great extension of the order statistics shown in Fig 5 of Erhardt et al. (2019). The authors interpretation of the analysis is however somewhat strongly formulated given the large uncertainties of the estimates: The postulate that if an atmospheric circulation change (effecting only Ca) would trigger the sea ice retreat of the DO events (in turn effecting Na) than all of the events should show a lead of Ca over Na at their onset. This is not wrong but would only ever occur in a scenario where we would observe these atmospheric and sea ice changes directly and without error, not through a set of proxy records and could reasonably exclude any influence of internal climate variability. All in all, that seems to comprise quite a high bar. I suggest the authors to tone down the interpretation of this otherwise very enlightening analysis.

After thorough review of this section we found inconsistencies in the reasoning. A stringent discussion

of this issue is beyond the scope of this answer. We refrained from presenting this approach in the revised manuscript and will return to this in future work.

L468f: How do the authors arrive at the conclusion that the observations cannot be used to investigate the transitions with Na leading? To put it sarcastically: If that is not possible, then why is the reverse, investigating the transitions with a Ca lead?

Sorry, we think there's a misunderstanding here. The question posed in the original manuscript was *which* are the specific DO events that potentially are led by sodium. This question can in fact not be answered unambiguously, since all statements on leads and lags for individual events are probabilistic.

However, the paragraph this comment referred to was removed from the manuscript.

480ff: The statement on the ability of the presented results to serve as evidence is somewhat unjustified. I do agree that on the base of the presented data the Null Hypothesis of a zero or larger lead cannot be rejected but in reverse that does not mean that the same evidence cannot be used at a later stage (combined with prior knowledge and more evidence).

The revised manuscript uses a slightly different wording (l.592). We agree, that at a later stage, the data and the analysis presented by Erhardt et al. may certainly be used in combination with more data ore other methods to draw conclusions on the sequence of events at the onset of DO events. However, in our opinion, the chosen wording does not conflict with the demands of the referee. Our statement 'However, if the common proxy interpretations hold true, our findings suggest that the hypothesis of an atmospheric trigger - either of hemispheric or synoptic scale - for the DO events should not be favoured over the hypothesis that a change in the North Atlantic sea-ice cover initiates the DO events.' refers to the data as presented by Erhardt at al. and additional information may certainly falsify this statement.

486ff: The possible existence of a process other than the processes that directly influence Ca or Na is an important note here and is likely the best explanation for what is visible in the data. The authors could spend a little more time on this point.

We thank the referee for this comment. It is true that the manuscript has a strong emphasize on the methods, while the possible physical mechanisms at work during DO-events are treated only to a limited extent. Given the restructuring of the manuscript and the strong focus on the hypothesis testing the potential conclusions that can be drawn from the analysis are somewhat narrower compared to the original manuscript. That is why we do not elaborate on potential triggers which are not reflected by the investigated proxies in the revised manuscript. Instead, we restrict our conclusions to the key finding, that the observed lag tendency cannot be discriminated from one that has arisen purely by chance and that hence, the hypothesis of an atmospheric trigger should not be favored over the hypothesis of a sea ice trigger at this stage.

**Point by point answer to referee 2:**

General Comments

1. The authors do not specifically discuss how their approach is different to Erhardt et al 2019. While the manuscript gives the impression that previous studies completely disregarded uncertainties and used expectation values, this is not true for Erhardt et al, who included the uncertainties in a rigorous manner in their own right. The difference lies in the interpretation of the estimated onsets as random

variables. Erhardt et al consider the uncertain samples as measurements of the same fixed quantity (a fixed time lag of Ca and Na at DO onsets), which allows them to simply multiply the individual MCMC posteriors to obtain a single posterior distribution that represents the measurement uncertainty of the fixed time lag. In contrast, the present study interprets the time lag to be a random variable, varying in between DO events. Thus, they cannot multiply the individual posteriors. In the discussion, the authors contrast their approach with the results one would obtain when completely discarding the uncertainties in the onset determination. It is not very surprising that including uncertainty yields hypothesis tests which are no longer significant. Instead, it would be more relevant to highlight the contrasting results of their approach to Erhardt et al.

1. We agree that this should have been described more clearly and discuss the differences in the underlying assumptions explicitly in the Methods section (3.4) and the Discussion section of the revised manuscript. Additionally, in Figure 5 compare the 'combined evidence' according to Erhardt et al. with results obtained for the uncertain sample mean (this study). While Erhardt et al. assume that the time lag between the calcium and the sodium transition was a physical constant, such as the speed of light, we argue that climate variability will cause this time lag to differ from event to event. Assuming that the DO mechanism remains unchanged between events, it is very convenient to treat the observed time lags as the result of a random experiment that was performed repeatedly. Even if the results of the hypothesis tests were not surprising given the uncertainties in the observations as the referee states, they still contradict the conclusions of Erhardt et al. who state: 'Taken at face value, this sequence of events suggests that the collapse of the North Atlantic sea-ice cover may not be the initial trigger for the DO events and indicates that synoptic and hemispheric at- mospheric circulation changes started before the reduction of the high-latitude sea-ice cover that ultimately coincided with the Greenland warming'. Our tests indicate that one cannot discriminate the observed tendency for a sodium / d18O lag with respect to Ca or the annual thickness from a lag that has occurred purely by chance and in the absence of a systematic mechanism. In this way, our results from the hypothesis tests add valuable information to the debate on the succession of events at DO onsets and we conclude contrarily: 'However, if the common proxy interpretations hold true, our findings suggest that the hypothesis of an atmospheric trigger - either of hemispheric or synoptic scale - for the DO events should not be favoured over the hypothesis that a change in the North Atlantic sea-ice cover initiates the DO events.' (l.592)

2. The authors speak of a rigorous propagation of uncertainties to p-values, among other things. To achieve this, they introduce probability distributions of the mean and other test statistics, which are formally represented using delta distributions arising from the empirical sample. In practice, these distributions are all computed by summing a random sample from the individual MCMC posteriors, which is essentially bootstrapping if I understand correctly. Can the authors comment on how the results of this work would be different if they simply summed the individual MCMC posteriors (which would be equivalent to bootstrapping as well), and then looked at the arising distribution of the lag, determining the probability of a lag >=0? This would be much simpler and the obvious alternative approach to the multiplication of the posteriors by Erhardt et al. It would probably also give a non-significant result regarding a Ca2+ lead.

Remark: In the original version of the manuscript only the lag between calcium and sodium was discussed. For this case, uncertain lags could be derived for 16 events.

2. We are not sure if we fully understand this comment. The referee proposes to 'sum the individual MCMC posteriors and look at the arising distribution of the lag'. We interpret this as follows: For each DO-event the MCMC samples 6000 lags from the corresponding posterior distribution. The referee proposes to put all these 16*6000 values together in one pot and regard them as observations of the

same quantity. Our objection to this approach draws on the following:

There is no physical quantity that would be represented by such a pooled / gathered distribution. Physically, 16 observations have been made for 16 **different** DO-events, each of which is uncertain and hence represented by 6000 MCMC-samples. Together, these 16 * 6000 values must be regarded as an empirical probability density in 16 dimensions and not in one dimension. Disregarding this would have severe consequences for further inference.

For example: Assume that one tries to observe the outcome of a repeated random experiment. In the first attempt to observe it one is uncertain whether the observation was either 1 or 2. In the second runone observes 2 or 3. Gathering these possible observations together results in a set of observations {1,2,2,3} which corresponds to mean u=2 and a standard deviation of 2/3. However, equation 14 yields four possible vectors (u,s) which are (u=1.5, s=0.5), (u=2,s=2), (u=2, s=0) and again (u=1.5, s=0.5). All four vectors carry the same probability weight. From this, one may compute the expectations  = 2 and <s> = ¾ of the uncertain quantities u and s .

We have included a paragraph on this issue in the discussion (l.571).

The referee states: 'In practice, these distributions are all computed by summing a random sample from the individual MCMC posteriors, which is essentially bootstrapping if I understand correctly.' Here, as well we are not sure if we understand this statement correctly. In our understanding a random sample from the posterior lag distribution for a single DO event corresponds to the set of 6000 values, sampled by the MCMC from the corresponding posterior probability distribution. The empirical density distribution that is induced by this MCMC-sample approximates the true posterior distribution for the DO-event in the sense of equation (4) (l. 170). The MCMC-sampling procedure that gives rise to the sample of 6000 values is not equivalent to bootstrapping. The MCMC algorithm is used to sample from a continuous probability density, because the distribution is too complex to use it directly for inference. Bootstrapping is used to generate synthetic samples from a finite size sample by 'drawing with replacement', not to sample from a continuous probability distribution.

Appendices A and B are dedicated to the empirical density distribution and how they arise from the MCMC sampling procedure.

3. I am wondering in what way the hypothesis tests introduced in Sec. 3.6-3.7 are necessary and add to the results? When reading the manuscript, I was surprised that hypothesis tests were introduced after distributions for the sample or population mean had already been given, which would directly allow to test the hypothesis of a mean >=0?

The starting point of this study was the observation by Erhadt et al, that transitions in Ca2+ and the annual layer thickness start on average about one decade earlier than their counterparts in Na+ and d18O. The current version of the manuscript sets out to answer the question, if this observed lag can be attributed to an underlying mechanism or whether it may have arisen simply by chance. For this aim, hypothesis tests constitute the right tool. We have preferred this line of inference over the derivation of an uncertainty distribution for the population mean, because it is appears natural at this stage and gives a clear answer to a well posed question.

4. Why did the authors restrict the analysis to Na and Ca, and omitted an analysis of the offsets of d18O and lambda? This would be an important consistency test, and might even be more relevant since lambda has a very direct meaning as a proxy, and d18O is the most important of all proxies.

This point was addressed in our first statement, we added the investigation of the other variables in the revised manuscript

Specific comments:

L15ff: I think the conclusions should be stated differently. In my interpretation, the analysis does not contradict a lead or lag. Rather, as a result of the large uncertainties in estimating the individual onset timings, it cannot be excluded that there are in fact no leads or lags. Similarly, on the grounds of the uncertainties in the individual event timings, there is not enough evidence to conclude that atmospheric reorganization systematically preceded sea ice retreat for all events.

We agree and would adjusted the abstract in the sense of the comment by the referee.

L34-35: The interstadials lasted up to 10 millennia, with 1.5 millennia being the average.

Thank you, we changed this to: 'The abrupt warming is followed by gradual cooling over centuries to millennia, before the climate abruptly transitions back to cold conditions'.

L130: Why does the transition detection fail for some events?

For sake of comparability we included the same DO events previously investigated by Erhardt et al. - all of these events are successfully fitted by the ramp-fit method. We removed the comment on the success criterion of the MCMC sampler, but included an explanation on the data selection (l.129)

L150: How would this depend on the autocorrelation of the noise? The Na and Ca records might have different noise structure and thus there is the potential for systematic biases indeed.

We revisited the robustness test carried out by Erhardt et al. and agree with the referee 1, that this legitimates the application of the Bayesian transition detection method. Furthermore, the focus of this study is the application of hypothesis tests to uncertain data samples. Hence, we removed all considerations regarding the performance of the ramp-fit method and rely on the assessment provided by Erhardt et al..

Eq. 7: Maybe the authors can elaborate specifically in the text what this approximation does. Since the order of MCMC samples for each event is arbitrary, by associating the i-th MCMC sample for every event, it seems like it is just a random sampling of m=6000 points in the joint space. If this is the case, why not simply sample randomly in the first place, and why not choose many more than m=6000 points? Or is this rather done in order to simplify the notation?

The referee is correct. Associating the i-th individual member of every event's MCMC-sample is a random choice and is indeed the simplest in terms of notation. The impact of this choice is discussed in detail in the Appendices A and B. In short, the sample is large enough such that the final results are robust against any sort of randomization of this choice.

L278ff: I am not sure why the authors say that they are only given "relative" data, since they estimate the onsets in the two proxies. Furthermore, since until this point the data was already given exclusively as onset timing differences, I wonder why it is necessary to introduce "paired samples" now? This is a bit confusing to the reader.

We agree that the sentence is not very helpful and removed it. In fact, the lags constitute relative data in the sense that there is no absolute time for the onset of the DO event. The timings of two transition onsets detected in different proxies in this study receive meaning when being compared to each other. Further, a measurement of the timing of the transition in one proxy from the i-th DO event cannot be compared to the timing of the transition in another proxy from the j-th DO event, that's why the data is

paired.

L348: I might have missed this somewhere, but how is the distribution of the test statistic under the null hypothesis constructed?

The original manuscript did not properly introduce the null distribution of the WSR test. It is mostly a combinatorical problem but must be computed explicitly. In the revised manuscript, in line 334 we provide guidance for the construction of the WSR null distribution, which can as well be found in lookup tables.

L396: Could the authors explain more explicitly why the sample and population mean distributions are so different, and what this means for the interpretation of the results? Which distribution should be preferred?

Note that we do not show the population mean's distribution anymore in the revised manuscript. For sake of clarity, we nevertheless respond here: Assuming that all DO events followed the same physical mechanism, we regard the observed time lags as the outcome of a repeated random experiment with the randomness being due to climate variability.  A random experiment is fully characterized by its population. The population mean indicates systematic leads or lags and hence is the decisive quantity. The explanatory power of the sample mean U (termed average lag by the referee) is limited to the fact that it is the best estimate (point estimate in case of a sample without uncertainty) of the population mean; conclusions based on the sample lag hence need to be accompanied with information on how likely it would be to obtain this sample mean from a population characterized by a null hypothesis

To give a simple example: If you want to judge whether a coin is biased, it is not sufficient to toss the coin 16 times (with head assigned a value of 0 and tail assigned 1) and report on a mean of, say, 0.3, arguing that it is different from 0.5, the *expected outcome* if the coin is unbiased. Rather, one would have to compute the probability of obtaining a mean of 0.3 or smaller under the assumption that the coin is unbiased. If this probability is small, one can (at a reported significance level) argue that the coin is biased.

Also for the reasons given in the introductory statement, we hence prefer applying hypothesis tests to the uncertain sample over deriving the distribution for the population mean.

The population mean distribution must necessarily be broader than the sample mean distribution. If the observation was free of any uncertainty, the sample mean would be a scalar. The t-distribution would induce a probability distribution for the potential population mean around this sample mean. Now, with the sample mean being uncertain itself, the certain sample mean probability distribution for the population is convoluted with the sample mean distribution and the convolution of two functions is always broader than either of the functions being convoluted with one another.

L441ff: Here the authors introduce another simple method to address the likelihood of a systematic Ca2+ lead. I think it would be more coherent if this were moved to the Results Section. Furthermore, I find the conclusions from this simple calculation too confident, and in spirit contradictory to the interpretation of the other results. The probability of obtaining exactly n=16 events with a Ca lead will always be very small when there is a relatively large measurement uncertainty of the individual lags (spanning both positive and negative values), even if all individual posteriors would be clearly centered at negative values. Just like the probability of flipping 16 out of 16 heads is still very low for a strongly biased coin. This does not allow one to contradict the hypothesis that all events would follow the same pattern with a Ca2+ lead. For the data at hand, all but two events show posterior distributions centered at a negative value. Maybe the authors could write instead that from the MCMC posteriors it seems

unlikely that all DO events occurred with a preceding abrupt change in Ca2+. However, this might merely reflect the fact that due to the large uncertainty in estimating the onsets, the individual MCMC posteriors have significant support for positive lags as well. Arguing like this would also be much more in line with the authors' earlier statements that they cannot infer an absence of causality from their non-significant tests.

The comment of the referee made us review the reasoning invoked in this section. In fact, we find inconsistencies and therefore not present this at all in the revised manuscript. We will try to reconcile the inconsistencies and derive further statements on a potential causal relationship between the two transitions at a later stage.

Figure 1 and Table 1: Just to be sure, can the authors confirm that the time scale they use really is years BP (before the year 1950 AD), and not years b2k (before the year 2000), which is what is commonly used in GICC05?

According to the data source (Erhardt et al. 2019), the age scale in Figure 1 is in fact given in years BP. The ages of the DO events, which are due to Rasmussen 2014 have been converted accordingly.

L18: ...holds true, the we conclude...

the sentence was removed

L98: Instead of "compression" rather say: ...due to the thinning of the annual layers in the core.

was corrected (l.117)

Figure 3: Maybe it would be good to choose a different color for the null hypothesis in panel b. Otherwise it gives the impression that it corresponds in some way to the blue distribution in panel a.

Thank you - this is true. This was adjusted – now Figure 4.

Eq. 14 and Eq. 16: I am wondering whether "u" in the second delta function should be replaced by the empirical mean sum(y_i)/n?

Mathematically this should be the same, since the first 'delta function' forces sum(y_i)/n to assume the value u. The equations now only appear in Appendix A. For sake of readability, we did not change the notation.

L235: Maybe "marginal distribution" would correspond better to the nature of this distribution?

The sentence was removed from the manuscript.

Eq. 17: What is the notation u_j and s_j?

The corresponding equation was removed from the manuscript.

Eq. 20: I am unfamiliar with this definition of a p-value. Is the integration not simply over all phi < phi_0 (for a one-sided test)? The definition here could lead to rather strange results for very asymmetric and long-tailed distributions.

The definition provided in Equation (20)[original manuscript] corresponds to the 2-sided test but in fact only holds true under certain conditions.  We now present a more common equation (11) for the onesided left-tailed p-value (l.251).

L387: missing delta in the sum.
Was removed.

Figure 4: Larger fonts and panels would be nice for better visibility.
We will adjust the figure to improve readability.

Eq. 35: It would be good if the authors could point to where the individual probabilities
in the product come from.
This section was removed from the manuscript.

References:

Erhardt, T. et al. Decadal-scale progression of the onset of Dansgaard-Oeschger warming events. Clim.
Past 15, 811–825 (2019).

---

## Referee Report (RR1)

**Review of the revised manuscript by Riechers&Boers**

To begin with, let me express my appreciation to the authors for taking the time and making an effort to completely revise their manuscript following the first round of reviews. The revised and streamlined methods section is now much easier to follow and more clearly conveys the authors approaches. With the expanded results section, the manuscript now also aims to not merely provide a statistical toolset but also paleo-climate relevant results. With this the authors have addressed the major concerns voiced by the reviewers. Judging from their thorough replies the authors have also addressed most of the concerns that the reviewers had in a satisfactory way.

Nevertheless, I still have major concerns regarding the methods the authors present which I unfortunately missed in the first round of reviews.

**Main remarks**

In their Eq. 29 and the Appendix C the authors derive the posterior distribution of the lag, averaged over all DO-events (even though the they choose to not call it that). This distribution essentially encodes the entire knowledge about the mean lag over the DO-events, given the model and the data, the quantity the statistical tests the authors make are concerned with. The fact that these tests than yield "non-significant" results for the proxy-pairs that put low posterior probability on a positive average lag is difficult to reconcile and merits a more detailed discussion by the authors.

One reason might be that the notion of significance levels and decision thresholds in the context of distributions of p-values is not as straight forward as the authors convey. As the authors point out because of the non scalar nature of the p-value distributions that they employ, they need a way to map them to a binary decision about the null hypothesis (L 284f). The authors list three different decision criteria, all employing the same significance level $\alpha = 0.05$:

1. the p-value value for the expected lag, averaged over the different events: $p(E(\mathbf{\Delta t}))$ not exceeding $\alpha$;

2. the expected p-value of, averaging over the MCMC samples of $\mathbf{\Delta t}$: $E(p)$ not exceeding $\alpha$; and finally

3. the fraction MCMC samples of $\mathbf{\Delta t}$ for which the p-values do not exceeded $\alpha$: $P(p < \alpha) < 0.90$.

Of these three criteria the authors postulate that only the latter two provide the propagation of all uncertainties to the decision about the significance of the lag. Hence the authors base their main argument of the results of the tests (2) and (3).

In the textbook meaning of the significance level $\alpha$, $\alpha$ denotes the probability that the significance test falsely rejects the null-hypothesis given that it is in fact true. Which is the meaning that almost all readers of the study will be familiar with. However, the significance criteria favoured by the authors to make their argument do not possess this property.

To test the implications of the decision levels proposed by the authors, one can run a simulation of the null-hypotheses repeatedly and assess the rate of false rejections for the different criteria. To do so, consider the following simulation, mimicking the null-hypothesis in the setting of the $t$-test: The lag for each of the $n$ events is assumed to be normally distributed as

$$\Delta t_n \sim N(\Delta T_n, 1^2) \tag{1}$$

representing the draws from the MCMC sampler for the vector of delays $\mathbf{\Delta t} = (\Delta t_1, \Delta t_2, ... \Delta t_n)$. For each of the events, $\Delta T_n$ is drawn i.i.d from

$$\Delta T_n \sim N(0, 1^2) \tag{2}$$

representing the sample of $n$ DO-events drawn from a normal distribution centred around zero.

Doing so repeatedly for 16 $\Delta T_n$ and than 6000 $\Delta t_i$ each, the effect of the different criteria by the authors can be tested given the null-hypothesis is true. Admittedly, this simulation uses an extremely idealised case for the null-hypothesis, but one could argue that drawing from standard Normal distribution, especially for the $t$-test, resembles an ideal case scenario. The code to generate the simulation results will of course be provided to the authors if desired. The same type of simulation could be done for the other two tests the authors use.

Figure 1 shows histograms of the outcomes of all three decision criteria after simulating 50000 tests under the null-hypothesis using the setup outlined above. In each of the panels, the red dashed line indicate the decision criterium proposed by the authors.

[Figure]

Figure 1: Results of simulation 50000 t-tests using the sampling scheme for the null-hypothesis outlined in the text. The red dashed lines indicate the decision criteria for significance at $\alpha = 0.05$, used by the authors.

As mentioned earlier, a significance level of $\alpha$ indicates that, given the null-hypothesis is true, in the long run exactly this fraction of the tests will have a p-value that is below this level. For a regular test statistic this is due to the fact, that under the null-hypothesis the p-values are distributed uniformly between 0 and 1. For the significance criteria the authors present, this is only the case for $p(E(\mathbf{\Delta T}))$ (1), as shown in the leftmost panel of Figure 1. Both other tests fail to exhibit this behaviour. This means that the significance criteria used by authors are not consistent with each other. In fact, both of the other tests (2, 3) use a much higher bar for significance than $\alpha$ would lead the reader to believe as indicated by the two other panels of the Figure. For (2), $E(p)$,

the probability of $E(p) < \alpha = 0.05$, given the null-hypothesis, is only 0.0017, almost a factor of 30 lower than the implied value of 0.05. The same holds true for the third proposed criterion, where out of the 50000 simulations, only five exceed the $P(p < \alpha) > 0.9$ threshold. Accordingly, by using these two criteria, the authors set the bar for a "systematic lag" a lot higher than the significance level of $\alpha = 0.05$ would make it appear. Consequently, both of these test do yield a non-rejection of the null hypothesis for all studied cases in turn leading the authors to their main conclusion: that there is no evidence for a lead-lag relation in any of the studied proxy pairs.

To conclude, I am not convinced that the testing criteria used by the authors to draw their main conclusions are fit for the purpose as stated in the manuscript. By implicitly using much more stringent significance criteria for their two preferred tests, without clearly communicating this to the reader, the authors unfortunately seem to have (accidentally) moved the goal post for what they are trying to investigate. In my eyes this necessitates either a complete re-evaluation of the results of the statistical tests and the conclusions drawn from these results by the authors or a removal of the tests in question from the manuscript.

As an interesting side note: the results from the first decision criterium, deemed "simplistic" by the authors (L 265), are in general agreement with both the "combined evidence" of Erhardt et al. (2019) as well as with the "uncertain sample mean" as derived by the authors. Maybe unsurprisingly, it indicates exactly the opposite of the two other tests: significant evidence at $\alpha = 0.05$ for a less than zero lag across all studied proxy pairs. Obviously, this decision criterion is most closely related to these probabilistic quantities, as it tests wether the average delay between a pair of proxies, taken over all observed events is significantly different from zero. Which is, as far as I understand, what the authors try to investigate.

Additionally, prompted by the authors statement about the novelty of their "uncertainty propagation to p-values" in their response: A cursory literature search (keywords *fuzzy p-values*, *bayesian p-values*) brought up a range of papers that seem to be dealing with p-values in settings similar to the setting the authors deal with here. It would be good if the authors set their approach into the context of aforementioned literature or highlight its conceptual differences should the they decide to further employ it in the next iteration of the study.

**Other remarks**

**L 129ff**  It is not true that Erhardt et al. (2019) only considered time series free from data-gaps. In fact, looking at Figure 3 in the presented manuscript, both the $Ca^{2+}$ as well as the $Na^{+}$ data series do in fact exhibit at least one section of missing data. Please reformulate.

**L 198ff**  The choice of terms for the different type of distributions is a little bit misleading. Posterior distributions as generate by Bayesian inference are probability distributions. Yes, posterior distributions carry the uncertainty about an inferred parameter conditional on the data and the model, but they are probability distributions nontheless. By using the neologism "uncertainty distributions" the authors implicate that they are more uncertain than their "probability distributions" generated by random experiments. This sets the tone for the discussion that follows in a very odd way. The authors should use the correct term "posterior probabilities". Changing this would also avoid such contortions as the "high uncertainty probabilities" (L 305)

**L 208ff**  Prescribing a "fixed pattern of causes and effects" is an overly strong interpretation. It is quite easy to imagine a range of mechanisms that have an indistinguishable imprint in the

proxy record but trigger a transition from stadial to interstadial conditions. For example the ocean processes alone, that the authors list in the introduction are probably very difficult to distinguish using the proxies presented here as they partly invoke very similar feedback mechanisms. I would suggest to reformulate "similar pattern of cause and effects" to convey the possible ambiguity as a discussion of the imprint in the proxy records by the different causes clearly goes beyond the focus of this manuscript. The same holds true for later implications of the one trigger of DO-Events that the authors make throughout the manuscript.

**L 201** The imperative "should" in regards to the setup of the frequentist analysis that follows should be replaced with "could" or "can".

**L 220** Either "be" or "bear".

**L 224** The footnote should either be added to and discussed in the paper or removed. As it is right now it is just a clever remark that does not contribute to the overall manuscript.

**L 263** I think it would be better to say that the sample no longer *only* carries the randomness of the population. As would be the case for a regular statistical test with certain values.

**L 284** In null-hypothesis significance testing the null-hypothesis can only be rejected.

---

## Referee Report (RR2)

Thanks to the authors for revising the manuscript and responding to my comments. The manuscript is very much improved, but I have a further comment to their response, followed by some small remarks on the revised manuscript.

**Regarding comment 2 and L572-581 in the revised manuscript.**

What I was proposing is to use a mixture distribution. The individual DO events are then regarded as subpopulations. Allowing for different subpopulations accounts for the fact that the DO events are heterogeneous, and thus cannot be expected to all show the exact same value of time lag in between proxies, just as the authors are arguing. Still, they are all DO events and thus it makes sense to ask questions on the population level, which is described by the mixture distribution. So one can e.g. simply test whether the population mean is greater or equal zero.

As I said previously, to me this is the obvious alternative to what Erhardt et al were doing. They are multiplying the individual posteriors, while in a mixture distribution the individual posteriors are added, thus yielding a larger variance. This increased uncertainty would most likely yield a non-significant lag in the proxies, in line with what is presented in the paper.

I understand from the viewpoint of the authors why they want to construct a 16-dimensional distribution. But I am not sure why they invoke "physical quantities". In my opinion this is just a matter of different interpretations of uncertainty, and I would like to hear why the authors think what they are doing is more physical. The way that the authors argued in their response so far is by assuming a priori that their interpretation is the only correct one. I do agree that the two procedures do not necessarily give the exact same results, but I would just like to hear from the authors why they think their complicated approach is necessary.

I guess the main difference to just using a mixture is that the procedure presented here accounts for the fact that we have just 16 observations, which gives additional uncertainty, as reflected e.g. by the larger standard deviation in their example. This information is lost when one just looks at the mixture distribution. But it can be easily recovered, e.g. when using a bootstrap test with the given sample size.

Related to this, in my earlier comments with bootstrapping I meant simply random sampling. A mixture distribution is constructed by random sampling from the individual components, and that's why I was thinking that this approach is very similar to the authors'.

Again, I am not saying what the authors are doing is wrong in any way, or identical to what I am proposing. The manuscript would simply benefit if the authors could clarify why not a much simpler, and equally reasonable, approach is taken.
At least statements like "*the combined set of MCMC samples does not correspond to any mathematical object and hence its interpretation remains unclear.*" in the revised manuscript need to be corrected.

**L86ff**: "*In order to review the statistical evidence for a potential systematic lags, we formalize the notion of a 'systematic lag': We call a lag systematic if it is enshrined in the random experiment in form of a population mean different from zero. Samples generated from such a*

*biased population would systematically (and not by chance) exhibit sample means different from zero. Accordingly, we*
*formulate the null hypothesis of a pairwise unbiased transition sequence, that is, a population mean equal to zero.*"

Could this maybe be reformulated, in order to also acknowledge the fact that whether or not a truly biased sample can be expected to systematically exhibit sample means (significantly) different from zero depends on the sample size? To me, this seems to be an important motivation for the procedure proposed here.

**Caption Fig. 2**: I was not sure at first whether the posteriors shown are from the authors or from Erhardt et al. Maybe this could be made specifically clear in the caption.

**L187**: indipendent → independent

**L797-798**: I don't understand what is meant here, maybe just a grammatical error?

---

## Editor Decision (ED1)

[revised manuscript text omitted]

$$\underline{\Delta}\hat{\mathbf{T}} = \left( \Delta\underline{T}\hat{T}_1, \Delta\underline{T}\hat{T}_2, ..., \Delta\underline{T}\hat{T}_n \right), \quad \text{with} \tag{13a}$$

$$\rho_{\underline{\Delta}\hat{\mathbf{T}}}(\underline{\Delta}\hat{\mathbf{t}}) = \prod_{i=1}^{n} \rho_{\Delta\underline{T}_i\hat{T}_i}(\Delta\underline{t}\hat{t}_i). \tag{13b}$$

300 Note that the uncertainty represented by the uncertain sample originates from the observation process - the sample no longer carries the generic randomness of the population $\mathcal{P}_{\Delta T}$ it was generated from. The $\Delta T_i$ $\Delta\hat{T}_i$ are no longer identically but yet independently distributed.

A simplistic approach to test  hypotheses on an uncertain sample would be to average over the uncertainty distribution and subsequently apply the test to the resulting expected sample

$$\text{E}(\underline{\Delta}\hat{\mathbf{T}}) = \left( \text{E}(\Delta\underline{T}\hat{T}_1), \text{E}(\Delta\underline{T}_2), ..., \text{E}(\Delta\underline{T}\hat{T}_n) \right) = \left( \int \Delta\underline{t}\hat{t}_1 \, \rho_{\Delta\underline{T}_1\hat{T}_1}(\Delta\underline{t}\hat{t}_1) \, d\Delta\underline{t}\hat{t}_1, \, ... \, , \int \Delta\underline{t}\hat{t}_n \, \rho_{\Delta\underline{T}_n\hat{T}_n}(\Delta\underline{t}\hat{t}_n) \, d\Delta\underline{t}\hat{t}_n \right).$$

305 (14)

 Averaging out uncertainties, however, essentially implies that the uncertainties are ignored and is thus always associated with a loss of information. The need for a more thorough treatment, with proper propagation of the  uncertainties, may be illustrated with a simple example. Consider a sample 310 $\mathbf{x} = (x_1, ..., x_n)$ that was generated from a population $\mathcal{P}_X$ and measured with very high precision such that the $\rho_{\hat{Y}_i}(\hat{y}_i)$ can be assumed to be $\delta$-peaks around the values $\hat{y}_i$. Assume that, for some reason, after the measurements have been carried out, the observer is unsure about the sign of the observed values. For example, in a voltage measurement one might have confused plus and minus. In this case, the joint uncertainty distribution allocates 50% probability to the value $(\hat{y}_1, ..., \hat{y}_n)$ and 50% probability to the value $(-\hat{y}_1, ..., -\hat{y}_n)$. The expected sample $\text{E}(\hat{\mathbf{Y}}) = 0$ is obviously not useful to test hypotheses on the population $\mathcal{P}_X$.

315 In contrast, uncertainty propagation gives rise to an uncertainty distribution of the $p$-value, which indicates the plausibility of a certain $p$-value in view of the data and in view of the limitations in the transition onset detection. The propagation of the uncertainties from the level of observation to the test statistic and finally to the $p$-value is illustrated in Fig. 4.

The uncertainty propagation relies on the  fact that applying a function $f: \mathbb{R} \to \mathbb{R}$ to a real valued random (uncertain) variable $X$  320 yields a new random (uncertain) variable $G = f(X)$, which is distributed according to

$$\rho_G(g) = \int \delta\left(f(x) - g\right) \rho_X(x) \, dx. \tag{15}$$

Analogously, the uncertain test statistic  $\hat{\Phi} = \phi(\underline{\Delta}\hat{\mathbf{T}})$ follows the distribution

$$\rho_{\underline{\Phi}\hat{\Phi}}(\phi'\hat{\phi}) = \int \delta(\phi(\underline{\Delta}\hat{\mathbf{t}}) - \phi'\hat{\phi}) \, \rho_{\underline{\Delta}\hat{\mathbf{T}}}(\underline{\Delta}\hat{\mathbf{t}}) \, d\underline{\Delta}\hat{\mathbf{t}}. \tag{16}$$

In general, primes are used to distinguish scalar values from functions, when necessary. Repeated application of Eq. 15 yields the uncertainty distribution of a given test's $p$-value $\hat{P} = p(\phi(\mathbf{\Delta\hat{T}}))$:

$$\rho_{P\hat{P}}(p'\hat{p}) = \int \delta\left(p(\phi'\hat{\phi}) - p'\hat{p}\right) \rho_{\Phi\hat{\Phi}}(\phi'\hat{\phi}) \, d\phi'\hat{\phi} = \int\int \delta\left(p(\phi'\hat{\phi}) - p'\hat{p}\right) \delta\left(\phi(\mathbf{\Delta t\Delta\hat{t}}) - \phi'\hat{\phi}\right) \rho_{\mathbf{\Delta T\Delta\hat{T}}}(\mathbf{\Delta t\Delta\hat{t}}) \, d\mathbf{\Delta t\Delta\hat{t}} \, d\phi'\hat{\phi}$$

$$= \int \delta\left(p(\phi(\mathbf{\Delta t\Delta\hat{t}})) - p'\hat{p}\right) d\mathbf{\Delta t\Delta\hat{t}}. \quad (17)$$

In the example shown in Fig. 4 the initial  uncertainties in the observations translate into an uncertain $p$-value that features both probability for significance and probability for non-significance. This illustrates the need for a criterion to project the uncertain $p$-value onto a binary decision space comprised of rejection and acceptance of the null hypothesis. We propose to consider the following criteria to facilitate an informed decision:

– *The hypothesis shall be rejected at the significance level $\alpha$ if and only if the expected $p$-value is less than $\alpha$, that is*

$$\int_0^1 p\hat{p} \, \rho_{P\hat{P}}(p\hat{p}) \, dp \, d\hat{p} < \alpha. \quad (18)$$

– *The hypothesis shall be rejected at the significance level $\alpha$ if and only if the probability for $p$ to be less than $\alpha$ is greater than a predefined threshold $\eta$ (we propose $\eta = 90\%$), that is*

$$\mathcal{P}_P(p < \alpha) = \int_0^\alpha \rho_P\pi(p) \, dp > \hat{P} < \alpha) = \int_0^\alpha \rho_{\hat{P}}(\hat{p}) \, d\hat{p} > \eta. \quad (19)$$

While the $p$-value of a certain sample indicates its extremeness with respect to the null distribution,  the expected $p$-value may be regarded as a measure of the uncertain sample's extremeness. Given the measurement uncertainty, the quantity $\pi(\hat{P} < \alpha)$ indicates the informed estimate of the observer that the true value of the measured sample is in fact statistically significant with respect to the null hypothesis. Thus, the first criterion assesses how 'strongly' the uncertain sample contradicts the null hypothesis, while the second criterion evaluates the likelihood of the uncertain sample  to contradict the null hypothesis. Depending on $\eta$, in many cases both criteria will yield the same decision. If not, the specific situation determines which of the criteria is more convenient. Under some circumstances one might want to guarantee that in fact the probability to achieve a significant test result is high – e.g. when mistakenly attested significance is associated with high costs. In these cases the second criterion is more appropriate even though it does not imply that the first criterion is fulfilled.

**3.3 Hypothesis tests**

We have introduced the notion of uncertain samples and its consequences for the application of hypothesis tests. Here, we shortly introduce the tests used to test our null hypothesis that the observed tendency for delayed transition onsets in $Na^+$

and $\delta^{18}O$ with respect to $Ca^{2+}$ and $\lambda$ has occurred by chance and that the corresponding populations $\mathcal{P}_{\Delta T}^{p,q}$ that characterize the  pairwise random lags $\Delta T^{p,q}$ do in fact not favour the tentative transition orders apparent from the observations. Mathematically, this can be formulated as follows:

– Let $\rho_{\Delta T}^{p,q}(\Delta t)$ be the probability density associated with the popuplation of DO transition onset lags $\mathcal{P}_{\Delta t}^{p,q}$ between the proxy variables $p$ and $q$ and let the observations  $\mathbf{\Delta \hat{T}}^{p,q}$ suggest a delayed transition of the proxy $q$ - that is,  the corresponding uncertainty  probabilities for negative  $\Delta \hat{T}_i$ across the sample according to Eq. 7. We then test the hypothesis $H_0$ : 'The mean value $\mu^{p,q} = \int \rho_{\Delta T}^{p,q}(\Delta t)\, d\Delta t$ of the population $\mathcal{P}_{\Delta T}^{p,q}$ is  greater   zero.'

We identified three tests that are suited for this task, namely the $t$-test, the Wilcoxon-signed-rank (WSR) test, and a bootstrap test. The WSR and the $t$-test are typically formulated in terms of paired observation $\{x_i, y_i\}$ that give rise to a sample of differences $\{d_i = x_i - y_i\}$ which correspond to the time lags  $\{\Delta t_i^{p,q}\}$ of different DO events (Rice, 2007; Lehmann and Romano, 2006, e.g.). The null distributions of the tests rely on slightly different assumptions  regarding the populations. Since we cannot guarantee the compliance of these assumptions, we apply the tests in combination to obtain a robust assessment.

**3.3.1 $t$-test**

The $t$-test (Student, 1908) relies on the assumption that the population of differences $\mathcal{P}_D$ is normally distributed with mean $\mu$ and standard deviation $\sigma$. For a random sample $\mathbf{D} = (D_1, ..., D_n)$ the test statistic

$$Z(\mathbf{D}) = \frac{U(\mathbf{D}) - \mu}{S(\mathbf{D})/\sqrt{n}} \tag{20}$$

follows a $t$-distribution $t_{n-1}(z)$ with $n-1$ degrees of freedom. Here, $U = \frac{1}{n}\sum D_i$ is the sample mean and $S = \frac{1}{n-1}\sum(U - D_i)^2$ is the samples' standard deviation. This allows to test whether an observed sample $\mathbf{d} = (d_1, ..., d_n)$ contradicts an hypothesis on the mean $\mu$. To compute the $p$-value for the hypothesis $H_0 : \mu \geq 0$ (left handed application) the null distribution is integrated from $-\infty$ to the observed value $z(\mathbf{d})$:

$$p_z(z(\mathbf{d})) = \int_{-\infty}^{z(\mathbf{d})} t_{n-1}(z')dz'. \tag{21}$$

The resulting $p$-value must then be compared to the predefined significance level $\alpha$.

The $t$-test can be generalized for application to an uncertain sample of the form  $\mathbf{\Delta \hat{T}} = (\Delta \hat{T}_1, ..., \Delta \hat{T}_n)$ as follows: Let  $\rho_{\mathbf{\Delta \hat{t}}}(\mathbf{\Delta \hat{t}})$ denote the uncertainty distribution of  $\mathbf{\Delta \hat{T}}$. Then according to Eq. 15 the distribution of the uncertain statistic  $\hat{Z}(\mathbf{\Delta \hat{T}})$ reads

$$\rho_{\hat{Z}}(\hat{z}) = \int \delta\left(\frac{u(\mathbf{\Delta \hat{t}})}{s(\mathbf{\Delta \hat{t}})/\sqrt{n}} - \hat{z}\right) \rho_{\mathbf{\Delta \hat{T}}}(\mathbf{\Delta \hat{t}})\, d\mathbf{\Delta \hat{t}}. \tag{22}$$

Finally, the distribution of the uncertain $p$-value may again be computed according to Eq. 15

$$\rho_{\underline{P}_z\hat{P}_z}(\underline{p}'\hat{p}_z) = \int \delta\left(p_z(\underline{z}\hat{z}) - \underline{p}'\hat{p}_z\right)\rho_{\underline{Z}\hat{Z}}(\underline{z}'\hat{z})\,\underline{dz}'d\hat{z} = \int \delta\left(\int_{-\infty}^{\underline{z}'\hat{z}} t_{n-1}(\underline{z}'')\,\underline{dz}'' - \underline{p}'\underline{dz} - \hat{p}_z\right)\rho_{\underline{Z}\hat{Z}}(\underline{z}'\hat{z})\,\underline{dz}'d\hat{z} \qquad (23)$$

and then be evaluated according to the two criteria formulated above.

**3.3.2 Wilcoxon-signed-rank**

Compared to the $t$-test, the WSR test (Wilcoxon, 1945) allows to relax the assumption of normality imposed on the generating population $\mathcal{P}_D$, and replaces it by the weaker assumption of symmetry with respect to its mean $\mu$ in order to test the null hypothesis $H_0 : \mu \geq 0$. The test statistic $W$ for this test is defined as

$$W(\mathbf{D}) = \sum_{i=1}^{n} R(|D_i|)\,\Theta(D_i), \qquad (24)$$

where $R(|D_i|)$ denotes the rank of $|D_i|$ within the sorted set of the absolute values of differences $\{|D_i|\}$. The Heaviside function $\Theta(D_i)$ guarantees that exclusively $D_i > 0$ are summed. The derivation of the null distribution is a purely combinatoric problem and its explicit form can be be found in lookup tables. Because $W \in \mathbb{N}_{[0,n(n+1)/2]}$ we denote the null distribution by $\mathcal{P}_W^0(w)$ to signal that this is not a continuous density. Explicitly, the null distribution can be derived as follows: First, the assumption of symmetry around zero (for the hypothesis $H_0 : \mu \geq 0$ the relevant null distribution builds on $\mu = 0$) guarantees that the chance for $D_i$ to be positive is equal to $\frac{1}{2}$. Hence, the number of positive outcomes $m$ follows a symmetric binomial distribution $\pi(m) = \binom{n}{m}(\frac{1}{2})^n$. For $m$ positive observations, there are $\binom{n}{m}$ different sets of ranks $\{r_1, ..., r_m\}$ that they may assume, and which are again due to the symmetry of $\mathcal{P}_D$ equally likely. Hence, for a given number of positive outcomes $m$ the probability to obtain a test statistic $w$ is given by the share of those $\binom{n}{m}$ configurations that yield a rank sum equal to $w$. Summing these probabilities over all possible values of $m$ yields the null distribution for the test statistic $w$.

For a given sample $\mathbf{d}$ we test the hypothesis $H_0 : \mu \geq 0$ by computing the corresponding one-sided $p$-value $p_w$, which is given by the cumulative probability that the null distribution assigns to $w'$ values smaller than the observed $w(\mathbf{d})$:

$$p_w(w(\mathbf{d})) = \sum_{i=1}^{n} \mathcal{P}_W^0(w_i')\,\Theta(w(\mathbf{d}) - w_i'). \qquad (25)$$

Since $W \in \mathbb{N}_{[0,n(n+1)/2]}$ it follows that $p_w$ assumes only discrete values in $[0,1]$ with the null distribution determining the mapping between these two sets.

The generalization of the WSR-test to the uncertain sample $\Delta\mathbf{T}\ \Delta\hat{\mathbf{T}}$ can be carried out almost analogously to the $t$-test. However, the fact that $W \in \mathbb{N}_{[0,n(n+1)/2]}$ makes it inconvenient to use a continuous probability density distribution. We denote the distribution for the uncertain $W(\Delta\mathbf{T})$ by $\hat{W}(\Delta\hat{\mathbf{T}})$ by

$$\mathcal{P}_{\underline{W}\hat{W}}(\underline{w}\hat{w}) = \int \delta\left(\sum_{i=1}^{n} R(|\Delta\underline{t}\hat{t}_i|)\,\Theta(\Delta\underline{t}\hat{t}_i) - \underline{w}\hat{w}\right)\rho_{\Delta\mathbf{T}\Delta\hat{\mathbf{T}}}(\Delta\underline{\mathbf{t}}\Delta\hat{\mathbf{t}})\,d\Delta\underline{\mathbf{t}}\Delta\hat{\mathbf{t}}. \qquad (26)$$

Given the one-to-one map from all $w \in \mathbb{N}_{[0, n(n+1)/2]}$ to the set of discrete potential values $p_w$ for $P_w$ in $[0, 1]$ determined by equation Eq. 25, the probability to obtain  $\hat{p}_w$ is already given by the probability to obtain the corresponding  $\hat{w}$. Hence, we find

$$\mathcal{P}_{P_w \hat{P}_w}(\underline{P}p_w(\underline{W}\hat{W}) = \underline{p}\hat{p}_w) = \mathcal{P}_{W \hat{W}}(\underline{w}\hat{w}). \tag{27}$$

**3.3.3 Bootstrap test**

Given an observed sample of differences $\mathbf{d} = (d_1, ..., d_n)$, a bootstrap test constitutes a third option to test the compatibility of the sample with the hypothesis that the population of differences features a mean equal to or greater than zero: $H_0 := \mu_0 \geq 0$. Guidance for the construction of a bootstrap hypothesis test can be found in Lehmann and Romano (2006) and Hall and Wilson (1991). The advantage of the bootstrap test lies in its independence from assumptions regarding the distributions' shape. Lehmann and Romano (2006) propose the test statistic

$$v = \sqrt{n}u, \tag{28}$$

with $u(\mathbf{d}) = \frac{1}{n}\sum_{i=1}^{n} d_i$ denoting the sample mean. In contrast to the  above two tests, the bootstrap test constructs the null distribution directly from the observed data. In the absence of assumptions, the best available approximation of the population $\mathcal{P}_D$ is given by the empirical density

$$\mathcal{P}_D(d) \sim \frac{1}{n}\sum_{i=1}^{n} \delta(d - d_i). \tag{29}$$

The empirical density does not necessarily comply with the null hypothesis and it thus has to be shifted accordingly:

$$\tilde{\rho}_D(d) = \sum_{i=1}^{n} \delta(d - d_i + u). \tag{30}$$

$\tilde{\rho}_D(d)$ corresponds to the borderline case of the null hypothesis $\mu = 0$. The null distribution for $v$ is then derived by resampling $m$ synthetic samples $\tilde{\mathbf{d}}_j = (\tilde{d}_1, ..., \tilde{d}_n)_j$ of size $n$ from $\tilde{\rho}_D(d)$ and computing $\tilde{v}_j = v(\tilde{\mathbf{d}}_j)$ for each of them. This corresponds to randomly drawing $n$ values from  the set $\mathbf{d} - u$ with replacement and  computing $v$ for the resampled vectors $m$

$$\rho_U^0(u) = \frac{1}{m}\sum_{i=1}^{m} \delta(u - \tilde{u}_i),$$

 times, where the index $j$ labels the iteration of this process. The resulting set $\{\tilde{v}_j\}_j$ induces the data driven null distribution for the test statistic

$$\rho_V^0(v) = \frac{1}{m} \sum_{j=1}^{m} \delta(v - \tilde{v}_j). \tag{31}$$

Setting $m = 10000$ we obtain robust null distributions for the cases $n = 16$, and $n = 20$ relevant for this study. The $p$-value of this bootstrap test is then computed as before in a one-sided manner

$$p_{bv}(u\underline{v}(\mathbf{d})) = \int_{-\infty} u(\mathbf{d})v(\mathbf{d}) \rho^0_{U\underline{V}}(u\underline{v}) \, du dv = \frac{1}{m} \sum_{i=1 j=1}^{m} \Theta\left( u\underline{v}(\mathbf{d}) - {}_i\tilde{v}_j \right), \tag{32}$$

where the right hand side equals the fraction of  resampled $\tilde{v}_j$ that are smaller than  $v(\mathbf{d})$ of the original sample .

In the case where the sample of differences is uncertain, as for  $\mathbf{\Delta \hat{T}} = (\Delta \hat{T}_1, ..., \Delta \hat{T}_n)$, the construction scheme for  $\rho_{\mathcal{V}}^0$ needs to be adjusted to  reflect these uncertainties.

$$\rho_U(u') = \int \delta\left( u' - \frac{1}{n} \sum_{i=1}^{n} \Delta t_i \right) \rho_{\mathbf{\Delta T}}(\mathbf{\Delta t}) \, d\mathbf{\Delta t}.$$

~~The shifted sample $\mathbf{\Delta T}^* = \mathbf{\Delta T} - U(\mathbf{\Delta T})$ can still be defined equivalently, but is now comprised of uncertain quantities. In turn, the bootstrapped synthetic samples will as well be comprised of uncertain individuals, which results is uncertain synthetic means $\tilde{U}_i$ whose distributions $\rho_{\tilde{U}_i}(\tilde{u}_i)$ can be derived in the sense of Eq. 15.~~

 In principle, each possible value $\mathbf{\Delta \hat{t}}$ for the uncertain $\mathbf{\Delta \hat{T}}$ is associated with its own null distribution $\rho_{\mathcal{V}}^0(v, \mathbf{\Delta \hat{t}})$. In this sense, the value for the test statistic

$$\rho_U^0(u, \tilde{u}_1, ..., \tilde{u}_m) = \frac{1}{m} \sum_{i=1}^{m} \delta(u - \tilde{u}_i) \rho_{\tilde{U}_i}(\tilde{u}_i).$$

$$\rho_U^0(u) = \int \rho_U^0(u, \mathbf{\tilde{u}}) \, d\mathbf{\tilde{u}} = \frac{1}{m} \sum_{i=1}^{m} \rho_{\tilde{U}_i}(u).$$

 $v(\mathbf{\Delta t})$ should be compared to the corresponding $\rho_{\mathcal{V}}^0(v, \mathbf{\Delta \hat{t}})$ to derive a $p$-value for this $\mathbf{\Delta \hat{t}}$. Eqs. 31 and 32 define a mapping from $\mathbf{\Delta \hat{t}}$ to its corresponding $p$-value. To compute the uncertainty

distribution for the $p$-
 -value, this map has to be evaluated for all potential $\Delta\hat{\mathbf{t}}$, weighted by the uncertainty distribution $\varrho_{\Delta\hat{\mathbf{T}}}(\Delta\hat{\mathbf{t}})$:

$$\rho_{P_b\hat{P}_v}(p_b\hat{p}_v) = \int \delta \int_{-\infty}^{u} \rho_U^0(u'\hat{p}_v - p_v(\Delta\hat{\mathbf{t}})) \, du' - p_b \rho_{U\Delta\hat{\mathbf{T}}}(u\Delta\hat{\mathbf{t}}) \, du \, d\Delta\hat{\mathbf{t}}. \tag{33}$$

The three tests are applied in combination in order to compensate their individual deficits. If the population $\mathcal{P}_{\Delta T}$ was truly Gaussian, the $t$-test would be the most powerful test, i.e., its rejection region would be the largest across all tests on the population mean (Lehmann and Romano, 2006). Since normality of $\mathcal{P}_{\Delta T}$ cannot be guaranteed, the less powerful Wilcoxon-signed-rank test constitutes a meaningful supplement to the $t$-test, relying on the somewhat weaker assumption that $\mathcal{P}_{\Delta T}$ is symmetric around zero. Finally, the bootstrap test is non-parametric and in view of its independence from any assumptions adds a valuable contribution.

**3.4 Comparison to the 'combined evidence' reported by Erhardt et al. (2019)**

For the derivation of the transition lag uncertainty distributions $\rho_{\Delta T_i^{p,q}}(\Delta t_i^{p,q})$ $\varrho_{\Delta\hat{T}_i^{p,q}}(\Delta\hat{t}_i^{p,q})$ of the i-th DO event between the proxies $p$ and $q$, we have directly adopted the methodology designed by Erhardt et al. (2019). However, our statistical interpretation of the resulting sets of uncertainty distributions $\{\rho_{\Delta t_1}^{p,q}(\Delta t_1), \ldots, \rho_{\Delta t_n}^{p,q}(\Delta t_n)\}$ $\{\varrho_{\Delta\hat{T}_1}^{p,q}(\Delta\hat{t}_1), \ldots, \varrho_{\Delta\hat{T}_n}^{p,q}(\Delta\hat{t}_n)\}$ derived from the set of DO events differs from the one proposed by Erhardt et al. (2019). In this section we explain the subtle yet important differences between the two statistical perspectives.

Given a pair of variables $(p,q)$, Erhardt et al. (2019) define what they call 'combined estimate' $\rho_{\Delta T}^*(\Delta t)$ $\varrho_{\Delta T^*}(\Delta t^*)$ as the product over all corresponding lag uncertainty distributions:

$$\rho_{\Delta T}^* \varrho_{\Delta T^*}(\Delta t^*) \propto \prod_{i=1}^{n} \rho_{\Delta T_i \Delta\hat{T}_i}(\Delta t^*). \tag{34}$$

This implicitly assumes that all DO events share the exact same time lag $\Delta t$ $\Delta t^*$ between the variables $p$ and $q$. This is realized by inserting a single argument $\Delta t$ $\Delta t^*$ into the different distributions $\rho_{\Delta T_i}(\cdot)$ $\varrho_{\Delta\hat{T}_i}(\cdot)$. Hence, the product on the right hand side of Eq. 34 in fact indicates the probability that all DO events assume the time lag $\Delta t$ $\Delta t^*$, provided that they all assume the same lag:

$$\rho_{\Delta T}^* \varrho_{\Delta T^*}(\Delta t^*) = \rho_{\Delta T}^* \varrho_{\Delta T^*}(\Delta t^*|\Delta t \hat{t}_1 = \ldots = \Delta t \hat{t}_n = \Delta t^*) = \frac{\prod \rho_{\Delta T_i}(\Delta t)}{\int_{\Omega} \prod \rho_{\Delta T_i}(\Delta t_i) \, d\Delta t_i} \frac{\prod \rho_{\Delta\hat{T}_i}(\Delta t^*)}{\int_{\Omega} \prod \rho_{\Delta\hat{T}_i}(\Delta\hat{t}_i) \, d\Delta\hat{t}_i}, \quad \Omega = \{\Delta\mathbf{t}\Delta\hat{\mathbf{t}} : \Delta t \hat{t}_i = \Delta \tag{35}$$

The denominator on the right hand side equals the probability that all DO events share a common time lag. Eq. 34 strongly emphasizes those regions where all uncertainty distributions $\rho_{\Delta T_i}(\Delta t_i)$ are $\varrho_{\Delta\hat{T}_i}(\Delta\hat{t}_i)$ are simultaneously substantially larger than zero. The 'combined evidence' answers the question: Provided that all DO events exhibit the same lag between the

transition onsets of $p$ and $q$, then how likely is it that this lag is given by $\Delta t$ $\Delta t^*$. Drawing on this quantity, (Erhardt et al., 2019) conclude that $\delta^{18}O$ and $Na^+$ 'on average' lag $Ca^{2+}$ and $\lambda$ by about one decade.

Thinking of the DO transition onset lags as i.i.d. random variables of a repeatedly executed random experiment takes into account the natural variability between different DO events and hence, it removes the restricting a priori assumption

490    $\Delta t_1 = ... = \Delta t_n$ $\Delta \hat{t}_1 = ... = \Delta \hat{t}_n$. In our approach we have related the potentially systematic character of lags to the population mean. Since the sample mean is the best point-estimate of a population mean, we consider it to reasonably indicate potential leads and lags, whose significance should be tested in a second step. Thus, we ascribe the sample mean a similar role as Erhardt et al. (2019) ascribe to the 'combined estimate' and therefore, we present a comparison of these two quantities in Sec. 4.1.

495    The mean of an uncertain sample $\hat{U} = u(\Delta \hat{T})$ is again an uncertain quantity and its distribution  reads

$$\rho_{\hat{U}}(\hat{u}) = \int \delta(\hat{u} - u(\Delta \hat{t})) \rho_{\Delta \hat{T}}(\Delta \hat{t}) \, d\Delta \hat{t}. \tag{36}$$

[revised manuscript text omitted]

 We emphasize that our results should not be misunderstood as evidence against the alternative hypothesis of a systematic lag. In the presence of a systematic lag ($\mu < 0$) the ability of hypothesis tests to reject the null hypothesis of no systematic lag ($(H_0 : \mu = 0)$) depends on the sample size $n$, the ratio between the mean lag $|\mu|$, the variance of the population, and on the precision of the measurement. Neither of these quantities is favourable in our case and thus, it is certainly possible that the ~~different processes involved from one DO event to the next. Fig. 2 clearly shows that the different events exhibit different time lags. Provided that the DO events were driven by the same process, physically they constitute different realizations and they exhibit great variability also in other variables such as the amplitude of the temperature change (Kindler et al., 2014) or the waiting times with respect to the previous event (Ditlevsen et al., 2007; Boers et al., 2018). The random experiment framework presented here moreover relates potential systematic leads and lags in the physical process that drives DO events to a bias in the corresponding population of lags between proxy variables. This allows for the physically meaningful formulation of a statistical hypothesis and a corresponding null hypothesis . By applying different hypothesis tests we have followed a well-established line of statistical inference~~null hypothesis cannot be rejected despite the alternative being true.

Our main purpose was the consistent treatment of observational uncertainties and we have largely ignored the vibrant debate on the qualitative interpretation of the proxies. Surprisingly, we could not find any literature on the application of hypothesis tests to uncertain samples of the kind discussed here. The theory of fuzzy $p$-values is  concerned with uncertainties either in the data or in the hypothesis.  however, not applicable to measurement uncertainties that are quantifiable in terms of probability density functions (Filzmoser and Viertl, 2004). We have proposed to propagate the uncertainties to the level of the $p$-values and to then consider the expected $p$-values and the share of $p$-values which indicate significance, in order to decide between rejection and acceptance. The $p$-value measures the extremeness of a sample with respect to the null distribution and we hence regard the expected $p$-value to be a suitable measure for the uncertain samples' extremeness.

The probability of the uncertain sample to be significant at a given level is also a reasonable indicator, which can be invoked in addition. In cases of high cost of a wrongly rejected null hypothesis, one might want to have a high degree of certainty that the uncertain sample actually contradicts the null hypothesis and hence a high probability for the uncertain $p$-value to be smaller than $\alpha$. In contrast, if the observational uncertainties are averaged out beforehand, crucial information is lost. The expected sample may either be significant or not, but the uncertainty about the significance can no longer be accurately quantified.

The potential of the availability of data from different sites has probably not been  fully leveraged in this study. Naively, one could think of the NEEM and NGRIP $(Ca^{2+}, Na^+)$ lag records as two independent observations of the same entity. However, the discrepancy in the corresponding sample mean uncertainty distributions  question how changes in

the climatic features such as sea-ice cover and atmospheric circulation are actually recorded by the proxies at different sites, and how important regional geographic differences are. Proxy-enabled modeling studies as presented by Sime et al. (2019) could shed further light on the question to what extent the NEEM and NGRIP sites record the same signal after an abrupt change of the climatic conditions. Also, a comparison of the NGRIP and NEEM records on an individual event level could

675 provide further insights how to combine these records statistically.

~~Alternatively to computing the sample mean or the 'combined estimate', it may seem attractive to simply 'add' the distributions $\rho_{\Delta T_i^{p,q}}(\Delta t_i^{p,q}$ obtained from the different events for one pair of proxies. This impression may be supported by the fact that the practical computation is carried out in terms of empirical densities $\bar{\rho}_{\Delta T_i}(\Delta t_i) = \frac{1}{m}\sum_{j=1}^{m}\delta(\Delta t_i - \Delta t_{i,j})$, comprised of 6000 values $\{\Delta t_{i,j}\}$ obtained via MCMC sampling. Given these $n$ times $m$ values, with $n$ denoting the number of DO events for~~

680 ~~the proxy pair and $m$ the number of MCMC sampled values for each pair, one might be tempted to pool them all together and call them a somewhat combined probability density estimate. In terms of continuous uncertainty densities, this pooling corresponds to averaging over $\rho_{\Delta T_i}(\Delta t_i)$ with respect to the index $i$. This would ignore the fact that the different uncertainty distributions aim to represent different quantities, namely the transition lags from physically different DO event realizations. Most importantly, the average over the $\rho_{\Delta T_i}(\Delta t_i)$ or alternatively the combined set of MCMC samples does not correspond to~~

[revised manuscript text omitted]

$$\tilde{\rho}^{p,q}_{\Delta\hat{\mathbf{T}}^{p,q}}(\Delta\hat{\mathbf{t}}^{p,q}) = \frac{1}{m} \sum_{j=1}^{m} \prod_{i=1}^{n} \delta(\Delta\hat{t}^{p,q}_i - \Delta t^{p,q}_{i,j}) = \frac{1}{m} \sum_{j=1}^{m} \delta(\Delta\hat{\mathbf{t}}^{p,q} - \Delta\mathbf{t}^{p,q}_j) \tag{A8}$$

constrains the set that determines $\tilde{\rho}_{\Delta\hat{\mathbf{T}}^{p,q}}(\Delta\hat{\mathbf{t}}^{p,q})$ to $m$ values, where those values from different DO events with the same MCMC index $j$ are combined:

$$\Delta\mathbf{t}^{p,q}_j = (\Delta t^{p,q}_{1,j},...,\Delta t^{p,q}_{n,j}). \tag{A9}$$

Again, the validity is checked by randomly permuting the sets $\{\Delta t^{p,q}_{i,j}\}$ for the individual DO events with respect to the index $j$ before the set reduction in the control runs.

Having found a numerically manageable expression for the empirical uncertainty distribution of the sample $\Delta\hat{\mathbf{T}}^{p,q}$ it remains to be shown how the hypothesis tests can be formulated on this basis. If $\{\Delta\mathbf{t}_j\}_j$ denotes the set of $n$ dimensional vectors forming the empirical uncertainty distribution for the sample of lags obtained from $n$ DO events, then the naive intuition

holds true and the corresponding set $\{\phi_j = \phi(\Delta \mathbf{t}_j)\}_j$ represents the empirical uncertainty distribution of the test statistic and correspondingly $\{p_\phi(\phi_j)\}_j$ characterizes the uncertain $p$-value. In the following, we examplarily derive this relation for the $t$-test - the derivations for the WSR and the bootstrap test are analogue.

Recall the statistic of the $t$-test

930
$$z(\mathbf{d}) = \frac{u(\mathbf{d}) - \mu}{s(\mathbf{d})/\sqrt{(n)}}. \tag{A10}$$

The empirical uncertainty distribution for a sample $\Delta \mathbf{T}$ $\Delta \hat{\mathbf{T}}$ induces a joint uncertainty distribution for the samples mean and standard deviation

$$\bar{\rho}_{U,S\,\hat{U},\hat{S}}(u\hat{u}, s\hat{s}) = \int \delta\left(u - \frac{1}{n}\sum_{i=1}^{n}\Delta \underline{t}\hat{t}_i\right) \delta\left(s - \frac{1}{n-1}\sum_{i=1}^{n}(u - \Delta \underline{t}\hat{t}_i)^2\right) \frac{1}{m}\sum_{j=1}^{m}\prod_{i=1}^{n}\delta(\Delta \underline{t}\hat{t}_i - \Delta t_{i,j})\, d\Delta \underline{t}\hat{t}_1 \dots d\Delta \underline{t}\hat{t}_n$$

935
$$= \frac{1}{m}\sum_{j=1}^{m}\delta\left(\underline{u}\hat{u} - \frac{1}{n}\sum_{i=1}^{n}\Delta t_{i,j}\right)\delta\left(\underline{s}\hat{s} - \frac{1}{n-1}\sum_{i=1}^{n}(\underline{u}\hat{u} - \Delta t_{i,j})^2\right). \tag{A11}$$

Let $u_j = \frac{1}{n}\sum_{i=1}^{n}\Delta t_{i,j}$, and $s_j = \frac{1}{n-1}\sum_{i=1}^{n}(u_j - \Delta t_{i,j})^2$. Then, the empirical uncertainty distribution for $(U,S)$ $(\hat{U},\hat{S})$ can be written as

$$\bar{\rho}_{U,S\,\hat{U},\hat{S}}(u\hat{u}, s\hat{s}) = \frac{1}{m}\sum_{j=1}^{m}\delta\left(u - u\hat{u} - u_j\right)\delta\left(s - s\hat{s} - s_j\right) \tag{A12}$$

The $(u_j, s_j)$ that form the empirical uncertainty distribution are simply the mean and standard deviation of those $\Delta \mathbf{t_j} = (\Delta t_{1,j}, \Delta t_{2,j}, \dots, \Delta t$ $\Delta \mathbf{t}_j = (\Delta t_{1,j}, \Delta t_{2,j}, \dots, \Delta t_{n,j})$
940 that form the vector valued empirical uncertainty distribution for $\Delta \mathbf{T}$. From $\bar{\rho}_{U,S}(u,s)$ $\Delta \hat{\mathbf{T}}$. From $\bar{\rho}_{\hat{U},\hat{S}}(\hat{u},\hat{s})$, the empirical uncertainty distribution for the test statistic $Z$ uncertain test statistic $\hat{Z}$ can be computed as follows:

$$\bar{\rho}_{Z\,\hat{Z}}(z\hat{z}) = \int \delta\left(z - \frac{u - \mu}{s/\sqrt{(n)}}\,\hat{z} - \frac{\hat{u} - \mu}{\hat{s}/\sqrt{(n)}}\right)\rho_{U,S\,\hat{U},\hat{S}}(u\hat{u}, s\hat{s})\, du\,d\hat{u}\, ds\,d\hat{s} = \frac{1}{m}\sum_{j=1}^{m}\delta\left(z - \hat{z} - \underbrace{\frac{u_j - \mu}{s_j/\sqrt{(n)}}}_{=z_j}\right). \tag{A13}$$

This shows, that for a given empirical uncertainty distribution for a sample of time lags $\bar{\rho}_{\Delta \mathbf{T}}(\Delta \mathbf{t}) = \frac{1}{m}\sum_{i=1}^{m}\delta(\Delta \mathbf{t} - \Delta \mathbf{t}_j)$ $\bar{\rho}_{\Delta \hat{\mathbf{T}}}(\Delta \hat{\mathbf{t}}) = \frac{1}{m}$
945 the corresponding distribution for the test statistic $Z(\Delta \mathbf{T})$ $\hat{Z} = z(\Delta \hat{\mathbf{T}})$ is formed by the set $\{z(\Delta \mathbf{t}_j)|j \in [1,m]\}$ where each $\Delta \mathbf{t}_j$ is a vector in $n$ dimensions. The uncertain (left-handed) $p$-value remains to be derived from $\bar{\rho}_Z(z)$ $\bar{\rho}_{\hat{Z}}(\hat{z})$:

$$\bar{\rho}_{P_z\,\hat{P}_z}(p\hat{p}_z) = \int \delta\left(p\hat{p}_z - \int_{-\infty}^{z\hat{z}} t_{n-1}(z')\, dz'\right)\bar{\rho}_{Z\,\hat{Z}}(z\hat{z})\, dz\,d\hat{z} \qquad = \frac{1}{m}\sum_{j=1}^{m}\delta\left(p\hat{p}_z - \underbrace{\int_{-\infty}^{z_j} t_{n-1}(z')\, dz'}_{=p_{z,j}}\right). \tag{A14}$$

Finally, the practical computation of the uncertain $p$-values boils down to an application of the test to all members of the set $\Delta \mathbf{t}_j$ that originates from the MCMC sampling used to approximate the posterior probability density for the ramp parameter

configuration $\Theta$. For the WSR test the expression

$$\bar{\rho}_{P_w \hat{P}_w}(p\hat{p}_w) = \frac{1}{m}\sum_{j=1}^{m}\delta\left(p\hat{p}_w - p_{w,j}\right) \quad \text{with} \quad p_{w,j} = p_w(\boldsymbol{\Delta}\mathbf{t}_j) \tag{A15}$$

can be derived analogously. The bootstrap test bears the particularity  that each $\boldsymbol{\Delta}\mathbf{t}_j$ induces its own null distribution.  Yet, the application of the  test to each individual $\boldsymbol{\Delta}\mathbf{t}_j$ induces a set of $p_{v,j} = p_v(\boldsymbol{\Delta}\mathbf{t}_j)$ that determines the empirical density

$$\bar{\rho}_{\hat{P}_v}(\hat{p}_v) = \frac{1}{m}\sum_{j=1}^{m}\delta\left(\hat{p}_v - p_{v,j}\right). \tag{A16}$$

**Appendix B: Results of the analysis for the control group**

As explained in Sec. A, we drastically reduce the cardinality of the sets that form the empirical densities  $\bar{\rho}_{\boldsymbol{\Delta}\hat{\mathbf{T}}^{p,q}}(\boldsymbol{\Delta}\hat{\mathbf{t}}^{p,q})$ at two points in the analysis. First, for the representation of the uncertain time lag  $\Delta\hat{T}_i^{p,q}$ between the proxies $p$ and $q$ at a given DO event, only 6000 out of the $6000^2$ possible values are utilized. Second, the set of vectors considered in the representation of  $\bar{\rho}_{\boldsymbol{\Delta}\hat{\mathbf{T}}^{p,q}}(\boldsymbol{\Delta}\hat{\mathbf{t}}) = \frac{1}{6000}\sum_{j=1}^{6000}\delta(\boldsymbol{\Delta}\hat{\mathbf{t}}^{p,q} - \boldsymbol{\Delta}\mathbf{t}_j^{p,q})$ is comprised of only 6000 out of the $6000^{16}$ theoretically available vectors. To cross-check the robustness of the results obtained within the limits of this approximation, we applied our analysis to a control group of 9 alternative realizations of the empirical uncertainty density for  $\boldsymbol{\Delta}\hat{\mathbf{T}}^{p,q}$ for each proxy pair. The control group uncertainty densities are constructed as follows: First, the empirical uncertainty distributions for the event specific lags  $\Delta\hat{T}_i^{p,q}$ are obtained via Eq. A6. In a second step, the joint empirical uncertainty distribution for  $\boldsymbol{\Delta}\hat{\mathbf{T}}^{p,q}$ is constructed from randomly shuffled empirical sets $\Delta t_{i,s_i(j)}^{p,q}$ of each DO event:

$$\tilde{\rho}^{p,q,ctrl}_{\boldsymbol{\Delta}\mathbf{T}\ \boldsymbol{\Delta}\hat{\mathbf{T}}^{p,q}}(\boldsymbol{\Delta}\mathbf{t}\boldsymbol{\Delta}\hat{\mathbf{t}}^{p,q}) = \frac{1}{m}\sum_{j=1}^{m}\prod_{i=1}^{n}\delta(\Delta t\hat{t}_i^{p,q} - \Delta t_{i,s_i(j)}^{p,q}). \tag{B1}$$

Here $s_i$ denotes an event specific permutation of the index set $\{1, ..., 6000\}$. Thus the empirical $\Delta t_{i,j}^{p,q}$ recombine between events and give rise to a new set of 6000 vectors that constitute 6000 empirical realizations of the uncertain  $\boldsymbol{\Delta}\hat{\mathbf{T}}^{p,q}$.

The results obtained from the control runs show only minor deviations from the results presented in the main text and thus confirm the validity of the reduction of the corresponding sets. Tab. B1 summarizes the results obtained by application of the hypothesis tests to the control group.

**Table B1.** Results obtained from the application of hypothesis tests to the control group. Reported are the mean $p$-values  $E(p(\mathbf{\Delta \hat{T}}))$ together with the probability of the uncertain sample to be smaller than the significance level  $\pi(p(\mathbf{\Delta \hat{T}}) < 0.05)$ and the $p$-values of the expected samples  $p(E(\mathbf{\Delta \hat{T}}))$ for all three tests. All results were derived from the corresponding empirical densities  $\bar{\ell}_{\mathbf{\Delta \hat{T}}_{R,q}}(\mathbf{\Delta \hat{t}}^{p,q})$. The results from the original analysis are given as well by the $p-q-0$ run for each proxy variable.

| proxies | run | E($\hat{P}$) | | | $\pi(\hat{P} < 0.05)$ | | | p( |
| --- | --- | --- | --- | --- | --- | --- | --- | --- |
| | | z | w | bs | z | w | bs | z |
| NGRIP:Ca$^{2+}$-Na$^+$- | 0 | 0.219 | 0.168 | 0.217 | 0.258 | 0.324 | 0.299 | 0.044 |
| NGRIP:Ca$^{2+}$-Na$^+$- | 1 | 0.218 | 0.166 | 0.215 | 0.246 | 0.316 | 0.292 | 0.044 |
| NGRIP:Ca$^{2+}$-Na$^+$- | 2 | 0.219 | 0.165 | 0.216 | 0.258 | 0.324 | 0.294 | 0.044 |
| NGRIP:Ca$^{2+}$-Na$^+$- | 3 | 0.22 | 0.166 | 0.217 | 0.254 | 0.322 | 0.295 | 0.044 |
| NGRIP:Ca$^{2+}$-Na$^+$- | 4 | 0.219 | 0.166 | 0.217 | 0.255 | 0.32 | 0.296 | 0.044 |
| NGRIP:Ca$^{2+}$-Na$^+$- | 5 | 0.218 | 0.166 | 0.216 | 0.254 | 0.319 | 0.293 | 0.044 |
| NGRIP:Ca$^{2+}$-Na$^+$-0.218 | 6 | 0.219 | 0.167 | 0.217 | 0.255 | 0.319 | 0.299 | 0.044 |
| NGRIP:Ca$^{2+}$-Na$^+$- | 7 | 0.219 | 0.167 | 0.217 | 0.252 | 0.319 | 0.295 | 0.044 |
| NGRIP:Ca$^{2+}$-Na$^+$- | 8 | 0.219 | 0.164 | 0.217 | 0.257 | 0.316 | 0.3 | 0.044 |
| NGRIP:Ca$^{2+}$-Na$^+$- | 9 | 0.218 | 0.165 | 0.216 | 0.261 | 0.32 | 0.302 | 0.044 |
| NGRIP:$\lambda$-$\delta^{18}$O-Na$^+$ | 0 | 0.093 | 0.091 | 0.086 | 0.469 | 0.484 | 0.524 | 0.023 |
| NGRIP:$\lambda$-$\delta^{18}$O-Na$^+$ | 1 | 0.092 | 0.091 | 0.085 | 0.467 | 0.482 | 0.516 | 0.023 |
| NGRIP:$\lambda$-$\delta^{18}$O-Na$^+$ | 2 | 0.093 | 0.092 | 0.086 | 0.462 | 0.489 | 0.519 | 0.023 |
| NGRIP:$\lambda$-$\delta^{18}$O-Na$^+$ | 3 | 0.092 | 0.09 | 0.085 | 0.465 | 0.482 | 0.516 | 0.023 |
| NGRIP:$\lambda$-$\delta^{18}$O-Na$^+$ | 4 | 0.093 | 0.09 | 0.086 | 0.471 | 0.488 | 0.529 | 0.023 |
| NGRIP:$\lambda$-$\delta^{18}$O-Na$^+$ | 5 | 0.093 | 0.092 | 0.086 | 0.468 | 0.492 | 0.522 | 0.023 |
| NGRIP:$\lambda$-$\delta^{18}$O-Na$^+$ | 6 | 0.092 | 0.089 | 0.085 | 0.47 | 0.488 | 0.521 | 0.023 |
| NGRIP:$\lambda$-$\delta^{18}$O-Na$^+$ | 7 | 0.092 | 0.091 | 0.085 | 0.461 | 0.486 | 0.515 | 0.023 |
| NGRIP:$\lambda$-$\delta^{18}$O-Na$^+$ | 8 | 0.093 | 0.091 | 0.086 | 0.477 | 0.486 | 0.525 | 0.023 |
| NGRIP:$\lambda$-$\delta^{18}$O-Na$^+$ | 9 | 0.093 | 0.091 | 0.086 | 0.475 | 0.488 | 0.524 | 0.023 |
| NGRIP:Ca$^{2+}$-$\delta^{18}$O-$\delta^{18}$O | 0 | 0.234 | 0.182 | 0.233 | 0.235 | 0.306 | 0.262 | 0.042 |
| NGRIP:Ca$^{2+}$-$\delta^{18}$O-$\delta^{18}$O | 1 | 0.234 | 0.182 | 0.232 | 0.231 | 0.294 | 0.257 | 0.042 |
| NGRIP:Ca$^{2+}$-$\delta^{18}$O-0.233$\delta^{18}$O | 2 | 0.234 | 0.18 | 0.232 | 0.226 | 0.3 | 0.254 | 0.042 |
| NGRIP:Ca$^{2+}$-$\delta^{18}$O-$\delta^{18}$O | 3 | 0.234 | 0.182 | 0.233 | 0.236 | 0.314 | 0.261 | 0.042 |
| NGRIP:Ca$^{2+}$-$\delta^{18}$O-$\delta^{18}$O | 4 | 0.234 | 0.181 | 0.232 | 0.234 | 0.308 | 0.261 | 0.042 |
| NGRIP:Ca$^{2+}$-$\delta^{18}$O-$\delta^{18}$O | 5 | 0.233 | 0.181 | 0.231 | 0.23 | 0.304 | 0.253 | 0.042 |
| NGRIP:Ca$^{2+}$-$\delta^{18}$O-$\delta^{18}$O | 6 | 0.234 | 0.181 | 0.232 | 0.228 | 0.306 | 0.253 | 0.042 |
| NGRIP:Ca$^{2+}$-$\delta^{18}$O-$\delta^{18}$O | 7 | 0.234 | 0.18 | 0.232 | 0.235 | 0.31 | 0.261 | 0.042 |
| NGRIP:Ca$^{2+}$-$\delta^{18}$O-$\delta^{18}$O | 8 | 0.234 | 0.183 | 0.232 | 0.236 | 0.313 | 0.263 | 0.042 |
| NGRIP:Ca$^{2+}$-$\delta^{18}$O-0.233$\delta^{18}$O | 9 | 0.234 | 0.182 | 0.232 | 0.231 | 0.307 | 0.257 | 0.042 |

**Table B2.** Continuation of Tab.B1.

| proxies | run | E($\hat{P}$) | | | $\pi(\hat{P} < 0.05)$ | | | $p(\mathrm{E}(\mathbf{\Delta\hat{T}}))$ | |
|---|---|---|---|---|---|---|---|---|---|
| | | z | w | bs | z | w | bs | z | w |
| NGRIP:$\lambda$--$\delta^{18}$O | 0 | 0.133 | 0.11 | 0.129 | 0.369 | 0.436 | 0.414 | 0.024 | 0.009 |
| NGRIP:$\lambda$--$\delta^{18}$O | 1 | 0.134 | 0.111 | 0.129 | 0.37 | 0.441 | 0.416 | 0.024 | 0.009 |
| NGRIP:$\lambda$--$\delta^{18}$O | 2 | 0.133 | 0.11 | 0.127 | 0.379 | 0.44 | 0.422 | 0.024 | 0.009 |
| NGRIP:$\lambda$--$\delta^{18}$O | 3 | 0.135 | 0.112 | 0.13 | 0.38 | 0.435 | 0.42 | 0.024 | 0.009 |
| NGRIP:$\lambda$--$\delta^{18}$O | 4 | 0.134 | 0.111 | 0.129 | 0.378 | 0.442 | 0.419 | 0.024 | 0.009 |
| NGRIP:$\lambda$--$\delta^{18}$O | 5 | 0.133 | 0.109 | 0.128 | 0.373 | 0.437 | 0.416 | 0.024 | 0.009 |
| NGRIP:$\lambda$--$\delta^{18}$O | 6 | 0.133 | 0.111 | 0.128 | 0.384 | 0.446 | 0.426 | 0.024 | 0.009 |
| NGRIP:$\lambda$--$\delta^{18}$O | 7 | 0.133 | 0.109 | 0.128 | 0.376 | 0.445 | 0.416 | 0.024 | 0.009 |
| NGRIP:$\lambda$--$\delta^{18}$O | 8 | 0.134 | 0.11 | 0.129 | 0.381 | 0.443 | 0.424 | 0.024 | 0.009 |
| NGRIP:$\lambda$--$\delta^{18}$O | 9 | 0.134 | 0.11 | 0.129 | 0.376 | 0.441 | 0.418 | 0.024 | 0.009 |
| NEEM:Ca$^{2+}$-Na$^+$- | 0 | 0.08 | 0.076 | 0.074 | 0.566 | 0.584 | 0.61 | 0.008 | 0.007 |
| NEEM:Ca$^{2+}$-Na$^+$- | 1 | 0.08 | 0.076 | 0.074 | 0.57 | 0.581 | 0.61 | 0.008 | 0.007 |
| NEEM:Ca$^{2+}$-Na$^+$- | 2 | 0.079 | 0.075 | 0.073 | 0.571 | 0.587 | 0.614 | 0.008 | 0.007 |
| NEEM:Ca$^{2+}$-Na$^+$- | 3 | 0.079 | 0.076 | 0.073 | 0.573 | 0.586 | 0.615 | 0.008 | 0.007 |
| NEEM:Ca$^{2+}$-Na$^+$- | 4 | 0.08 | 0.077 | 0.074 | 0.572 | 0.584 | 0.612 | 0.008 | 0.007 |
| NEEM:Ca$^{2+}$-Na$^+$- | 5 | 0.08 | 0.076 | 0.074 | 0.571 | 0.579 | 0.608 | 0.008 | 0.007 |
| NEEM:Ca$^{2+}$-Na$^+$- | 6 | 0.08 | 0.077 | 0.074 | 0.565 | 0.577 | 0.609 | 0.008 | 0.007 |
| NEEM:Ca$^{2+}$-Na$^+$- | 7 | 0.08 | 0.077 | 0.074 | 0.57 | 0.583 | 0.612 | 0.008 | 0.007 |
| NEEM:Ca$^{2+}$-Na$^+$- | 8 | 0.079 | 0.075 | 0.073 | 0.57 | 0.58 | 0.614 | 0.008 | 0.007 |
| NEEM:Ca$^{2+}$-Na$^+$- | 9 | 0.078 | 0.075 | 0.072 | 0.567 | 0.576 | 0.608 | 0.008 | 0.007 |

**Appendix C: Computation of the uncertain sample mean**

In the main text, we stated that the uncertain sample mean is given by the pairwise convolution of the individual uncertainty distributions that describe the uncertain sample members. Here, we show how the uncertain sample mean can be computed if the individual uncertainty distributions are known.

Consider $n$ random variables which are independently, yet not identically distributed:

$$\mathbf{X} = (X_1, ..., X_n) \quad \text{with} \quad X_i \sim \rho_{X_i}(x_i)\, dx_i. \tag{C1}$$

in analogy to the

$$\mathbf{\Delta\hat{T}}^{p,q} = (\Delta\hat{T}_1^{p,q}, ..., \Delta\hat{T}_n^{p,q}) \quad \text{with} \quad \Delta\hat{T}_i^{p,q} \sim \rho_{\Delta\hat{T}_i^{p,q}}(\Delta\hat{t}_i^{p,q}) \tag{C2}$$

 Further, let

$$U = \frac{1}{n} \sum_{i=1}^{n} X_i \tag{C3}$$

denote the mean of the sample of random variables, which is in turn a random variable by itself. In order to compute the distribution $\rho_U(u)\, du$ we introduce the variable $V = nU$ and the sequence of variables

$$V_j = \sum_{i=1}^{j} X_i, \tag{C4}$$

such that $V_n = V$. From C4 it follows that

$$V_{j+1} = V_j + X_{j+1} \tag{C5}$$

and hence

$$\rho_{V_{j+1}}(v_{j+1})\, dv_{j+1} = \int_{-\infty}^{\infty} \int_{-\infty}^{\infty} \rho_{V_j}(v_j)\, \rho_{X_{j+1}}(x_{j+1})\, \delta(v_{j+1} - v_j - x_{j+1})\, dx_{j+1}\, dv_j\, dv_{j+1}$$

$$= \int_{-\infty}^{\infty} \rho_{V_j}(v_j)\, \rho_{X_{j+1}}(v_{j+1} - v_j)\, dv_j\, dv_{j+1}. \tag{C6}$$

Self-iteration of C6 yields

$$\rho_{V_{j+1}}(v_{j+1})\, dv_{j+1}$$

$$= \int_{-\infty}^{\infty} \int_{-\infty}^{\infty} \underbrace{\rho_{V_{j-1}}(v_{j-1})\, \rho_{X_j}(v_j - v_{j-1})\, dv_{j-1}}_{=\rho_{V_j}(v_j)}\, \rho_{X_{j+1}}(v_{j+1} - v_j)\, dv_j\, dv_{j+1}$$

$$= \ldots$$

$$= \int_{-\infty}^{\infty} \ldots \int_{-\infty}^{\infty} \prod_{i=1}^{j+1} \rho_{X_i}(v_i - v_{i-1})\, dv_{i-1}\, dv_{j+1}, \tag{C7}$$

where $v_0 = 0$. With $V_n/n = U$ the distribution for the uncertain sample mean reads

$$\rho_{V_n}(v_n)\, dV_n = \rho_{V_n}(nu)\, n\, du = \rho_U(u)\, du \tag{C8}$$

and thus

$$\rho_U(u)\, du = \int_{-\infty}^{\infty} \prod_{i=1}^{n} \rho_{X_i}(v_i - v_{i-1})\, dv_{i-1}\, n\, du, \tag{C9}$$

with $v_0 = 0$ and $v_n = nu$.

---

## Author Response (AR2)

**Author's Response**

July 1, 2021

**Contents**

**1 Answer to Referee 1**

**1.1 General Remarks**

First of all, we would like to thank the referee for the careful second review of our manuscript. Before we address the comments point by point, we will discuss the main criticism, namely the decision criteria we use to reject or not reject the respective null-hypotheses.

We would like to emphasize that we acknowledge the arguments brought up against our decision criteria and that after a careful review we have refined the explanation of the criteria based on the referees input. Here, for sake of clarity, we will first summarize the status of the discussion. Subsequently, we repeat the argument presented by referee 1 in favour of the criterion (1). We will then examine this argument in detail and explain why we are still convinced that the combination of the criteria (2) and (3) yields a more meaningful assessment than criterion (1). Nonetheless, due to the comments of the referee, with respect to the decision criteria we changed the manuscript as follows:

- We added a paragraph that motivates the propagation of uncertainties with an example, where averaging out uncertainties yields undesired results. (l.282ff)

- We added two sentences on the interpretation of the criteria (2) and(3): 'Given the measurement uncertainty the quantity $\pi(\hat{P} < \alpha)$ indicates the informed estimate of the observer that the true value of the measured sample is in fact statistically significant with respect to the null hypothesis. Thus, the first criterion assesses how 'strongly' the uncertain sample contradicts the null hypothesis and the second criterion evaluates the likelihood of the uncertain sample to contradict the null hypothesis. Depending 315 on $\eta$, in many cases both criteria will yield the same decision. If not, the specific situation determines which of the criteria is more convenient.' (l.311)

- In the previous version of the manuscript, we referred to the criteria (2) and (3) as 'criteria for significance' in two instances. This was changed, as the they consitute decision criteria rather, while the significance of the sample cannot be ultimately assessed.

  l.515: 'neither of the two criteria for significance' was replaced by 'neither of the two criteria for rejecting the null hypothesis'.

  l.525: 'The first criterion for significance is hence not met by any of the pairs. Also, the probabilityfor significance is below 60% for all pairs and all tests as shown by the pie charts, so also the second criterion is missed.' was replaced by

  'Also, the probability for significance is below 60% for all pairs and all tests as shown by the pie charts. Thus, for all proxy pairs and for all tests the formulated decision criteria do not allow to reject the null hypothesis of pairwise unbiased populations.'

- In the same sense, in line 102 we changed:

  'This changes the results from significant to non-significant when compared to averaging out these uncertainties at individual transition lags'

  to

  'If detection uncertainties are averaged out at the level of individual transition lags, temporal delays in the $\delta^{18}$O and Na$^+$ transitions with respect to their counterparts in Ca$^{2+}$ and the annual layer thickness are indeed pairwise statistically significant. In contrast, under rigorous propagation of uncertainty several tests consistently fail to reject the null hypothesis across all considered pairs of proxies.'

- We emphasize more clearly than before that not-rejecting the null-hypothesis does not provide evidence against the alternative hypothesis, in particular given the large uncertainties in the observations. l.561: ' We emphasize that our results must not be misunderstood as evidence against the alternative hypothesis of a systematic lag. In the presence of a systematic lag ($\mu < 0$) the ability of hypothesis tests to reject the null hypothesis of no systematiclag (($H_0 : \mu = 0$)) depends on sample size $n$, the ratio between the mean lag $|\mu|$ and the variance of the population, and on the precision of the measurement. Neither of these quantities is favourable in our case and thus, it is certainly possible that we failed to reject the null hypothesis despite the alternative being true.

**1.2 Statistical Setting**

We are given a sample of observed time lags $\mathbf{\Delta t}^{p,q} = \Delta t_1^{p,q}, ..., \Delta t_n^{p,q}$ between the proxy variables $p$ and $q$, where each observation stems from a different DO event. We assume that the process which generated these time lags is qualitatively the same, such that we can think of the variable $\Delta T$ as a random variable that will assume a specific value $\Delta t$ when a DO event occurs, that is, when the random experiment is performed. The random experiment is characterized by its population $\mathcal{P}_{\Delta T}$. The sample $\mathbf{\Delta t} = (\Delta t_1, ..., \Delta t_n)$ of observations enables us to test hypothesis regarding $\mathcal{P}_{\Delta T}$.

In classical hypothesis testing, the realizations of the random variable $\Delta t_i$ are assumed to be measured (or observed) with infinite precission. That is, each $\Delta t_i$ is assigned a scalar value in the observation process. However, in our case each $\Delta t_i$ can be estimated only with limited precission. Instead of scalar values we characterize the $\Delta t_i$ by means of probability density distribution $\rho_{\Delta t_i}(\Delta t_i^*)$ induced by the linear ramp model.

**Notation**

Due to the uncertainty in the measurement process the true value $\Delta t_i$ must be regarded as a random variable whose probability is characterized by $\rho_{\Delta t_i}(\Delta t_i^*)$. We denote with $\Delta t_i^*$ possible values for the true value. The randomness encoded in $\Delta t_i$ has nothing to do with the initial random experiment $\mathcal{P}_{\Delta T}$ any more, but is only due to measurement uncertainty. We therefore introduced the term 'uncertain variable' in the manuscript and called $\rho_{\Delta t_i}$ an uncertainty distribution.

$\rho_{\Delta t_i}(\Delta t_i^*)$ quantifies the uncertainty about the individual $\Delta t_i$ in a Bayesian sense, indicating how plausible or probable a certain value $\Delta t_i^*$ for $\Delta t_i$ is in view of the data. Importantly, the measurement of the $\Delta t_i$ cannot be repeated. Therefore, the $\rho_{\Delta t_i}(\Delta t_i^*)$ must really be understood as a measure of plausibility in view of the data and not as the probability to obtain a value $\Delta t_i^*$ in a re-peated process. To distinguish between PDF's that characterize a true random experiment in a frequentist understanding and those that quantify uncertainty (in the Bayesian sense), in the manuscript we introduced the term uncertainty distribution for the latter ones.

This being said, one needs to find a way how to incorporate this uncertainty in the assessment of an hypothesis on the population $\mathcal{P}_{\Delta T}$, and in particular in corresponding statistical hypothesis tests. Three different approaches are discussed in our manuscript.

1. Averaging out the uncertainty of the measurement at the level of the individual $\Delta t_i$'s yields a sample comprised of scalars.

$$\mathrm{E}(\boldsymbol{\Delta T}) = (\int \Delta t_1^* \rho_{\Delta t_1}(\Delta t_1^*) \, d\Delta t_1^*, ..., \int \Delta t_{16}^* \rho_{\Delta t_{16}}(\Delta t_{16}^*) \, d\Delta t_{16}^*). \quad (1)$$

   Classical hypothesis tests can be applied to such an expected sample with-out any further ado. However, this approach is associated with a loss of information. Furthermore, it is insensitive to the degree of uncertainty involved in the problem and would thus likely lead to overconfidence and to too easy rejection of null hypotheses. We argue that a setup to test sta-tistical significance must account for uncertainties of the kind treated in our manuscript. In particular it must at least in principle be possible that the uncertainties are so large, that the null-hypothesis cannot be rejected anymore.

The other possibilities rely on the propagation of the uncertainty to the p-value based on the general notion that any function of a random variable constitutes a random variable itself:

$$Y = f(X) \rightarrow \rho_Y(y) = \int \delta(f(x) - y) \, \rho_X(x) \, dx. \quad (2)$$

This allows to compute an uncertain p-value, whose plausibility is indicated by

$$P_{val}(\boldsymbol{\Delta T}) \sim \rho_{P_{val}}(p_{val}^*) = \int \delta(p_{val}(\boldsymbol{\Delta t}) - p_{val}^*) \rho_{\boldsymbol{\Delta T}}(\boldsymbol{\Delta t}) \, d\boldsymbol{\Delta t}, \quad (3)$$

where we use the notation $P_{val}(\mathbf{\Delta T}) \sim \rho_{P_{val}}(p_{val}^*)$ to indicate that the random variable $P_{val}(\mathbf{\Delta T})$ follows the distribution induced by the density $\rho_{P_{val}}(p_{val}^*)$. Note that in Eq. 3 we omitted the intermediate step of propagating the uncertainty to the test statistic. In order to decide between acceptance and rejection of the null-hypothesis based on the uncertain p-value given by the random variable $P_{val}$, we formulated two criteria:

2. If the probability for the uncertain $P_{val}$ to be less than the chosen significance level $\alpha$ exceeds a certain threshold $\eta$,

$$P(P_{val} < \alpha) \overset{!}{>} \eta, \tag{4}$$

the null-hypothesis shall be rejected ($\alpha = 5\%$ and $\eta = 90\%$ were chosen in the manuscript). As explained above, we emphasize here that the PDF $\rho_{P_{val}}(p_{val})$ quantifies the uncertainty that we have about the $p$-value due to the *inability* to measure the individual realizations $\Delta t_i$ precisely. One could therefore also say that, in order to reject the hypothesis, we want to at least be certain to a level of 90% that the true value of the uncertain sample in fact significantly contradicts the null-hypothesis and thus, that the true $p$-value is less than $\alpha$. Otherwise, we prefer not to reject the hypothesis. We would also like to emphasize that a statement on the level of certainty about the significance of the sample with respect to the null hypothesis does not violate the notion of a p-value and its typical interpretation.

3. As a second option we proposed to compare the expected $p$-value

$$\mathrm{E}(P_{val}) = \int \rho_{P_{val}}(p_{val}^*) \, p_{val}^* \, dp_{val}^*, \tag{5}$$

to the a priori chosen significance level. In general, the $p$-value associated with a sample is a measure for the extremeness of the sample with respect to the null-hypothesis. Thus, the expected $p$-value reflects the overall extremeness of the uncertain sample with respect to the null hypothesis. $P(P_{val} < \alpha)$ only gives information about how likely the sample is to contradict the null-hypothesis at the chosen significance-level. $\mathrm{E}(P_{val})$ takes into account how strongly potential values for $\mathbf{\Delta T}$ that are assigned probability larger than zero contradict or support the null hypothesis. Therefore, these two quantities should be considered in combination. We proposed to reject the null-hypothesis if the expected $p$-value is less than $\alpha$

$$\mathrm{E}(P_{val}) \overset{!}{<} \alpha. \tag{6}$$

**1.3   Argument by the referee in favour of criterion (1)**

The referee criticizes that the criteria (2) and (3) do not comply with the general meaning and common understanding of the significance level: 'In the textbook

meaning of the significance level $\alpha$, $\alpha$ denotes the probability that the significance test falsely rejects the null-hypothesis given that it is in fact true. Which is the meaning that almost all readers of the study will be familiar with.'

First, we would like to clarify that the criteria we introduced are 'decision' criteria, they are not significance criteria. The significance of any possible value for $\mathbf{\Delta t}$ is decided based on the comparison of the corresponding $p$-value $p_{val}(\mathbf{\Delta t})$ with the significance level $\alpha$. Propagating the uncertainty associated with $\mathbf{\Delta t}$ we obtain probabilities for both, the sample being significant and the sample being non-significance w.r.t. to the null-hypothesis. A decision must hence be taken between rejecting or not rejecting the hypothesis, even though the significance of the sample cannot be assessed unambiguously. It seems appealing to place the same requirement on the decision criteria as on the significance criterion itself. Namely, that the decision criteria should reject the hypothesis with probability equal to the significance level, as formulated by the referee.

This being said, we would like to clarify that the interpretation of the $p$-value given by the referee is in our view not precise. The statement that $\alpha$ indicates the rate of wrongly rejecting the null-hypothesis may hold in many cases, but it does not hold if a hypothesis on a parameter of a population is formulated in terms of an inequality. E.g. in our manuscript the null hypothesis assumes a population mean $\mu \geq 0$. If $\mu$ happens to be as large as, say, $\mu = 3$, the null-hypothesis is obviously fulfilled. Nevertheless, the chances for rejection are smaller than the significance level. Only in the 'least favourable' case that still complies with the null hypothesis, namely $\mu = 0$, the probability for rejection is equal to the significance-level. The significance level $\alpha$ thus constitutes an *upper limit* for the probability of wrongly rejecting the null-hypothesis.

Given an uncertain sample, can one formulate a decision criterion that decides between rejection and acceptance of the null hypothesis and that rejects wrongly at most with a probability of the significance level? The referee argues that the first decision criterion would posses these characteristics. We will show in the following why this is not the case and why instead the assessment of the null-hypothesis should be based on the criteria (2) and (3) in cases where the sample is uncertain (in the sense defined above).

**1.4 Arguments against criterion (1)**

Whenever one averages out quantification of uncertainty, information is obviously lost. Consider a goalkeeper that knows that an opponent player always shoots the ball either in the left or right corner, but never in the middle. If the keeper averaged over this uncertainty distribution, he would always stay in the middle of the goal but never save the ball. Especially when dealing with bimodal or multimodal distributions, averaging can lead to very misleading results.

**1.4.1 Illustrative example**

To give an example more closely related to the situation in our manuscript, consider the following:

- Let $\mathbf{x} = (x_1, ..., x_n)$ denote the true value of a sample generated from a population $\mathcal{P}_X$ that is known to be Gaussian.

- Assume that in the measurement process you can only observe the absolute values $y_i = |x_i|$. Thus, there is a 50% chance that $x_i = y_i$ and another 50% chance for $x_i = -y_i$ for each individual $x_i$.

- Assume the low standard deviation of the sample $\mathbf{y} = (y_1, ..., y_n)$ allows to almost certainly exclude that the true values $x_i$ have pairwise different signs. Hence the true value of the sample is either $\mathbf{x} = (y_1, ..., y_n)$ or $\mathbf{x} = (-y_1, ..., -y_n)$.

- Assume you want to test the null hypothesis that the population mean $\mu$ is equal to zero.

- If you apply criterion (1) you would average over the uncertainty distribution to obtain the expected sample $\mathrm{E}(\mathbf{X}) = (0, ..., 0)$. Application of the t-test to the averaged sample would in this case always lead to acceptance of the null-hypothesis. Thus, it would neither fulfill the requirement stated by the referee nor deliver any other meaningful insight.

- Application of criterion (2) in turn will yield the correct decision because $p_{val}(y_1, ..., y_n) = p_{val}(-y_1, ..., -y_n)$ and hence, one either finds $P(p_{val} < 0.05) = 1$ or $P(p_{val} < 0.05) = 0$.

- The expected $p$-value that is considered in criterion (3) coincides with the $p$-value of the true value of the sample and would likewise yield the correct decision.

This example shows that averaging out the uncertainty on the level of the observations can lead to meaningless decisions, in particular in the case of not unimodal uncertainty distributions. In fact, the posterior distributions of the transition onsets of individual proxies are multimodal for many of the DO transitions under study (for example see Fig.3 manuscript where the posterior distribution for the transition onset of calcium at the GI-12c onset is shown - the figure is included below). If we strictly followed the paradigm of averaging out uncertainties on the observational level, then this should consequently be done prior to the computation of the transition onset lag $\Delta t^{p,q}$ between the proxies $p$ and $q$, which is a function of the two uncertain transition onset times of the proxies $p$ and $q$. This would again alter the results of the analysis.

**1.4.2 Setup designed by referee 1**

Above we have given an example that motivates the propagation of the measurement uncertainty to the level of the $p$-value and shows that criterion (1) does not necessarily fulfill the requirement suggested by the referee. Here, we discuss the setup introduced by the referee to argue in favour of criterion (1). We argue that the setup must be interpreted slightly differently with the consequence that it cannot serve as an argument for criterion (1).

The referee considers a Gaussian population $P_X = \mathcal{N}(\mu_X = 0, \sigma_X = 1)$ and realizes a sample $\mathbf{x} = (x_1, ..., x_{16})$ from this population. We will denote the true value of the sample by $\mathbf{x}$ in the following. $\mathbf{x}$ corresponds to the $\Delta T_n$ in the referees' comment. Measuring the values $x_i$ necessarily involves uncertainty, which referee 1 models as a second-level normal distribution around the true value. It is important to note here that this second level distribution does not constitute an analogue to the uncertainty distributions $\rho_{\Delta t_i}(\Delta t_i^*)$ in our manuscript, which quantify the plausibility that the true value $\Delta t_i$ equals $\Delta t_i^*$. This is explained in detail below. The most obvious difference is that the distributions introduced by the referee are designed such that the expected value necessarily coincides with the true value, but this cannot be guaranteed in the real application case.

Again, if $x_i$ can be observed directly and without the need to introduce anything like the ramp fit model, the measured value $y_i$ will deviated from the true value $x_i$ due to measurement uncertainty. Typically (as done by the referee) one assumes a Gaussian distribution for the probability to measure $y_i$ if the true value is $x_i$:

$$\mathcal{P}_{Y_i|x_i}(Y_i|x_i) \sim \mathcal{N}(x_i, \sigma_{\mathrm{obs}}), \tag{7}$$

where $\sigma_{\mathrm{obs}}$ quantifies the measurement uncertainty in the setup. Correspondingly, the error $\Delta_i = x_i - Y_i$ that you make in the measurement follows a normal distribution as well

$$\mathcal{P}_{\Delta_i|x_i} = \mathcal{N}(0, \sigma_{\mathrm{obs}}). \tag{8}$$

Generally, $\mathcal{P}_{\Delta_i|x_i}$ is in fact independent of $x_i$ such that $\mathcal{P}_\Delta = \mathcal{N}(0, \sigma_{\mathrm{obs}})$ conveniently describes the errors $(\Delta_1, ..., \Delta_{16})$ as i.i.d. random variables.

If $\sigma_{\mathrm{obs}}$ is known, from a measured value $y_i$ you can in turn deduce a probability distribution that quantifies the probability that the true value $x_i$ is given by some $x_i^*$:

$$P_{X_i^*}(x_i = x_i^*|y_i) \sim \mathcal{N}(y_i, \sigma_{\mathrm{obs}}). \tag{9}$$

We see here that the expectation of this distribution, $\mathrm{E}(X_i^*)$, does not coincide with the true value $x_i$ but instead with the measured value $y_i$. These considerations are illustrated in Fig. 1.

We assume that the referee had in mind something like repeated measurements of the true value. Repeated measurement would indeed correspond to sampling from the distribution of measured values given a true value $\mathcal{P}_{Y_i|x_i}$. However, the situation in our study is such that the uncertainty distribution for the true value must be quantified after a single measurement. Assuming that multiple measurements of individual true values $x_i$ were possible, then the measured values $y_{i,j}$ would in the referees setup be distributed normally around the true value (see Eq. 7) where $j$ indicates the j-th repetition of the measurement of the i-th true value. From the set of $m$ measurements of $x_i$: $\mathbf{y_i} = (y_{i,1}, ..., y_{i,m})$ the mean of the distribution $\mathcal{P}_{Y_i|x_i}$ - and hence the true value $x_i$ - can be estimated. Either a point estimate can be derived as done by the referee by taking

$$x_i^* = \bar{Y}_i^m = \frac{1}{m} \sum_{j=1}^m y_{i,j} \tag{10}$$

[Figure]

Figure 1: Illustration of measurement uncertainty. On the top level, there is the population that characterizes the random experiment. From this population, three true values are realized, measuring each is associated with uncertainty. The probability to measure $y_i$ if $x_i$ is the true value is shown in light gray in the second level. When the measurement is executed, one value is realized from each of these distributions - the measured value which is indicated in dark green. Given that the uncertainty of the measurement process is known, one can then specify an uncertainty distribution for the true value, based on the measured value (green Gaussian distributions in the bottom panel).

or a distribution $\mathcal{P}_{X_i^*|y_{i,1},...,y_{i,m}}(x_i^* = x_i)$ for the estimator $x_i^*$ can be derived by means of the t-distribution (see original version of the manuscript). This uncertainty distribution, which estimates the true value $x_i$ based on $m$ measurements $y_{i,1},...y_{i,m}$, is much narrower than the distribution $\mathcal{P}_{Y_i|x_i}$. So in the example provided by the referee, the criterion (1) should be compared with the application of the criteria (2) and (3) to the uncertainty distributions that can be derived for the true value from repeated measurement via the t-distribution (assuming that the distribution for $\Delta$ were Gaussian). The referee might have had in mind that in the case of repeated measurement a test statistic $T(\mathbf{x})$ should be computed for each measured sample $T_j = T(y_{1,j},...,y_{16,j})$ and that the $T_1,...,T_m$ would correspond to a set of $p$-values $p_{val_1},...,p_{val_m}$. In that case the $\{p_{val_j}\}_j$ could be interpreted as a representation of the uncertainty on the $p$-value, but this is not the case in our situation. One aims to compute the $p$-value that corresponds to the true sample $x_1,...,x_{16}$. As mentioned before, multiple measurement of each $x_i$ substantially reduced the uncertainty on each $x_i$. Thus, before computing the test statistic, the uncertainty distribution $P_{X_i^*|y_{i,1},...,y_{i,m}}$ can be computed, which will be a t-distribution centered around $\bar{Y}_i^m$. The width of this distribution will be significantly smaller than $\sigma_{\mathrm{obs}}$. The remaining uncertainty should then be propagated according to Eq. 19.

**1.4.3 $\chi^2$-test**

We have shown above that what the referee interprets as the uncertainty distribution is in fact not the uncertainty distribution, but the distribution of the measured value $Y_i$ around a true value $x_i$. Furthermore, we explained why in case of repeated measurements one should not compute $p$-values for each measured sample $y_{1,j},...y_{16,j}$, but should instead use the multiple measurements to infer uncertainty distributions $\mathcal{P}_{X_i^*|y_{i,1},...,y_{i,m}}$ for the estimator of true values $x_i^*$. This uncertainty is small compared to $\sigma_{\mathrm{obs}}$ and should then be propagated in the sense of Eq. 19. We would thus politely like to argue that the referee compares mathematical objects that cannot be compared. If the 6000 samples from $\mathcal{P}_{Y_i|x_i}$ really were to be interpreted as multiple measurements of the true value, then proper propagation of the uncertainty to $\mathcal{P}_{X_i^*|y_{i,1},...,y_{i,m}}$ would indeed yield rejection rates of 5% for the criteria (2) and (3) in the setup designed by the referee.

For the case of a single measurement we will show how under the correct interpretation of the measurements $y_i$ criterion (1) in addition does not reject a true null-hypothesis at a 5% rate, as stated by the referee. We will discuss a $chi^2$-test, because in the particular case of the t-test the 5% rejection holds true due to the symmetry of the problem.

Consider a normally distributed random variable $X$ with $\mathcal{P}_X = \mathcal{N}(\mu_X = 0, \sigma_X)$ and a sample $\mathbf{x} = (x_1,...,x_n)$ of $n$ realizations. Furthermore, assume that the measurement errors are distribution normally as above

$$\mathcal{P}_\Delta = \mathcal{N}(0, \sigma_{\mathrm{obs}}), \tag{11}$$

such that the measured value $Y_i = x_i + \Delta_i$ corresponding to the true value $x_i$ is

distributed according to $\mathcal{P}_{Y_i|x_i} = \mathcal{N}(x_i, \sigma_{\mathrm{obs}})$. We distinguish the following two situations:

1. single measurement of the true sample $\mathbf{x} = (x_1, ..., x_n)$ yields a single realization of the measured sample $\mathbf{y} = (y_1, ..., y_n)$

2. repeated measurement of the true sample yields for each true value $x_i$ a set of measured values $\mathbf{y_i} = (y_{i,1}, ..., y_{i,m})$.

We will first discuss the case of a single measurement, because this is the one in analogy to the manuscript. Without any knowledge about the true values $x_i$ the unconditional distribution for the random variables $Y_i$ is given by the convolution of $\mathcal{P}_X$ and $\mathcal{P}_\Delta$

$$\mathcal{P}_Y = \mathcal{N}(0, \sigma_X) \circledast \mathcal{N}(0, \sigma_{\mathrm{obs}}) = \mathcal{N}(0, \sqrt{\sigma_X^2 + \sigma_{\mathrm{obs}}^2}). \tag{12}$$

Hence, measuring a single sample of true values is effectively the same as sampling from a normal distribution with standard deviation $\sigma_Y = \sqrt{\sigma_X^2 + \sigma_{\mathrm{obs}}^2}$ and mean $\mu_Y = 0$. Suppose you were given one measured sample $\mathbf{y} = (y_1, ..., y_n)$ As explained above, knowing $\sigma_{\mathrm{obs}}$ allows you to quantify an uncertainty distribution for the true value from this measurement:

$$\mathcal{P}_{\mathbf{X}^*|\mathbf{y}} = \mathcal{N}(\mathbf{y}, \sigma_{\mathrm{obs}}) \quad \text{or componentwise} \quad \mathcal{P}_{X_i^*|y_i} = \mathcal{N}(y_i, \sigma_{\mathrm{obs}}) \tag{13}$$

This means that, given $\mathbf{y}$, the probability that the true value equals $\mathbf{x}^*$ is normally distributed around the measured $\mathbf{y}$, with standard deviation $\sigma_{\mathrm{obs}}$.

Now suppose you wanted to test the null-hypothesis $H_0 : \sigma_X < \sigma_0$, knowing that $X$ is normally distributed. Under this condition, the test statistic

$$\chi^2 = \frac{(n-1)s^2}{\sigma_0^2} \tag{14}$$

follows a $\chi^2$ distribution of $n-1$ degrees of freedom. Here, $s^2 = \frac{1}{n-1}\sum_{i=1}^n (x_i - u)^2$ denotes the samples' variance with sample mean $u = \frac{1}{n}\sum_{i=1}^n x_i$. $\sigma_0$ is the variance that defines the null-hypothesis. In a one-sided significance test $\alpha = 0.05$ the most extreme 5% at the high end of the possible values for $\chi^2$ comprise the rejection region $\Omega_K$. Referee 1 now proposed to first collapse the uncertainty distribution $\mathcal{P}_{\mathbf{X}^*|\mathbf{y}}$ to its expected value

$$\mathrm{E}(\mathbf{X}^*) = \mathbf{y} \tag{15}$$

and subsequently apply the hypothesis test. It becomes clear that in this setup one effectively tests the standard deviation $\sigma_Y = \sqrt{\sigma_X^2 + \sigma_{\mathrm{obs}}^2}$ instead of the desired $\sigma_X^2$. For $\sigma_{\mathrm{obs}}^2 \sim \sigma_X^2$ one finds $\sigma_Y \sim \sqrt{2}\sigma_X$. If then $\sigma_X \lesssim \sigma_0$ the probability to reject the hypothesis easily rises above the chosen significance level under the use of criterion (1), since the effective standard deviation exceeds the hypothesized standard deviation by a factor of $\sqrt{2}$ even though the null-hypothesis is in fact true. Importantly, we see here that large observational

[Figure]

Figure 2: Figure 3 from the manuscript

uncertainty limits the ability to test hypothesis. In the setup discussed by referee 1, the results obtained by the application of criterion (1) were robust against the measurement uncertainty which contradicts our physical intuition. Please note that all criteria converge to the classical hypothesis testing case when $\Delta$ or more generally speaking, the uncertainty tends to zero.

In the case where repeated measurements of the true value $x_i$ are possible one can estimate the true value $x_i$ with higher precision by averaging over the observed values

$$\bar{Y}^m = (Y_{i,1}, ..., Y_{i,m}). \tag{16}$$

For a given $x_i$ the variable $\bar{Y}^m$ will be distributed around $x_i$ with variance $\frac{\sigma_{\text{obs}}^2}{m}$. Hence, the generic random variable $\bar{Y}^m$ is normally distributed around zero with an effective standard deviation of $\sigma_{\bar{Y}^m} = \sqrt{\sigma_X^2 + \frac{\sigma_{\text{obs}}^2}{m}}$. The mechanism is the same as above and again, the chances to wrongly reject the null-hypothesis may be higher than 5% if one would follow criterion (1), because the effective variance exceeds the true variance. The impact of the measurement uncertainty is reduced by the repeated measurement.

**1.5 Point-by-Point answer to the referee**

Additionally, prompted by the authors statement about the novelty of their "uncertainty propa- gation to p-values" in their response: A cursory literature search (keywords fuzzy p-values, bayesian p-values) brought up a range of papers that seem to be dealing with p-values in settings similar to the setting the authors deal with here. It would be good if the authors set their approach into the context of aforementioned literature or highlight

its conceptual differences should the they decide to further employ it in the next iteration of the study.

During our work on the manuscript, we encountered the concept of fuzzy $p$-values as well. However, we found that it did not match our case precisely. Filtzmoser (2004) writes with regards to fuzzy data:

> Real observations of continuous quantities are not precise numbers but more or less non-precise. The best description of such data is by so-called non-precise numbers. Such observations are also called fuzzy. The fuzziness is different from measurement errors and stochastic uncertainty. It is a feature of single observations from continuous quantities. Errors are described by statistical models and should not be confused with fuzziness. In general fuzziness and errors are superimposed.

Instead of PDF's, in the fuzzy $p$-value theory the uncertainty about data is expressed in terms of characteristic functions. Also an adoption of this concept to our case fails due to the properties that characteristic functions are required to fulfill (see for example Filtzmoser, 2004 and Parchami, 2008).

However, we agree with the referee that fuzzy $p$-values are worth mentioning in the context of our work and added the sentence 'The theory of fuzzy $p$-values is in fact concerned with uncertainties either in the data or in the hypothesis, however, it is not applicable to measurement uncertainties that are quantifiable in terms of probability density functions' in line 569.

Regarding the Bayesian $p$-value, Gelman et al. (2004) write:

> *Posterior predictive p-values.* To evaluate the fit of the posterior distribution of a Bayesian model, we can compare the observed data to the posterior predictive distribution. In the Bayesian approach, test quantities can be functions of the unknown parameters as well as data because the test quantity is evaluated over draws from the posterior distribution of the unknown parameters. The Bayesian p-value is defined as the probability that the replicated data could be more extreme than the observed data, as measured by the test quantity:
>
> $$p_B = Pr(T(y^{rep}, \theta) \geq T(y, \theta)|y), \qquad (17)$$
>
> where the probability is taken over the posterior distribution of $\theta$ and the posterior predictive distribution of $y^{rep}$ (that is, the joint distribution, $p(\theta, y^{rep}|y)$:
>
> $$p_B = \int \int I_{T(y^{rep}, \theta) \geq T(y, \theta)} p(y^{rep}|\theta) p(\theta|y) dyrepd\theta, \qquad (18)$$

where $I$ is the indicator function. In this formula, we have used the property of the predictive distribution that $p(y^{rep}|\theta, y) = p(y^{rep}|\theta)$.

Hence, the Bayesian $p$-value assesses the fit of a model to data. In our application, that would be the fit of the linear-ramp model to the transition data. But this concept does not seem applicable for assessing the significance of the uncertain sample of transition onset lags with respect to the null-hypothesis.

L 129ff It is not true that Erhardt et al. (2019) only considered time series free from data-gaps. In fact, looking at Figure 3 in the presented manuscript, both the Ca 2+ as well as the Na + data series do in fact exhibit at least one section of missing data. Please reformulate.

In the caption of their Fig.3 Erhardt et al write 'No timing results are given for transitions where there are data gaps in one of the necessary datasets.' However, it is true that DO events with minor gaps in the data around the transition are used for the analysis nonetheless.

We corrected the statement, which now reads: 'For their analysis, Erhardt et al. (2019) only considered time series around DO events that do not suffer from **substantial** data gaps.'

L 198ff The choice of terms for the different type of distributions is a little bit misleading. Posterior distributions as generate by Bayesian inference are probability distributions. Yes, posterior distributions carry the uncertainty about an inferred parameter conditional on the data and the model, but they are probability distributions nontheless. By using the neologism "uncertainty distributions" the authors implicate that they are more uncertain than their "probability distributions" generated by random experiments. This sets the tone for the discussion that follows in a very odd way. The authors should use the correct term "posterior probabilities". Changing this would also avoid such contortions as the "high uncertainty probabilities" (L 305)

The referee is certainly correct that mathematically, there is no difference between the distributions that we term 'uncertainty distributions' and standard probability distributions represented by probability density functions (PDFs). However, their interpretation is somewhat different: a PDF that characterizes a random experiment can be thought of as the probability to observe a certain value of the random variable in a repeated next execution of the experiment. Contrarily, an uncertainty distribution is a measure of plausibility but the uncertain variable cannot be observed repeatedly. We believe that the term 'uncertainty distribution' is a useful way to highlight this difference and in fact helps the reader not to confuse the different origins of randomness involved in the analysis. We would

therefore like to keep using the term 'uncertainty distribution' and have not changed this in the revised manuscript.

For sake of clarity, and to avoid ambiguous notation, we have marked all uncertain quantities with a hat - that is, all random variables that inherit their randomness from the Bayesian transition onset detection. The same holds true for potential values they might assume. We added explanation on how true values and uncertain values are related and how the two-level randomness must be understood in line 267.

'The left panel in Fig. 4 illustrates this situation: from an underlying population $\mathcal{P}_X$ a sample $\mathbf{x} = (x_1, ..., x_6)$ is realized, with the $x_i$ denoting the true values of the individual realizations. However, the exact value of $x_i$ can not be measured due to measurement uncertainties. Instead an estimator $\hat{Y}_i$ is introduced together with the uncertainty distribution $\rho_{\hat{Y}_i}(\hat{y}_i)$ that expresses the observers belief about how likely a specific value $\hat{y}_i$ for the estimator $\hat{Y}_i$ is to agree with the true value $x_i$. The $\hat{Y}_i$ correspond to the $\Delta \hat{T}_i^{p,q}$. For the $x_i$ there is no direct correspondence in the problem at hand, because this quantity in practice cannot be excessed and hence must not be denoted explicitly in the practical case. We call the vector of estimators $\hat{\mathbf{Y}} = (\hat{Y}_1, ..., \hat{Y}_n)$ an uncertain sample in the following.'

Also, in the cause of this, the notation especially in Section 3 changed in many places.

However, we agree that 'high uncertainty probability' is not a well comprehensible term and have thus replaced 'that is, we observe high uncertainty probabilities for negative $\Delta T_i$ across the sample according to Eq. 7' with 'that is, the corresponding uncertainty distribution indicate high probabilities for negative $\Delta T_i$ across the sample according to Eq. 7.' (l.326)

L 208ff Prescribing a "fixed pattern of causes and effects" is an overly strong interpretation. It is quite easy to imagine a range of mechanisms that have an indistinguishable imprint in the proxy record but trigger a transition from stadial to interstadial conditions. For example the ocean processes alone, that the authors list in the introduction are probably very difficult to distinguish using the proxies presented here as they partly invoke very similar feedback mechanisms. I would suggest to reformulate "similar pattern of cause and effects" to convey the possible ambiguity as a discussion of the imprint in the proxy records by the different causes clearly goes beyond the focus of this manuscript. The same holds true for later implications of the one trigger of DO-Events that the authors make throughout the manuscript.

The referee is of course right that different mechanisms could easily have left the same or at least an indistinguishable imprint in the proxy data. However, we express very clearly that the assumed one to one mapping

– $\delta^{18}$O → temperature

- $Ca^{2+} \rightarrow$ state of the atmosphere
- $Na^+ \rightarrow$ sea ice
- $\lambda \rightarrow$ local precipitation

is potentially oversimplified and that we leave the discussion about the correct proxy interpretation to the experts. In order to think of the $\Delta t_i$ from different DO events as an i.i.d. random variable, one *has to* assume that all DO events were triggered by the same physical process, which in turn necessarily 'prescribes a fixed pattern of causes and effects for all DO events' - at least on the scale of interaction between the climatic subsystems represented by the proxies. In turn, if the pattern of causes and effects was different between DO events, then the physical mechanism was not the same and we could not treat $\Delta t$ as an i.i.d. random variable.

For clarity, we added: 'at least on the scale of interaction between climatic subsystems represented by the proxies under study' to the corresponding sentence in line 213.

L 210 The imperative "should" in regards to the setup of the frequentist analysis that follows should be replaced with "could" or "can".

Thank you, we fully agree and replaced 'should' with 'can'.

L 220 Either "be" or "bear".

Corrected: 'bear'.

L 224 The footnote should either be added to and discussed in the paper or removed. As it is right now it is just a clever remark that does not contribute to the overall manuscript.

The footnote is very technical and addresses the mathematically interested reader. Placing the statement in a footnote clearly signals that it this content is not required to follow the reasoning in the main text. If the editor thinks that using footnotes in this way is not adequate, we will be glad to incorporate this footnote into the main text.

L 263 I think it would be better to say that the sample no longer only carries the randomness of the population. As would be the case for a regular statistical test with certain values.

The formulation 'the sample no longer only carries the randomness of the population' suggests that now the sample carries the randomness of the population and the randomness due measurement uncertainty simultaneously. However, the sample of DO transition onset lags has already been realized and therefore does not carry the randomness of the population any more. It only carries the randomness due to the measurement.

L 284 In null-hypothesis significance testing the null-hypothesis can only be rejected.

With regards to this, Romano and Lehmann (2006) write:

> We now begin the study of the statistical problem that forms
> the principal subject of this book, the problem of hypothesis
> testing. As the term suggests, one wishes to decide whether or
> not some hypothesis that has been formulated is correct. The
> choice here lies between only two decisions: accepting or reject-
> ing the hypothesis. A decision procedure for such a problem is
> called a test of the hypothesis in question.

The term 'null-hypothesis' is used to expressed that this hypothesis entails
the 'no effect', 'no difference' or 'no causal relation' that one aims to
reject. If the test fails to reject the null-hypothesis it must certainly be
acceptance, for the time being. However, in the revised manuscript we
highlight more strongly than before that in our case we cannot reject the
null-hypothesis due to the large uncertainties and that this result should
not be interpreted as evidence against the alternative (see l.562ff).

At the beginning of the discussion section some text parts have been moved
around and some parts have been shortened. Now the section reads more
consise while the content has not changed.

In the previous version we used the term 'biased' to characterize a pop-
ulation with mean different from zero. Since a 'bias' in statistics usually
means a systematic distortion of measurements, we replaced the terms
'biased' and 'unbiased' with the terms 'non-neutral' and 'neutral', respec-
tively.

**References**

Peter Filzmoser and R. Viertl: Testing hypotheses with fuzzy data: The fuzzy
p-value. Metrika: International Journal for Theoretical and Applied Statistics,
2004, vol. 59, issue 1.

Abbas Parchami, Mahmoud Taheri and Mashaallah Mashinchi: Fuzzyp-value
in testing fuzzy hypotheseswith crisp data. Stat Papers (2010) 51:209–226.

Lehmann, E. L. & Romano, J. P. Testing Statistical Hypothesis. Design vol.
102 (Springer US, 2006).

Erhardt, T. et al. Decadal-scale progression of the onset of Dansgaard-Oeschger
warming events. Clim. Past 15, 811–825 (2019).

**2 Answer to Referee 2**

**2.1 General Remarks**

First of all we would like to thank the referee for the careful second review. Be-
fore we address the comments point by point, we will discuss the main critizism,

which is why we do not use a mixture distribution to statistically assess the significance of the sample of uncertain DO time lags. The simple answer is: we do, but we failed to make this clear earlier. We therefore apologize for our reservation towards this comment, which was brought up by the referee already in the first review and which we had misunderstood. Accordingly the paragraph from line 572 onwards (previous manuscript) was removed in the revised version.

The referee proposes to use a mixture distribution to test whether the population mean is greater than or equal to zero. In fact, the bootstrap test that we carry out does exactly this, though it is not immediately obvious. Given a sample $\mathbf{x} = (x_1, ..., x_n)$ generated from a population $\mathcal{P}_X$, the idea behind bootstrapping is that the empirical distribution (or mixture distribution as termed by the referee) $\bar{\rho}_X(x) = \frac{1}{n} \sum_{i=1}^{n} \delta(x - x_i)$ approximates the population to a certain degree. Let $H_0 : \mu_X \geq 0$ denote the null-hypothesis that the mean $\mu_X$ of the population $P_X$ is greater than or equal to zero. In the absence of any further information on the populations shape, a null-distribution for testing the hypothesis can be constructed by modifying the mixture distribution such that it complies with the null-hypothesis.

In order to test the mean of a population, Lehmann and Romano (2006) propose the use of the test statistic

$$T^n = n^{1/2} \, u, \tag{19}$$

with $u = \frac{1}{n} \sum_{i=1}^{n} x_i$ denoting the sample mean. To construct a null-distribution for the test statistic, they take the mixture distribution shifted by the sample mean

$$\rho_{X'}^n(x') = \frac{1}{n} \sum_{i=1}^{n} \delta(x' - x_i + u), \tag{20}$$

such that $\rho_{X'}^n(x')$ has mean 0. Hence, it fulfills the null-hypothesis while simultaneously capturing the characteristics of the mixture distribution. Furthermore, this choice guarantees that 1) $\rho_{X'}^n \overset{n \to \infty}{\Rightarrow} \rho_{X'}$ such that $\rho_{X'}$ has mean 0 as well (criteria for convergence are given in Lehmann and Romano (2006)), and 2) that in this limit the probability for rejection is not higher than $\alpha$ (the chosen significance level) for any original population $P_X$ that fulfills $H_0$. For testing the inequality considered in $H_0$, the case where the null-distribution has zero mean is the decisive one. From the shifted mixture distribution $\rho_{X'}^n$ a data-driven null-distribution can be computed by resampling $m$ samples of size $n$ $\mathbf{X}_j^* = (x_1^*, ..., x_n^*)_j$ from $\rho_{X'}^n$ and computing the test statistic $T_j^n = T^n(X_j^*)$ for each of these 'synthetic samples'. For a given significance level $\alpha$ the $1 - \alpha$-th percentile of the set $\{T_1^n, ..., T_m^n\}$ establishes a rejection region for the original $T^n(\mathbf{x})$.

In the previous version of our manuscript we used the sample mean itself as a test statistic. This has been corrected and we now use the statistic $T^n$ as proposed by Lehmann and Romano (2006). Furthermore, in our study the original sample $\mathbf{x}$ as well as its sample mean $u(\mathbf{x})$ and its corresponding test statistic are uncertain. This uncertainty is propagated rigorously through all the steps described above.

When the referee proposed to use the mixture distribution to 'test whether the population mean is greater or equal zero', he or she might have had in mind to simply resample samples of size $n$ $\mathbf{X}_j^* = (x_1^*, ..., x_n^*)_j$ from a non-shifted mixture distribution. Then one could compute their means $u_j^* = \frac{1}{n} \sum_{i=1}^{n} x_{i,j}^*$ and compare the $\alpha$-th percentile to the mean of the null hypothesis $\mu_0 \geq 0$.

Lehmann and Romano (2006) and Hall and Wilson (1991) provide arguments why for rigorous hypothesis testing the shifting of the mixture distribution is required. While taking this into account, effectively we indeed use the mixture distribution to assess the significance of the given sample with respect to the null-hypothesis, just as proposed by the referee.

**2.2 Point by Point answer to the referee**

L380 In response the referee's remark on using the mixture distribution, we rewrote section 3.3.3 and changed the test statistic. Accroding to these changes, Figures 6. and 7. as well as Table 1 and Table B1 were updated.

L86ff "In order to review the statistical evidence for a potential systematic lags, we formalize the notion of a 'systematic lag': We call a lag systematic if it is enshrined in the random experiment in form of a population mean different from zero. Samples generated from such a biased population would systematically (and not by chance) exhibit sample means different from zero. Accordingly, we formulate the null hypothesis of a pairwise unbiased transition sequence, that is, a population mean equal to zero." Could this maybe be reformulated, in order to also acknowledge the fact that whether or not a truly biased sample can be expected to systematically exhibit sample means (significantly) different from zero depends on the sample size? To me, this seems to be an important motivation for the procedure proposed here.

We agree that the sample size is one of the factors that limits the ability to detect a systematic lag between the proxy variables. The probability to reject the null-hypothesis assuming a truly biased population is a non-trivial quantity and depends on the sample size, the strength of the bias, and the variance of the population simultaneously. Since these factors interact in a non-trivial way, we do not believe this point should be raised already at the stage were we still aim to properly define the statistical problem. Instead we have included a paragraph in the discussion that discusses the chance to reject the null-hypothesis.

Caption Fig. 2 I was not sure at first whether the posteriors shown are from the authors or from Erhardt et al. Maybe this could be made specifically clear in the caption.

We reproduced all posterior probability density estimates adopting the method and the data provided by Erhardt et al. We agree that this was not made sufficiently clear in the caption. Hence, we changed the sentence

'The probability density estimates for the transition onsets with respect to the timing of the DO event according to Rasmussen et al. (2014) are shown in arbitrary units for all proxies' to 'The posterior probability densities for the transition onsets with respect to the timing of the DO event according to Rasmussen et al. (2014) are shown in arbitrary units for all proxies. They were recalculated using the data and the method provided by Erhardt et al. (2019).'

L187  indipendent → independent

Corrected.

L797-798  I don't understand what is meant here, maybe just a grammatical error?

We are not quite sure we understand what the referee finds problematic here. We changed the sentence 'For a given proxy pair the starting point for the statistical analysis however, is the uncertain sample $\mathbf{\Delta T}^{p,q} = (\Delta T_1^{p,q}, ..., \Delta T_n^{p,q})$ characterized by the $n$ dimensional uncertainty distribution $\rho_{\mathbf{\Delta T}}^{p,q}(\mathbf{\Delta t}^{p,q}) = \prod \rho_{\Delta T_i^{p,q}}(\Delta t_i^{p,q}).$' to 'For a given proxy pair the starting point for the statistical analysis is, however, the uncertain sample $\mathbf{\Delta T}^{p,q} = (\Delta T_1^{p,q}, ..., \Delta T_n^{p,q})$, **which is** characterized by the $n$-dimensional uncertainty distribution $\rho_{\mathbf{\Delta T}}^{p,q}(\mathbf{\Delta t}^{p,q}) = \prod \rho_{\Delta T_i^{p,q}}(\Delta t_i^{p,q}).$' and hope that this clarifies the statement.

At the beginning of the discussion section some text parts have been moved around and some parts have been shortened. Now the section reads more consise while the content has not changed.

**References**

Lehmann, E. L. & Romano, J. P. Testing Statistical Hypothesis. Design vol. 102 (Springer US, 2006).

Erhardt, T. et al. Decadal-scale progression of the onset of Dansgaard-Oeschger warming events. Clim. Past 15, 811–825 (2019).

---

## Author Response (AR3)

Dear Laurie,

first, we would like to thank you for editing our paper and coordinating this detailed and fruitful review process. We appreciated your last set of comments and implemented the proposed changes. There are just two points we would like to mention:

- l.4: You rightfully noted, that the sea ice retreat can be evidenced for Nordic Seas. Apparently, for the North Atlantic, it is harder to find evidence from marine cores for sea ice extent during stadials and interstadial. Erhardt et al. introduce sodium as a proxy for North Atlantic sea ice:
  - 'In turn, this allows us to interpret the stadial-interstadial changes in sodium concentrations in the ice cores as qualitative indicators of the extent of the sea-ice cover in the North Atlantic.' (Erhardt et al., 2019)

In the manuscript we follow the proxy interpretation as provided by Erhardt. Also, our description of DO events is by far not complete in term of changes that accompany Greenland warming. Thus, we think it is reasonable to not mention the Nordic Seas sea ice dynamics explicitly, even though these are better established than the North Atlantic sea ice dynamics.

- l.55: you proposed to replace 'According to this proxy variable interpretation' with 'According to these proxy records'. Here, we intentionally chose the word 'interpretation'. With this, we want to express, that the statement on DO events that follows, is conditioned on the interpretation of the proxy variables. If this makes sense to you, we would like to keep the word 'interpretation', but replace 'variable' with 'records'.
  - According to this proxy record interpretation, DO events are found to comprise not only sudden warming, but also sudden increase in local precipitation amounts, retreat of the Nordic Seas' and North Atlantic sea ice cover, and changes of hemispheric circulation patterns.
- l309: You questioned, if the example given to motivate uncertainty propagation was a good one. We agree, that the provided example is in fact not very close to the study and thus we replaced it. While the former example aimed at showing how averaging uncertainties before testing leads to meaningless results, the new example makes a slightly different point but is more related to the study: It highlights the fundamental fact, that large uncertainties in the measurement prevent detailed inference on the population. That is, even in the presence of arbitrarily large uncertainties, the scheme of averaging first and testing second, suggests that one could still take an informed decision about rejecting or accepting the null hypothesis. However, this cannot be true and is thus a fundamental inconsistency of this scheme. We hope that the revised manuscript now conveniently motivates the need for uncertainty propagation by pointing toward this inconsistency that arises when uncertainties are averaged out.
- In the previous version of the manuscript, in the introduction we formulated the null hypothesis 'that pairwise no transition sequence is physically favoured' (l.99 in the version edited by you) which corresponds to a population mean equal to zero. Later in the manuscript we slightly modified the null hypothesis and made it one-sided: 'testing if the sample favours no or the opposite lag' (l.260 in the edited version).
  - Now, we reconciled this, and changed the abstract and introduction accordingly:
    - 'with respect to the null hypothesis that the proposed transition order is in fact not systematically favoured' (1.9)
    - 'Accordingly, we formulate the null hypothesis that the proposed transition sequence is in fact not physically favoured. In mathematical terms this corresponds to an underlying

population of lags with mean equal to zero or with reversed sign with respect to the observed lags.' (1.89)

- 'Here, we formalize the investigation of systematic lead-lag relationships between the proxy transitions. The random experiment framework allows to relate a suspected transition sequence to a mean of the generating population P ΔT which differs from zero in the according direction. Evidence for the suspected sequence can then be achieved by testing the null hypothesis of a population mean equal to zero or opposed to the suspected lag direction. If this null hypothesis can be rejected based on the observations, this would constitute a strong hint for a systematic, physical lag, and would hence potentially yield valuable information on the search for the mechanism(s) and trigger(s) of the DO transitions.' (1.237)
- We slightly modified the table B1 for sake of clarity and added the table in csv format as a supplement
- Other then that, we adopted all proposed changes.

all the best, Keno